# Aurora A regulates the material property of spindle poles to orchestrate nuclear organization at mitotic exit

Ashwathi Rajeevan[1,3], Vignesh Olakkal [1,3], Madhumitha Balakrishnan [1,3], Dwaipayan Chakrabarty[1,3], François Charon [2], Daan Noordermeer [2] & Sachin Kotak [1✉]

## Abstract

**Animal cells dismantle their nuclear envelope (NE) at the beginning and reconstruct it at the end of mitosis. This process is closely coordinated with spindle pole organization: poles enlarge at mitotic onset and reduce in size as mitosis concludes. The significance of this coordination remains unknown. Here, we demonstrate that Aurora A maintains a pole-localized protein NuMA in a dynamic state during anaphase. Without Aurora A activity, NuMA shifts from a dynamic to a solid state and abnormally accumulates at the poles, causing the segregated chromosome sets to bend around the NuMA-enriched poles. NuMA localization at the poles relies on interactions with dynein/dynactin, its coiled-coil domain, and an intrinsically disordered region (IDR). Mutagenesis experiments revealed that cation-$\pi$ interactions within IDR are key for NuMA pole localization, while glutamine residues trigger the solid-state transition of NuMA upon Aurora A inhibition. We propose that maintaining the proper material properties of the spindle poles is a key step in choreographing the accurate organization of the nucleus and genome post-mitosis.**

**Keywords** Aurora A; NuMA; Material Property; Spindle Poles; Nuclear Shape
**Subject Categories** Cell Adhesion, Polarity & Cytoskeleton; Cell Cycle

## Introduction

The vast majority of eukaryotic cells maintain and protect their genetic information within a single nucleus for most of the cell cycle. Within the nucleus, chromosomes adopt a complex 3D organization (Cremer and Cremer, 2010), and changes in nuclear shape can influence gene regulation (Akhtar and Gasser, 2007; Almonacid et al, 2019; Tajik et al, 2016). Indeed, changes in nuclear shape can be harmful and often associated with aging and cancer

(Zwerger et al, 2011; Chow et al, 2012). Thus, understanding the mechanisms that regulate nuclear shape is crucial for maintaining long-term cellular function in both health and disease.

In cycling somatic animal cells, mitotic entry is marked by genome condensation and nuclear envelope (NE) disassembly (Antonin et al, 2016). Concurrently, spindle poles (referred to as "poles"), which include centrosomes and associated proteins, are reinforced to generate robust microtubule asters that assemble the mitotic spindle, ensuring efficient and timely chromosomes capture for their segregation. In contrast, during mitotic exit, the poles are restructured, their microtubule-nucleating capacity diminishes, and the nuclear envelope reassembles. At the poles, centrosomes consist of centrioles surrounded by a dynamic protein mass called pericentriolar material (PCM) matrix (Bornens, 2002; Woodruff et al, 2014; Conduit et al, 2015). Centrosomes increase their microtubule nucleation capacity at mitotic onset due to a rapid increase in the PCM matrix (Palazzo et al, 2000). The temporal accumulation of several PCM-localized proteins, including pericentrin (PCNT) (PLP in *Drosophila melanogaster*), Cep192 (SPD-2 in *Caenorhabditis elegans;* Spd-2 in *D. melanogaster*), Cdk5Rap2 (SPD-5 in *C. elegans*, Cnn in *D. melanogaster*), is required for PCM assembly (Woodruff et al, 2014; Conduit et al, 2015). This rapid PCM expansion is controlled by the activity of regulatory kinases, such as Cyclin-dependent kinase (Cdk1), Polo-like kinase (Plk1), and Aurora A (Hannak et al, 2001; Hamill et al, 2002; Haren et al, 2009; Joukov et al, 2014; Conduit et al, 2014; Woodruff et al, 2017; Kapoor and Kotak, 2019; Ohta et al, 2021; Watanabe et al, 2020; Chinen et al, 2021). In addition to the centrosomal components, pole organization is controlled by spindle assembly factors such as nuclear mitotic apparatus protein (NuMA). NuMA is a large coiled-coil protein, and it localizes at the poles in proximity to the centrosomes (Merdes et al, 1996; Merdes and Cleveland, 1998; Merdes et al, 2000; Dionne et al, 1999; Silk et al, 2009; Radulescu and Cleveland, 2010; Hueschen et al, 2017; Kiyomitsu and Boerner, 2021). NuMA levels progressively increase at the poles during mitotic entry and decrease at mitotic exit (Kotak et al, 2013). The accumulation of NuMA at the poles is hypothesized to be regulated by the activity of the microtubule-dependent minus-end motor dynein/dynactin complex (Merdes et al, 1996; He et al, 2023). Without NuMA, spindle microtubules fail to focus at the poles, compromising pole integrity (Gaglio et al, 1995; Silk

[1]Department of Microbiology and Cell Biology (MCB), Indian Institute of Science (IISc), 560012 Bangalore, India. [2]Université Paris-Saclay, CEA, CNRS, Institute for Integrative Biology of the Cell (I2BC), 91198 Gif-sur-Yvette, France. [3]These authors contributed equally: Ashwathi Rajeevan, Vignesh Olakkal, Madhumitha Balakrishnan, Dwaipayan Chakrabarty. ✉E-mail: sachinkotak@iisc.ac.in

et al, 2009; Hueschen et al, 2019). The accumulation of NuMA at the poles is regulated by mitotic kinases: Cdk1-mediated phosphorylation promotes NuMA enrichment at the poles, while Aurora A-mediated phosphorylation maintains NuMA in a dynamic state (Kotak et al, 2013; Kiyomitsu and Cheeseman, 2013; Seldin et al, 2013; Gallini et al, 2016; Kotak et al, 2016; Keshri et al, 2020). Despite these insights, a comprehensive understanding of which motifs or domains present in NuMA are crucial for its accumulation at the poles is lacking. Furthermore, it is unclear whether Aurora A activity is continuously required during anaphase to preserve NuMA dynamics at the poles. Crucially, the potential consequences of a failure to dissolve pole structures near newly forming nuclei during mitotic exit have not been explored in any cellular system.

In recent years, there has been significant interest in exploring the material properties of poles/centrosomes (Woodruff, 2021). One idea is that the formation of these structures involves liquid-liquid phase separation (LLPS) or biomolecular condensation (Woodruff et al, 2017; Rale et al, 2018). In vitro, the key PCM component in *C. elegans* SPD-5 can self-assemble into spherical structures in the presence of a crowding agent, and these structures can recruit factors essential for microtubule growth and stability (Woodruff et al, 2017). Similarly, NuMA has recently been hypothesized to concentrate at the poles via LLPS (Sun et al, 2021; Ma et al, 2022). Notably, the C-terminus of NuMA is intrinsically disordered, and at a high concentration (40 μM) as a recombinant protein in the presence of a crowding agent, it assembles into micron-sized droplets that undergo fusion and accumulate microtubules (Sun et al, 2021). However, whether the sub-micromolar concentration of NuMA in a cell (<1 μM; Hein et al, 2015) allows it to accumulate at the poles simply via its ability to undergo LLPS remains unknown.

In this study, by utilizing chemical and genetic approaches in combination with fixed and live-cell imaging, we investigated the post-mitotic function of Aurora A kinase. We show that Aurora A activity during anaphase is essential to alter the material property of NuMA from a non-dynamic pathological state (referred to as a solid-state) to a dynamic state. In the absence of Aurora A activity, NuMA abnormally accumulates at the poles, leading to the bending of segregated mitotic chromosomes around these atypical NuMA-based poles. This disrupts the organization of the newly formed nuclei and their genome. The ability of NuMA to accumulate at the poles in vivo depends on the N-terminal dynein/dynactin-binding domain and its tendency to engage in homotypic multivalent interactions through its coiled-coil and C-terminal intrinsically disordered region (IDR). Furthermore, we show that arginine and aromatic residues in the IDR facilitate its accumulation at the poles by cation-π interactions, while glutamine residues promote solid-state phase transition upon Aurora A inhibition. Based on these results, we propose that the material property of poles must be tightly controlled to ensure proper organization of the developing nuclei at mitotic exit. This work highlights the importance of precise spatiotemporal coordination between the membrane (nucleus) and the membraneless organelle (poles) for cellular well-being.

# Results

## Aurora A activity during anaphase is essential for proper nuclear organization

While analyzing Aurora A function in mitosis, we noted that a significant number (94%; $n = 34$) of HeLa cells that rapidly exit

mitosis (within 30 min) in the presence of a specific Aurora A inhibitor MLN8237 harbored abnormal organization of the newly formed nuclei without showing any sign of chromosome instability and chromosome segregation errors (Fig. 1A–D; Movies EV1 and EV2). This data indicates that Aurora A activity during anaphase before nuclear envelope reformation (NER) is crucial for proper nuclear organization. Supporting this notion, we detected significant levels of active auto-phosphorylated Aurora A [phosphorylated at threonine 288 (T288$^P$); Walter et al, 2000; Littlepage et al, 2002] at the poles during anaphase (Fig. EV1A,B).

To explore Aurora A function during anaphase before NER, we sought to establish a genetic tool that would effectively remove Aurora A during this time. Building on previous studies of sea urchin CyclinB1, where the N-terminus was shown to be sufficient for proteasome-mediated degradation during anaphase (Glotzer et al, 1991), we identified the first 86 amino acids of human CyclinB1 (referred to as CycB) as containing the probable degron sequence. This was validated by fusing this region to AcGFP (*Aequora coerulescens* GFP; referred to as CycB-AcGFP), which led to the rapid degradation of CycB-AcGFP during anaphase (Figs. 1E,G and EV1C,D). Next, we fused Aurora A (67–403 amino acids) to CycB and AcGFP (referred to as CycB-Aurora A$^r$-AcGFP; Fig. 1E) and generated transgenic cell lines. The N-terminal sequence (1–66 amino acids) of Aurora A was intentionally removed because (i) it contains a recognition site for the APC/C$^{Cdh1}$ (Littlepage and Ruderman, 2002), and we intended to drive Aurora A degradation solely by CycB E3 ubiquitin ligase APC/C$^{Cdc20}$, (ii) a monoclonal antibody against the N-terminus of Aurora A can be used to detect endogenous Aurora A, and (iii) an siRNA against the N-terminus of Aurora A can deplete endogenous protein. We chose an engineered cell line that expresses catalytically active CycB-Aurora A$^r$-AcGFP in amounts indistinguishable from the endogenous Aurora A (Fig. 1F) and localizes similarly to endogenous Aurora A (Fig. EV1E). CycB-Aurora A$^r$-AcGFP significantly rescued the mitotic index and chromosome instability errors seen upon endogenous protein depletion (Fig. EV1F,G). As expected, CycB-Aurora A$^r$-AcGFP is swiftly degraded during the metaphase-to-anaphase transition (Fig. 1G,H). Cells expressing CycB-Aurora A$^r$-AcGFP failed to accurately separate their chromosomes (Fig. EV1H), a phenotype linked to Aurora A function in anaphase (Reboutier et al, 2013). These experiments established that CycB-Aurora A$^r$-AcGFP expressing cells could be utilized to study Aurora A-dependent function/s at mitotic exit. Using this cell line, we determined the significance of Aurora A for the proper nuclear morphology at the mitotic exit. Intriguingly, we found that CycB-Aurora A$^r$-AcGFP expressing cells, but not cells expressing Aurora A$^r$-AcGFP, lacking the CycB degron sequence, showed misshapen nuclei at the mitotic exit upon endogenous Aurora A depletion (Figs. 1I–K and EV1I–K). These findings reinforce the idea that Aurora A activity during mitotic exit is essential for cells to achieve correct nuclear morphology. Since this phenotype was independent of the previously defined role of Aurora A in spindle assembly (Berdnik and Knoblich, 2002; Hannak et al, 2001; Tsai and Zheng, 2005; Haren et al, 2009; Joukov et al, 2014; Kapoor and Kotak, 2019), we decided to study this new role of Aurora A in depth.

## Aurora A dissolves pole-localized NuMA in anaphase to ensure the proper nuclear organization

Notably, MLN8237-treated or CycB-Aurora A$^r$-AcGFP expressing cells showed a peculiar bending of segregated chromosomal mass at

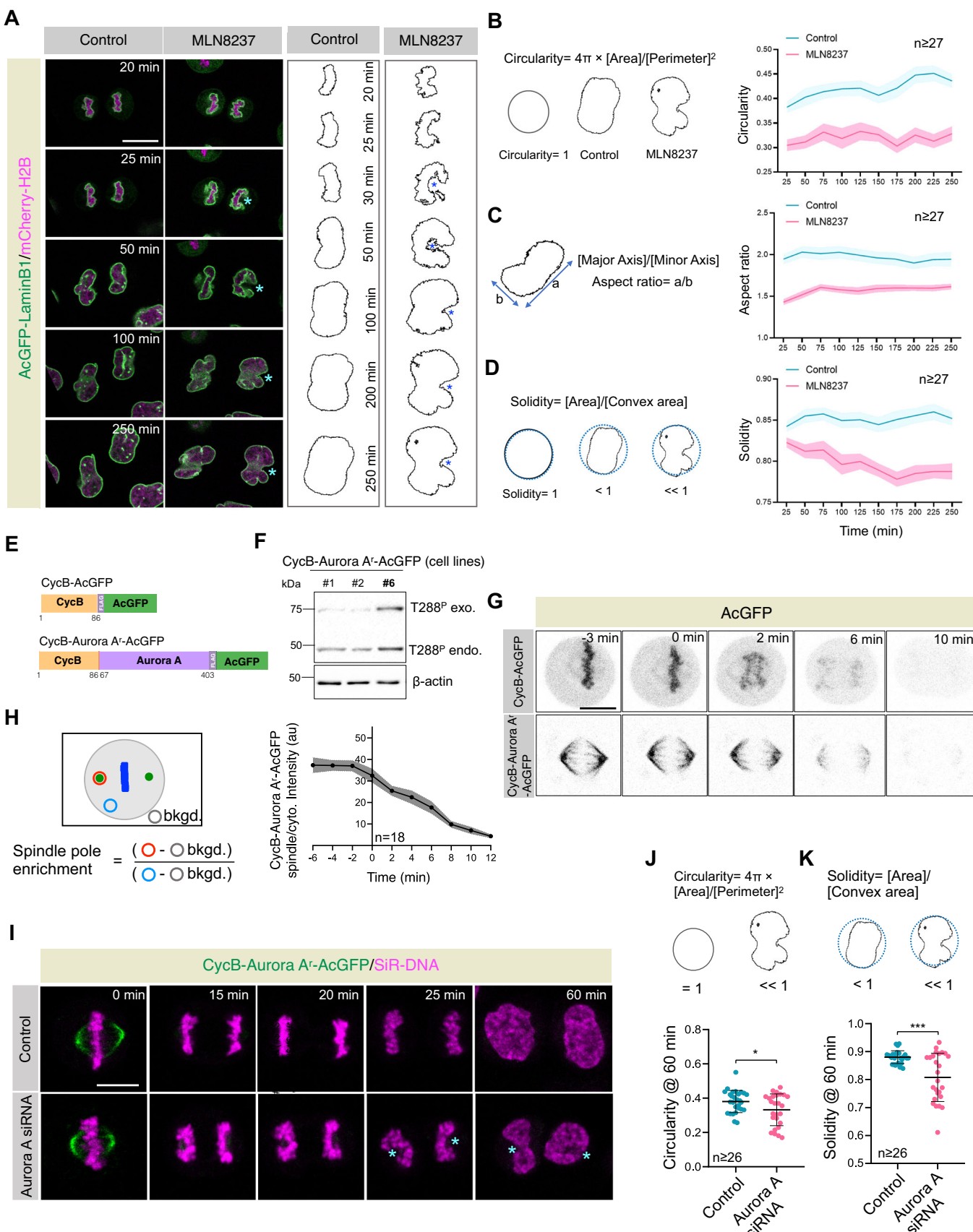

**Figure 1. Aurora A activity is vital for proper nuclear morphology at the mitotic exit.**

(A) Confocal live-cell imaging of HeLa cells stably coexpressing AcGFP-LaminB1 (green) and mCherry-H2B (magenta) that are treated with either DMSO (control) or Aurora A inhibitor MLN8237. Here and in subsequent Fig. panels, timepoint $t = 0$ min was set to the metaphase-to-anaphase transition (also see Movies EV1 and EV2). Asterisks in MLN8237-treated cells depict the bending of chromosome ensemble mass around the position where poles are usually located. The shape of the chromosome ensemble is shown on the right in these conditions. In all experiments where inhibitors are dissolved in solvent DMSO, DMSO was utilized as a control. (B–D) Morphological analysis of various nuclear parameters [circularity (B), aspect ratio (C), and solidity (D)] in control and MLN8237-treated cells at different time points post metaphase-to-anaphase transition. Curves and shaded areas indicate mean ± SEM. The $p$ values for circularity and aspect ratio for all the time points is <0.001, however, the $p$ values for solidity at 25 min post metaphase to anaphase transition is 0.028, and for the rest of the time points is <0.001. (E) Schematic representation of CycB and siRNA resistant form of Aurora A construct with AcGFP- and mono-FLAG-tag at the C-terminus (CycB-AcGFP and CycB-Aurora A$^r$-AcGFP, respectively). (F) Immunoblot analysis of protein extracts prepared from mitotically synchronized HeLa cells stably expressing three independent clones (#1, #2, and #6) of CycB-Aurora A$^r$-AcGFP. Extracts were probed with antibodies directed against autocatalytically active Aurora A (T288$^P$) and β-actin. Endogenous and exogenous T288$^P$ bands are indicated. Line #6 is utilized for all future analyses. The molecular mass is indicated in kilodaltons (kDa) on the left. (G) Confocal live-cell imaging of HeLa cells stably expressing CycB-AcGFP or CycB-AuroraA$^r$-AcGFP. (H) Schematic of the method to quantify spindle pole intensity of CycB-Aurora A$^r$-AcGFP [in arbitrary unit (au)] and the outcome of such analysis over time. Curve and shaded areas indicate mean ± SEM. Here and in subsequent Fig. panels, bkgd. represents background intensity. (I) Confocal live-cell imaging of HeLa cells stably expressing CycB-AuroraA$^r$-AcGFP (green) and probed for silicon-rhodamine DNA (SiR-DNA; magenta) to visualize chromosomes ensemble and nuclear shape post-mitosis in control and upon transfection with Aurora A siRNA for 60 h. Also, see Fig. EV1I–K for nuclear shape analysis of cells expressing AuroraA$^r$-AcGFP in control and upon Aurora A depletion. (J, K) Nuclear shape analysis [circularity (J) and solidity (K)] from the confocal live-cell imaging of cells, as mentioned in panel (I). Asterisks in Aurora A siRNA-transfected cells depict the bending of the chromosome ensemble mass around the position where poles are usually located. The quantifications on the right represent mean ± SD. Exact $p$ values are *$p = 0.0245$ (J) and ***$p < 0.001$ (K). $p$ values are denoted as follows: *$p < 0.05$; ***$p < 0.001$ as determined by two-tailed unpaired Student's $t$-test. Scale bars in (A, G, I) represent 10 µm. Source data are available online for this figure.

the position where typically the poles are present (Fig. 1A,I; indicated by asterisks). Earlier, we and others reported that Aurora A inactivation during metaphase causes spindle orientation defects by reducing the cortical levels of NuMA and concomitantly increasing its levels at the poles (Gallini et al, 2016; Kotak et al, 2016). Therefore, to characterize pole-localized NuMA and the segregated mass of chromosomes ensemble in anaphase, we acutely inhibited Aurora A by treating HeLa cells with MLN8237 (50 nM) for 2 h. This regimen significantly inhibits active Aurora A (T288$^P$) accumulation at the centrosome without substantially increasing the number of monopolar spindle formation and chromosome instability errors, which are often seen in cells treated with MLN8237 for a longer duration (Fig. EV2A–D; Asteriti et al, 2014). Using these conditions, we found that NuMA, which swiftly dissolves at poles in anaphase, is significantly enriched at the poles in Aurora A-inhibited cells (Fig. 2A,B). Moreover, we observed that the segregated chromosome sets had bent around the abnormally localized NuMA at poles during the telophase and early G1 phase of the cell cycle upon Aurora A inhibition (Fig. 2A–C; Movies EV3 and EV4). Similar observations were made in other cell lines, including U2OS, HEK293, and hTERT-RPE1 (Fig. EV2E–H). Geometric measurements showed that NuMA geometry at the poles significantly differs in Aurora A-inhibited cells (Fig. EV2I,J). The significant accumulation of NuMA during anaphase was not because of excess NuMA in MLN8237-treated cells (Fig. EV2K); instead, enriched levels at the poles during anaphase are most likely due to overall decreased levels of NuMA in the cytoplasm (Fig. EV2L).

To directly visualize the impact of NuMA accumulation at the poles and its effect on nuclear morphology, we simultaneously visualized NuMA and the NE by live-cell imaging in HeLa cells stably coexpressing mCherry-NuMA and AcGFP-LaminB1 in control and MLN8237-treated cells, respectively. In control cells, AcGFP-LaminB1 localized around the segregated mitotic chromosome ensemble at ~15 min post-anaphase onset, and at this time, NuMA levels at the poles were significantly reduced (Fig. 2D,E). In contrast, in cells that were acutely treated with MLN8237, NuMA at the poles was readily visible for a considerably longer duration of

~1 h or more, and the newly formed nuclei were bent around the NuMA-localized poles (Fig. 2D–F). Similar observations were made in cells coexpressing mCherry-NuMA with nuclear envelope marker GFP-Nup107 (Fig. EV2M,N).

NuMA is localized to the nucleus post-mitosis after NER (Rajeevan et al, 2020; Serra-Marques et al, 2020). Notably, cells that are acutely treated with MLN8237 show strong enrichment of NuMA at the poles, and in these cells, NuMA is weakly accumulated in the developing nuclei (Appendix Fig. S1A,B). Therefore, the nuclear shape defects upon Aurora A inhibition could also be due to reduced levels of NuMA in the nucleus. To test this directly, we followed nuclear shape in cells expressing siRNA-resistant AcGFP-tagged NuMA that lacks the nuclear localization signal (AcGFP-NuMA$^r_{\Delta NLS}$) upon endogenous protein depletion (Appendix Fig. S1C). Importantly, AcGFP-NuMA$^r_{\Delta NLS}$ expressing cells did not reveal any nuclear shape defects post-mitotically (Appendix Fig. S1D,E). This observation argues against the possibility that the accumulation of misshapen nuclei in Aurora A-inhibited cells is caused by weak NuMA levels in the newly formed nuclei.

Next, to specifically link NuMA pole accumulation with Aurora A activity during anaphase, we generated stable cells coexpressing CycB-Aurora A$^r$-AcGFP and mCherry-NuMA. Consistent with the above results using Aurora A inhibitor, we found that NuMA levels were significantly enriched at the pole in cells expressing CycB-Aurora A$^r$-AcGFP upon endogenous Aurora A depletion (Fig. 2G,H). Aurora A phosphorylates NuMA at Serine 1969 in its C-terminus (Kettenbach et al, 2011). The C-terminus of NuMA with a phospho-dead alanine replacement at S1969 (S1969A) abnormally accumulates at the poles during metaphase, similar to cells treated with MLN8237 (Gallini et al, 2016). Therefore, we tested if the expression of a full-length AcGFP-tagged NuMA containing the S1969A replacement (AcGFP-NuMA$^r_{S1969A}$) in cells depleted for endogenous NuMA would lead to its accumulation at the poles during anaphase. And if so, could AcGFP-NuMA$^r_{S1969A}$ accumulation at the poles bend the segregated chromosome ensemble and the newly formed nuclei? As envisaged, AcGFP-NuMA$^r_{S1969A}$ robustly accumulated at the poles in contrast to the wild-type AcGFP-NuMA$^r$ (Appendix Fig. S1F,G). Also, the

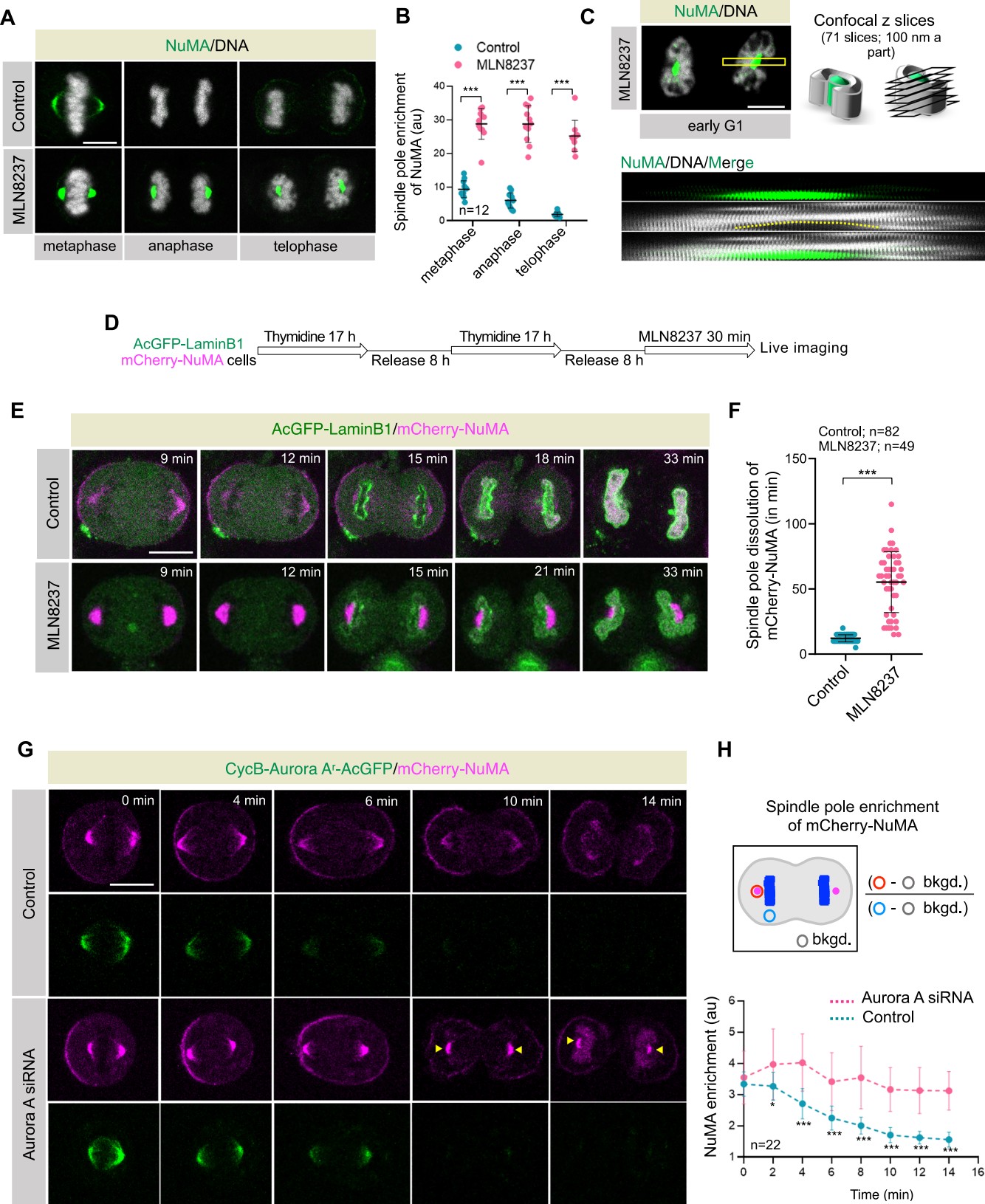

**Figure 2.  Aurora A activity dissolves NuMA at the poles at the mitotic exit.**

(A, B) Representative images from immunofluorescence (IF) analysis of HeLa cells during metaphase, anaphase, and telophase stages in control and upon acute Aurora A inhibition with MLN8237 (50 nM) for 2 h. Cells were stained for NuMA (green), and DNA is shown in gray (A). Spindle pole intensity of NuMA (in au) was calculated at these different stages as depicted in Fig. 1H (B). The quantification on the right represents mean ± SD. Exact $p$ values from left to right are ***$p < 0.001$, ***$p < 0.001$, ***$p < 0.001$. (C) Representative images from IF analysis of HeLa cells in the early G1 phase of the cell cycle while treated with MLN8237 (also see Movies EV3 and EV4). Cells were stained for NuMA (green), and DNA is shown in gray. The midplane section of the cell is shown. The schematic on the right represents an area (marked by yellow in the image) covered from top to bottom and sliced into 71 z-projections with a step size of 100 nm to build the kymograph at the bottom. Note the displacement of the DNA (and thus the nucleus, shown by the dashed yellow line) around the abnormally localized NuMA pole. (D) Synchronization method for acute Aurora A inhibition during mitotic exit. (E) Confocal live-cell imaging of mitotically synchronized HeLa cells stably coexpressing AcGFP-LaminB1 (green) and mCherry-NuMA (magenta) in the absence (control) or presence of MLN8237. (F) Quantification of the dissolution time (in min) of mCherry-NuMA at the poles in control and MLN8237-treated cells with respect to (w.r.t.) metaphase-to-anaphase transition, bars indicate mean ± SD. Exact $p$ value is ***$p < 0.001$. (G) Confocal live-cell imaging of HeLa cells stably coexpressing CycB-AuroraAʳ-AcGFP (green) and mCherry-NuMA (magenta) during anaphase in control and upon transfection with Aurora A siRNA. Recording was started 60 h post-transfection for control siRNA and Aurora A siRNA. (H) Schematic of the method to quantify spindle pole enrichment of mCherry-NuMA intensity (in au) during anaphase in cells stably coexpressing CycB-AuroraAʳ-AcGFP and mCherry-NuMA in control and after transfection with Aurora A siRNA. Error bars indicate mean ± SD. Exact $p$ values from left to right are $p = 0.3858$ (ns), *$p = 0.0163$, ***$p < 0.001$, ***$p < 0.001$, ***$p < 0.001$, ***$p < 0.001$, ***$p < 0.001$, ***$p < 0.001$. $p$ values are denoted as follows: *$p < 0.05$; ***$p < 0.001$ as determined by two-tailed unpaired Student's $t$-test. Scale bars in (A, C, E, G) represent 10 μm. Source data are available online for this figure.

pole-localized AcGFP-NuMAʳ$_{S1969A}$ had bent the segregated chromosome ensemble and the nascent nuclei at the mitotic exit (Appendix Fig. S1F). Based on these results, we conclude that Aurora A activity in anaphase is important to release NuMA from the spindle poles. Without Aurora A activity, NuMA strongly accumulates at the poles behind the segregating chromosome sets, thus creating a roadblock for cells to attain proper nuclear morphology during NE-reformation. Additionally, we show that Aurora A-mediated phosphorylation at S1969 of NuMA dynamically regulates pole-localized NuMA in metaphase and during anaphase.

## Aurora A inhibition does not enrich other PCM-localized proteins

Because NuMA localizes in proximity to the PCM (Compton et al, 1992; Compton and Luo, 1995; Radulescu and Cleveland, 2010), we wondered whether abnormally increased levels of NuMA at the poles upon Aurora A inactivation is accompanied by an augmentation in PCM-localized proteins. To address this, we analyzed the localization of Cdk5Rap2, PCNT, Cep192, γ-tubulin, and Cep152 during anaphase in untreated control and MLN8237-treated cells. We found no noticeable enrichment for these centrosomal localized proteins in MLN8237-treated cells (Appendix Fig. S2). In contrast, PCNT, γ-tubulin, and Cep152 levels decreased at the centrosome in MLN8237-treated cells (Appendix Fig. S2). Interestingly, microtubules that directly associate with NuMA (Du et al, 2002; Haren and Merdes, 2002) also remain unaltered at the poles in Aurora A-inhibited anaphase cells (Appendix Fig. S2). These data suggest that NuMA enrichment at the poles, mediated by Aurora A inactivation, does not cause enrichment of centrosomal proteins, at least the ones we tested.

## Aurora A controls the material state of NuMA at the poles

Transiently expressed GFP-NuMA shows significantly slower recovery at the poles in fluorescence recovery after photobleaching (FRAP) experiments upon Aurora A inhibition during metaphase (Gallini et al, 2016). We obtained similar results in the HeLa cells stably expressing AcGFP-tagged NuMA at levels indistinguishable from the endogenous proteins during metaphase using two

independent Aurora A-specific inhibitors-MLN8237 and MK5108 (Fig. EV3A–D; Sana et al, 2022; Shimomura et al, 2010). Similarly, FRAP analysis of AcGFP-NuMA at the poles during anaphase in cells acutely treated with MLN8237 revealed that Aurora A inhibition significantly impairs AcGFP-NuMA fluorescence recovery (Fig. 3A,B). The half-time for the recovery ($T_{1/2}$) of the AcGFP signal after photobleaching of untreated cells was ~20 s, which was significantly delayed to ~70 s upon Aurora A inhibition (Fig. 3C). Concomitantly, there was a substantial reduction in the mobile fraction from ~70% to about ~30% (Fig. 3D). These observations indicate that the Aurora A activity is continuously required throughout mitosis to keep NuMA in a dynamic state at the poles.

To confirm this observation by an independent means, we created a HeLa cell line stably expressing NuMA with N-terminally tagged photoconvertible fluorescent monomeric protein mEOS3.2 (referred to as mEOS-NuMA; Fig. 3E; Zhang et al, 2012). Using these cells, we photoconverted mEOS-NuMA at one pole from green (depicted as gray) to red using a pulse of 405 nm and measured the appearance of photoconverted red fluorescence signal onto the other non-irradiated pole (Fig. 3E). As expected from the FRAP analysis; the photoconverted red signal is swiftly localized to the non-photoconverted gray pole in control cells, indicating the dynamic nature of pole-localized NuMA (Fig. 3F). However, cells that were acutely treated with MLN8237 failed to exchange the red fluorescence signal to the other non-photoconverted poles (Fig. 3F). Similarly, cells expressing the mEOS-tagged NuMA that cannot be phosphorylated by Aurora A at S1969 failed to exchange fluorescence signals from one pole to another (Fig. EV3E).

Next, we sought to investigate the organization of endogenous NuMA protein at the poles in the presence of and upon acute Aurora A inactivation. To this end, we utilized a three-dimensional (3D) lattice-structured illumination super-resolution microscope (3D-SIM²), which has increased resolution compared to conventional confocal microscopy. This analysis revealed that NuMA organizes into an extended network at the poles, which we refer to as "meshwork" (Fig. 3G). These structures are present in the proximity of the PCM matrix (monitored by PCNT) during metaphase and anaphase (Fig. 3G). Importantly, NuMA transformed into highly compact structures in cells treated with MLN8237 (Fig. 3G; Appendix Fig. S3). These experiments suggest that Aurora A activity during mitosis keeps NuMA pools at the poles in a dynamic state in the form of a

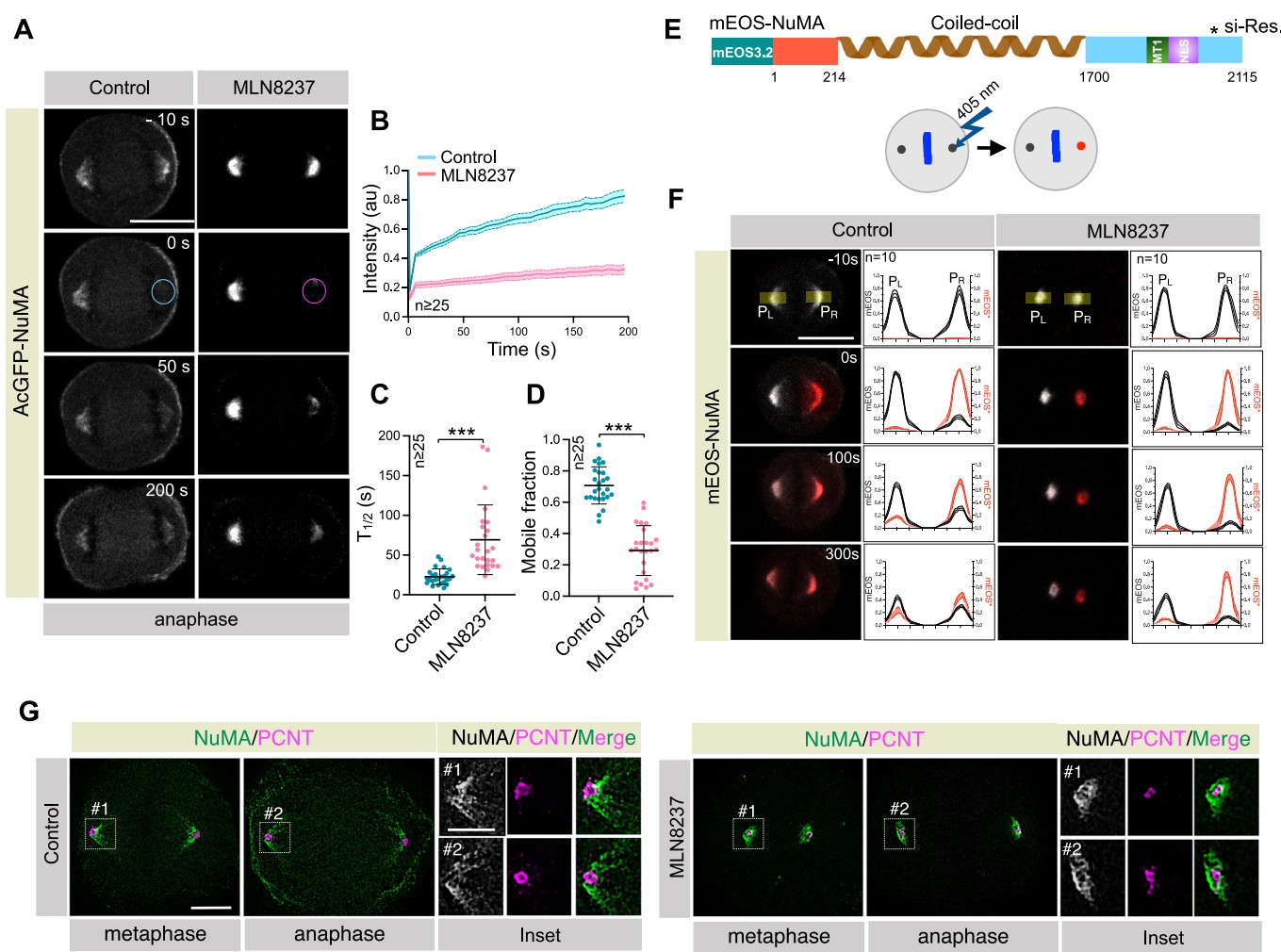

**Figure 3. Aurora A regulates the material property of NuMA at the poles.**

(A, B) FRAP analysis of HeLa cells stably expressing AcGFP-NuMA (gray) in the absence or presence of MLN8237 during anaphase (A). Time is indicated in seconds (s). The bleached regions of the control and MLN8237-treated cells are shown by blue and magenta circles, respectively. The AcGFP recovery profile of the bleached area for control and MLN8237-treated cells (B) is shown on the right. Curves and shaded areas indicate mean ± SEM. Note the remarkably slow recovery of pole-localized AcGFP-NuMA in MLN8237-treated anaphase cells. Also, note that spindle pole levels of AcGFP-NuMA at the unbleached pole in control cells are reducing as anaphase progresses. (C, D) The half-time of recovery [$T_{1/2}$] (C) and the mobile fraction (D) of untreated and MLN8237-treated anaphase cells. Error bars: mean ± SD. Exact *p* values are ***$p < 0.001$ (C) and ***$p < 0.001$ (D). (E, F) Domain organization of mEOS3.2-tagged NuMA (mEOS-NuMA). In this and other Fig. panels Microtubule binding domain 1 and nuclear localization signals are shown as MT1 and NLS, respectively. As shown in the schematics, HeLa cells expressing mEOS-NuMA were photoconverted from green (shown as gray) to red at one pole by the exposure of 405 nm laser (E), followed by assessing the accumulation of photoconverted mEOS-NuMA on the non-photoconverted pole by confocal live-cell imaging (F). A Line-scan with a line thickness of 3 μm (roughly similar to the size of the poles) was performed for the unperturbed left pole ($P_L$) or perturbed right pole ($P_R$) for untreated and MLN8237-treated cells. Ten cells were analyzed (see Methods). Scale bar 10 μm. Error bars: mean ± SEM. (G) Representative images from the super-resolution 3D-SIM² analysis of HeLa cells immunostained with anti-NuMA (green) and anti-PCNT (magenta) during metaphase and anaphase in the absence and upon acute treatment with MLN8237. Insets on the right show the pole-localized NuMA (gray) and PCNT (magenta). $n \geq 10$ cells were analyzed for control and MLN8237 in metaphase and anaphase, and the representative image is shown above. See Appendix Fig. S3 for an independent example of a metaphase and anaphase cell that is treated with MLN8237. *p* value is denoted as follows: ***$p < 0.001$ as determined by two-tailed unpaired Student's *t*-test. Scale bars in (A, F) represent 10 μm. The scale bar in (G) represents 5 μm for the cell and 2 μm for the insets. Source data are available online for this figure.

meshwork. In the absence of that, NuMA undergoes a material-state transition into a non-dynamic compact state, which, from now onwards, we refer to as "solid."

## The Dynein binding and self-oligomerization capability of NuMA is critical for its pole accumulation

The pole localization of NuMA during metaphase and its altered dynamic state upon Aurora A inhibition has recently been linked to

its ability to undergo liquid-liquid phase separation (LLPS) (Sun et al, 2021; Ma et al, 2022). However, these conclusions relied on: (1) in vitro assays with recombinant C-terminal NuMA at high concentrations in the presence of polyethylene glycol (PEG) as a crowding agent; (2) cellular experiments overexpressing GFP-tagged NuMA fragments alongside endogenous protein; and (3) the use of 1,6-hexanediol, which not only disrupts weak hydrophobic interactions but can also impair kinases and phosphatases function (Duster et al, 2021). Such approaches may lead to misinterpretation

of LLPS phenomena, as highlighted previously (McSwiggen et al, 2019; Alberti et al, 2019; Hedtfeld et al, 2024). Therefore, we investigated the mechanism of NuMA pole accumulation in vivo by examining cells expressing various deletion/point-mutation constructs of NuMA following endogenous NuMA depletion. We further sought to investigate how Aurora A maintains its correct (dynamic) material state at the poles, providing a more physiologically relevant understanding of its regulation during mitosis.

Sun et al, found that NuMA tagged with mClover assembled into multiple "condensates-like" bodies that occasionally fuse upon nocodazole treatment (Sun et al, 2021). This observation allowed authors to conclude that NuMA assembles into liquid-like condensates in vivo without microtubules and associated dynein/dynactin. In contrast, we found that the concentration of nocodazole (100 ng/ml = ~332 nM) utilized by the authors is insufficient to completely depolymerize microtubules (Fig. 4A,B). When microtubules are fully depolymerized using a much higher concentration of nocodazole (>1.5 μM), NuMA does not assemble into condensates (Fig. 4A,B). These results indicate that NuMA's ability to accumulate at the poles cannot simply be explained by its potential to undergo LLPS without microtubule and possibly dynein/dynactin motor interaction.

The accumulation of NuMA at the poles is hypothesized to be regulated by active dynein/dynactin-mediated transport of NuMA (Merdes et al, 1996; He et al, 2023). Therefore, to test if NuMA requires the dynein/dynactin motor for its robust accumulation at the poles, we tested the localization of AcGFP-tagged NuMA lacking the N-terminal (1–705 amino acids) dynein/dynactin interaction module (Fig. 4C; Kotak et al, 2012) (AcGFP-NuMA$^r_{\Delta DBD}$). AcGFP-NuMA$^r_{\Delta DBD}$ failed to accumulate significantly at poles in cells depleted for endogenous NuMA (Fig. 4C,D; compare ii vs. i and EV4A,B; also see Fig. EV4C–E for the efficacy of NuMA depletion, Fig. EV4F for chromosome instability errors associated with this, and other constructs mentioned below upon endogenous protein depletion, and Fig. EV4G for immunoblot analysis). Since recombinant full-length NuMA can assemble into higher-order multimeric assemblies through its coiled-coil and C-terminal domains in vitro (Harborth et al, 1999), we reasoned that synergy between the coiled-coil region and the C-terminus might promote NuMA accumulation at the poles. We therefore investigated the localization of AcGFP-tagged mutant NuMA lacking most of its coiled-coil domain, except 213-705 amino acid residues, which are required for dynein/dynactin interaction (Kotak et al, 2012; Okumura et al, 2018; Renna et al, 2020) (AcGFP-NuMA$^r_{NC}$; Figs. 4C and EV4A–G). Notably, AcGFP-NuMA$^r_{NC}$ enrichment was significantly reduced at the poles (Fig. 4C,D; compare iii vs. i, and EV4A,B). Next, we examined the relevance of the C-terminus of NuMA (1700–2115 amino acids) for its accumulation at the poles. The C-terminus of NuMA is predicted to be largely disordered (Fig. 5A: identified using https://mobidb.bio.unipd.it/; AlphaFold Protein Structure Database; Necci et al, 2021; Jumper et al, 2021) and because of this, we hypothesized that it might engage in weak homotypic multivalent interactions (see the next section). As expected, the expression of AcGFP-NuMA$^r_{(1–1699)}$ lacking the C-terminal intrinsically disordered region (IDR) failed to accumulate at the poles (Fig. 4C,D; compare iv vs. i, and EV4A–G). Similarly, cells expressing AcGFP-NuMA$^r_{(1700–2115)}$, containing only C-terminus IDR, had a significantly reduced AcGFP signal at the poles (Fig. 4C,D; compare v vs. i, and EV4A–G). These experiments indicate that the synergy

between N-terminal dynein/dynactin binding, a large coiled-coil domain, and the C-terminus IDR is critical for robust NuMA accumulation at the poles.

## Arginine and aromatic residues present in NuMA IDR govern multivalent homotypic interactions for its pole accumulation

Because IDR sequences in proteins promote weak interactions (Shin and Brangwynne, 2017; Alberti and Hyman, 2021; Holehouse and Kragelund, 2024), we sought to investigate the potential of NuMA C-terminal IDR (Fig. 5A) in engaging multivalent homotypic interactions for its pole accumulation. To this end, we employed the Corelet system (Fig. 5B; Bracha et al, 2018). This tool utilizes multivalent 24-mer Ferritin as a "core particle," which acts as a platform to assemble light-induced condensates in the nucleus using IDR-dependent multivalent interactions. Because the last 58 amino acid residues of NuMA are necessary for its interaction with chromatin in the nucleus (Rajeevan et al, 2020), and its engagement with chromatin may prevent the light-induced homotypic interaction potential of NuMA$_{(1700–2115)}$ in Corelet experiments, we deleted the last 58 amino acids from the NuMA C-terminus IDR sequence (referred to as NuMA$^s_{C-ter}$). Notably, NuMA$^s_{C-ter}$ containing 1700–2057 residues formed light-induced condensates analogous to the IDR of HNRNPA1c (Fig. 5C,D; Bracha et al, 2018). Likewise, NuMA full-length, lacking the last 58 amino acid residues [NuMA$_{(1–2057)}$], could assemble into condensates (Fig. EV5A,B). This data indicates that the NuMA$^s_{C-ter}$ IDR and NuMA$_{(1–2057)}$ can undergo multivalent protein-protein interaction to assemble into condensates, at least when expressed in the nucleus. Next, we sought to investigate the nature of amino acids in IDR facilitating multivalent interactions. The IDR sequence contains 39 arginine (R) and 6 tyrosine (Y), 7 phenylalanine (F), and 1 tryptophan (W) [collectively referred to as Aromatic (Aro)] residues, which are mostly conserved (Figs. 5A and EV5C). These amino acids might allow protons in the guanidino moiety on the arginine side chains to form cation-π interactions with the electrons in the benzene ring of aromatic residues. To test the role of these residues in engaging in multivalent interactions, we designed two constructs; in one, we mutated all arginine residues in NuMA$^s_{C-ter}$, except those in the NLS, to glycine. In the other construct, we mutated all the aromatic residues in NuMA$^s_{C-ter}$ to alanine. Strikingly, the light-dependent condensate formation of arginine-mutated NuMA [NuMA$^s_{C-ter(R>G)}$] via the Corelet system is entirely abrogated (Fig. 5C,D). Similarly, we found that mutation of aromatic residues significantly affected the IDR-mediated condensate formation (Fig. 5C,D).

Next, we sought to examine the relevance of these residues in the context of the full-length NuMA protein for its pole localization during mitosis. Notably, unlike the control AcGFP-NuMA$^r$, AcGFP-NuMA$^r_{(R>G)}$ and AcGFP-NuMA$^r_{(Aro>A)}$ failed to localize at the poles (Fig. 5E–G; compare iii & iv vs. i; also see Fig. EV4F for chromosome instability errors associated with mutant proteins upon endogenous NuMA depletion, and EV5D for immunoblot analysis). We conclude that collective interactions between the arginine and aromatic residues in NuMA IDR promote NuMA pole accumulation during mitosis. Our data further suggest that multiple NuMA IDR sequences, once brought closer with the help of dynein/dynactin motor and the large coiled-coil domain, engage

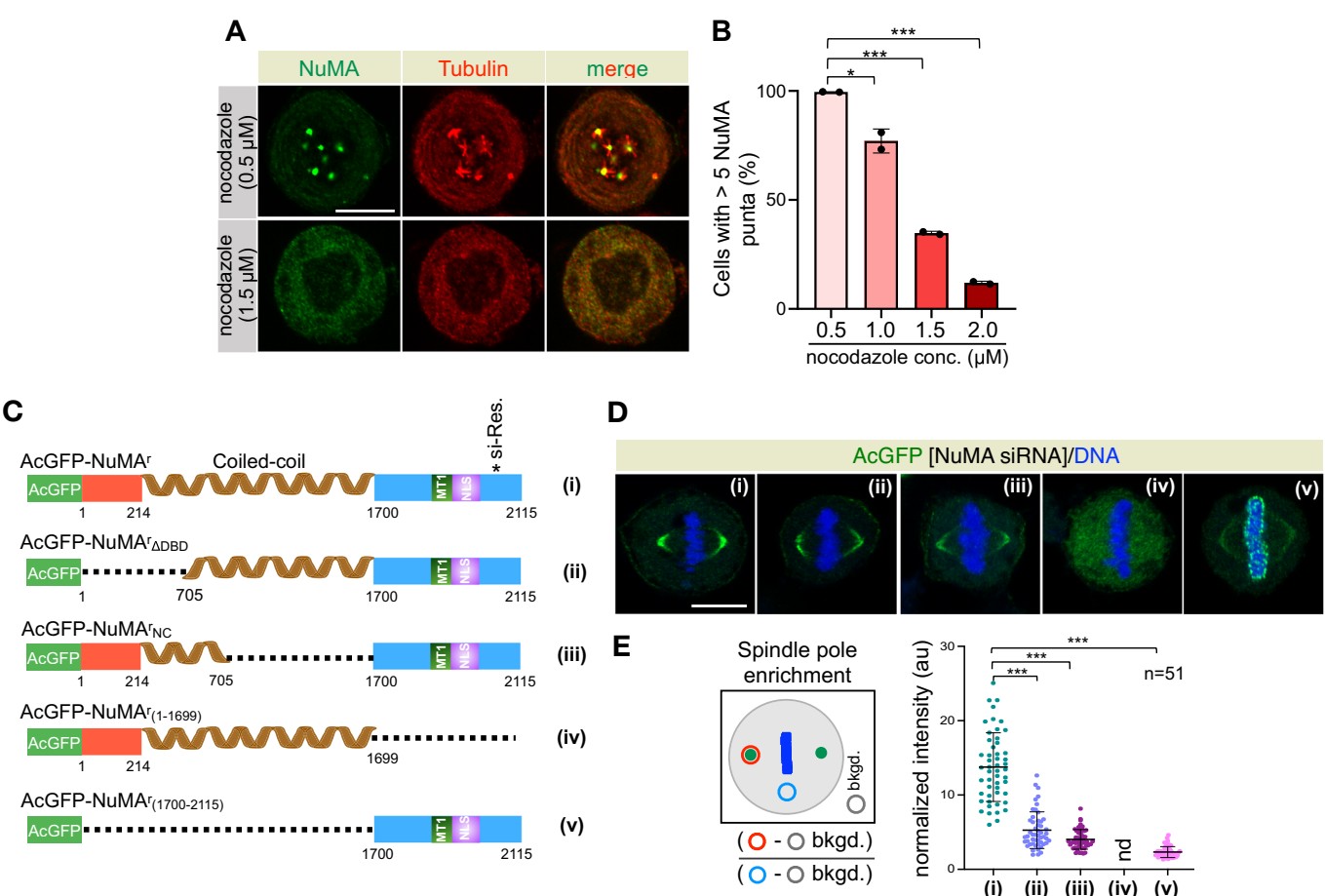

**Figure 4. Dynein/dynactin binding capacity of NuMA synergizes with its ability to make multimeric interactions for its pole accumulation.**

(A, B) IF analysis of HeLa cells after treatment with various concentrations of microtubule poison nocodazole for 17 h. These cells were stained with anti-NuMA (green) and anti-β-tubulin (red) antibodies. Quantification in B depicts the presence of more than 5 NuMA-based condensates at various nocodazole concentrations. Error bars: mean ± SD from two independent experiments ($n > 500$ cells each). Exact $p$ values from left to right are *$p = 0.0145$, ***$p < 0.001$, ***$p < 0.001$. (C) Domain organization of AcGFP-tagged and siRNA-resistant wild-type full-length NuMA (i) and various deletion constructs (ii–v). (D) IF analysis of HeLa cells expressing all NuMA constructs mentioned in (C) at 72 h after transfections with NuMA siRNA. These cells were stained with anti-GFP (green) antibodies. The DNA is shown in blue. Please note that AcGFP-NuMA$^r_{(1700–2115)}$ localizes on the chromosomes during metaphase, as shown previously (Rajeevan et al, 2020; Serra-Marques et al, 2020). The depletion efficiency of endogenous NuMA is shown in Fig. EV4C–E, and the % of chromosome instability associated with the expression of these constructs upon endogenous protein depletion is shown in Fig. EV4F. (E) Schematic of the method to quantify spindle pole enrichment of various NuMA constructs (i–v). Note that AcGFP-NuMA$^r_{(1-1699)}$ does not localize to the poles, and therefore its pole intensity was not determined (nd). Error bars: mean ± SD. Also, see EV4A, B for AcGFP fluorescence intensity at the poles divided by the microtubule intensity at the poles. Exact $p$ values from left to right are ***$p < 0.001$, ***$p < 0.001$, ***$p < 0.001$. nd represents not determined. $p$ values are denoted as follows: *$p < 0.05$; ***$p < 0.001$ as determined by two-tailed unpaired Student's $t$-test. Scale bars in (A, D) represent 10 μm. Source data are available online for this figure.

in multivalent IDR-mediated homotypic interactions to enrich NuMA at the poles (see Discussion and Fig. 9).

## Glutamine residues in NuMA IDR are crucial for the dynamic-to-solid transition

Aurora A phosphorylates NuMA at S1969 in its IDR (Kettenbach et al, 2011). Cells expressing mutant NuMA containing the S1969A replacement strongly accumulate at the poles (Appendix Fig. S1F,G), indicating that Aurora A-based phosphorylation at S1969 keeps NuMA in a dynamic state. We reasoned that in the absence of Aurora A-mediated phosphorylation, NuMA IDR undergoes conformational changes, forcing NuMA to change its material state from dynamic to solid. If this hypothesis is correct, which amino

acids are responsible for this phenotype upon Aurora A inactivation? Glutamine (Q) residues are implicated in promoting the hardening of proteins that have been associated with the onset of age-related neurodegenerative pathologies (Patel et al, 2015; Shin and Brangwynne, 2017; Wang et al, 2018; Zbinden et al, 2020). Therefore, we investigated the relevance of glutamine residues present in NuMA IDR in facilitating solid configuration upon Aurora A inhibition. Surprisingly, we found no difference in the condensate assemblies of mutant NuMA where glutamine residues are replaced with glycine in the Corelet system (Fig. 5C,D). However, the mutant NuMA IDR was relatively more dynamic and dissolved significantly faster than the wild-type IDR (Fig. 5C,D). To characterize the relevance of glutamine residues for pole localization in mitosis, we examined the localization of either AcGFP-

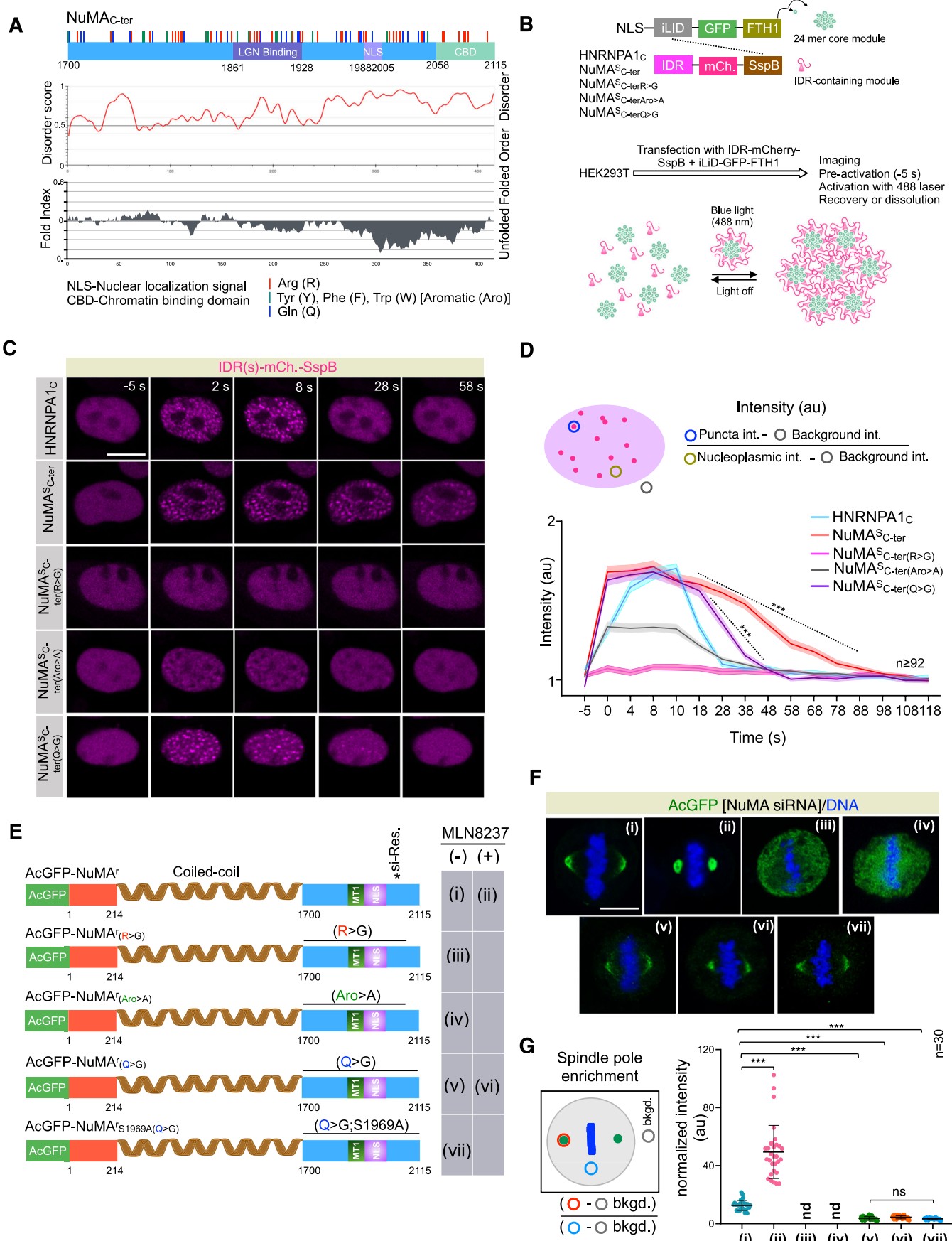

◄

**Figure 5.   Arginine and aromatic residues within NuMA's IDR mediate multivalent homotypic interactions, whereas glutamine residues promote material hardening upon Aurora A inhibition.**

(A) Distribution of arginine (R; red), tyrosine, phenylalanine, tryptophan (Aro; green), and glutamine (Q; blue) residues within NuMA$_{C-ter}$. The chromatin binding domain (CBD), LGN binding region, and nuclear localization domain (NLS) are marked. IUPred, intrinsic disorder prediction (Erdos and Dosztanyi, 2020); FOLD, folding prediction using the PLAAC website (http://plaac.wi.mit.edu). (B) Schematic representation of the Corelet system (Bracha et al, 2018). Corelet consists of two modules: 1) GFP-tagged ferritin core (24 mer), which contains photo-activatable iLID domains, and 2) self-interacting intrinsic disorder region (IDR), mCherry-tag, and light-sensitive iLID binding partner SspB (shown by the dashed line). Upon illumination with a 488 nm laser, up to 24 IDR modules are captured by each core, which assembles into condensates via multivalent IDR interactions in a reversible light-dependent manner. This tool is used to study light-dependent multivalent interactions of NuMA$_{C-ter}$ IDR lacking chromatin binding domain (CBD; referred to as NuMA$^S_{C-ter}$) and its mutant forms, as indicated. IDR of HNRNPA1c was used as a positive control. (C) Confocal live-cell imaging of Corelet-expressing HEK293 cells with various IDRs, as indicated. Representative images show robust condensate assembly when wild-type NuMA$^S_{C-ter}$ IDR was used in Corelet, but not when either arginine (R) or aromatic (Aro) residues in NuMA$^S_{C-ter}$ IDR were mutated. Also note that the condensates of NuMA$^S_{C-ter}$ IDR mutated for glutamine (Q) residues dissolve significantly rapidly than the wild-type NuMA$^S_{C-ter}$ IDR. (D) Schematic representation of the quantification method, intensity (in au), and the dynamics of Corelet-based condensates of HNRNPA1c, NuMA$^S_{C-ter}$, and NuMA$^S_{C-ter}$ mutant fragments. Curves and shaded areas indicate mean ± SEM. As indicated, the behavior of more than 90 clusters were analysed from a minimum of 20 cells in each condition. (E) Domain organization of AcGFP-tagged and siRNA-resistant wild-type full-length NuMA (i) and various mutant constructs, which are either left untreated (iii, iv, v, and vii), or treated with MLN8237 (ii and vi), as indicated. (F) IF analysis of HeLa cells expressing either wild-type or mutant NuMA constructs with or without treatment with MLN8237 (+/−), as specified in panel (E), 72 h after transfections with NuMA siRNA. These cells were stained with anti-GFP (green) antibodies. The DNA is shown in blue. Also, see Fig. EV4F for the % of chromosome instability associated with the expression of these constructs upon endogenous protein depletion. (G) Schematic of the method to quantify spindle pole intensity of various NuMA constructs with or without treatment with MLN8237 (+/−), as specified in (E). Note that AcGFP-NuMA$^r_{(R>G)}$, and AcGFP-NuMA$^r_{(Aro>A)}$ do not localize to the poles, and therefore their intensity at the poles cannot be determined (nd). Error bars: mean ± SD. Also, see EV4A, B for AcGFP fluorescence intensity at the poles divided by the microtubule intensity at the poles. Exact *p* values are ***$p < 0.001$ (i vs ii), ***$p < 0.001$ (i vs v), ***$p < 0.001$ (i vs vi), ***$p < 0.001$ (i vs vii), $p = 0.0814$ (ns) (v vs vii). nd represents not determined. *p* values are denoted as follows: ns-$p \geq 0.05$; ***$p < 0.001$ as determined by two-tailed unpaired Student's *t*-test. Scale bars in (C, F) represent 10 μm. Source data are available online for this figure.

---

tagged NuMA, where all the 24 glutamine residues in its IDR were mutated to glycine (NuMA$^r_{(Q>G)}$; Fig. 5E). Notably, we found that AcGFP-NuMA$^r_{(Q>G)}$ localizes to the poles, albeit significantly less than the wild-type AcGFP-NuMA$^r$ (Fig. 5E–G; compare v *vs.* i; also see Figs. EV4A,B,F and EV5D). Notably, the weakly localized AcGFP-NuMA$^r_{(Q>G)}$ failed to enrich robustly at the poles in cells treated with Aurora A inhibitor MLN8237, compared to AcGFP-NuMA$^r$ (Fig. 5E–G; compare i & ii vs. v & iv). Similarly, the AcGFP-NuMA$^r_{(Q>G)}$ construct consisting of a replacement of S1969 to non-phosphorylatable alanine [AcGFP-NuMA$^r_{(S1969A)(Q>G)}$] did not strongly accumulate at the poles, compared to AcGFP-NuMA$^r_{(Q>G)}$ (Fig. 5E–G; see vii; also see Figs. EV4A,B,F and EV5D). The subtle enrichment of AcGFP-NuMA$^r_{(Q>G)}$ at poles upon Aurora A inhibition could not be because of its weak localization, as cells that stably expressed low amounts of AcGFP-NuMA$^r$ strongly enrich AcGFP-NuMA$^r$ at the poles upon Aurora A inhibition (Fig. EV5E,F). Instead, the weak localization of AcGFP-NuMA$^r_{(Q>G)}$ at the poles is likely due to its highly dynamic nature, as monitored by FRAP analysis (Fig. EV5G–J), which is also supported by the above-mentioned Corelet experiment. Thus, we conclude that glutamine residues in the IDR of NuMA ensure homeostatic dynamicity of NuMA at the poles and promote material state transition of NuMA from a dynamic to a solid state in the absence of Aurora A activity.

## Artificially increased homotypic multivalent Interactions between NuMA molecules mimic Aurora A inactivation

Thus far, our data suggest that Aurora A phosphorylation at S1969 on NuMA may prevent strong homotypic multivalent interactions between NuMA molecules at poles. If this hypothesis is correct, artificially increasing NuMA multivalency should lead to a similar outcome as upon Aurora A inactivation. To test this, we made a fusion construct between NuMA and Kaede. Kaede is a homotetrameric protein with a size of 116 kDa (Dittrich et al, 2005). We reasoned that a fusion of Kaede with NuMA-which itself is capable of assembling into

dimer/multimer using its coiled-coil and intrinsically disordered C-terminus (Harborth et al, 1999) would lead to the formation of higher-order multiprotein assemblies of Kaede-NuMA fusion at the poles, somewhat analogous to cells that are inactivated for Aurora A kinase (Fig. 6A). As hypothesized, Kaede-NuMA fusion proteins were significantly enriched at the poles during metaphase and anaphase (Fig. 6B). Kaede-NuMA fusion protein complexes were also detected at the poles during telophase and G1 phase of the cell cycle, in contrast to monomeric AcGFP-tagged NuMA (Fig. 6B; Movies EV5 and EV6). Notably, Kaede-NuMA accumulated at the poles had the potential to bend the chromosome ensemble and nascent nuclei around it. This data presents a proof-of-principle experiment whereby increasing the multivalent interactions between NuMA molecules independent of Aurora A inactivation can force the accumulation of a non-dynamic pool of NuMA at the poles, which is sufficient to bend the newly formed nuclei around the poles, similarly to Aurora A inhibition.

## Inducing asymmetry in Aurora A localization drives asymmetry in NuMA accumulation

Active Aurora A, phosphorylated at threonine 288, strongly accumulates at centrosomes (Fig. EV1A; Magnaghi-Jaulin et al, 2019). We reasoned that if centrosome-localized Aurora A in the vicinity of NuMA-based poles is responsible for keeping NuMA in a dynamic state, then removing catalytically active Aurora A solely from one centrosome should create asymmetry in the material state (dynamic to solid) of NuMA and thus its accumulation. It was recently shown that in cells treated with the Polo-like kinase 4 (Plk4) inhibitor centrinone to block centriole duplication (Fig. 7A; Wong et al, 2015), NuMA accumulates significantly at the poles during metaphase (Appendix Fig. S4A; Chinen et al, 2020). Also, the fluorescence recovery profile of NuMA at the poles in the FRAP experiment was significantly slower in centrinone-treated cells compared to control cells (Fig. 7B–D; Chinen et al, 2020). Since Aurora A inhibition similarly impacts NuMA localization and its dynamicity at the poles, we wondered whether centrinone-treated

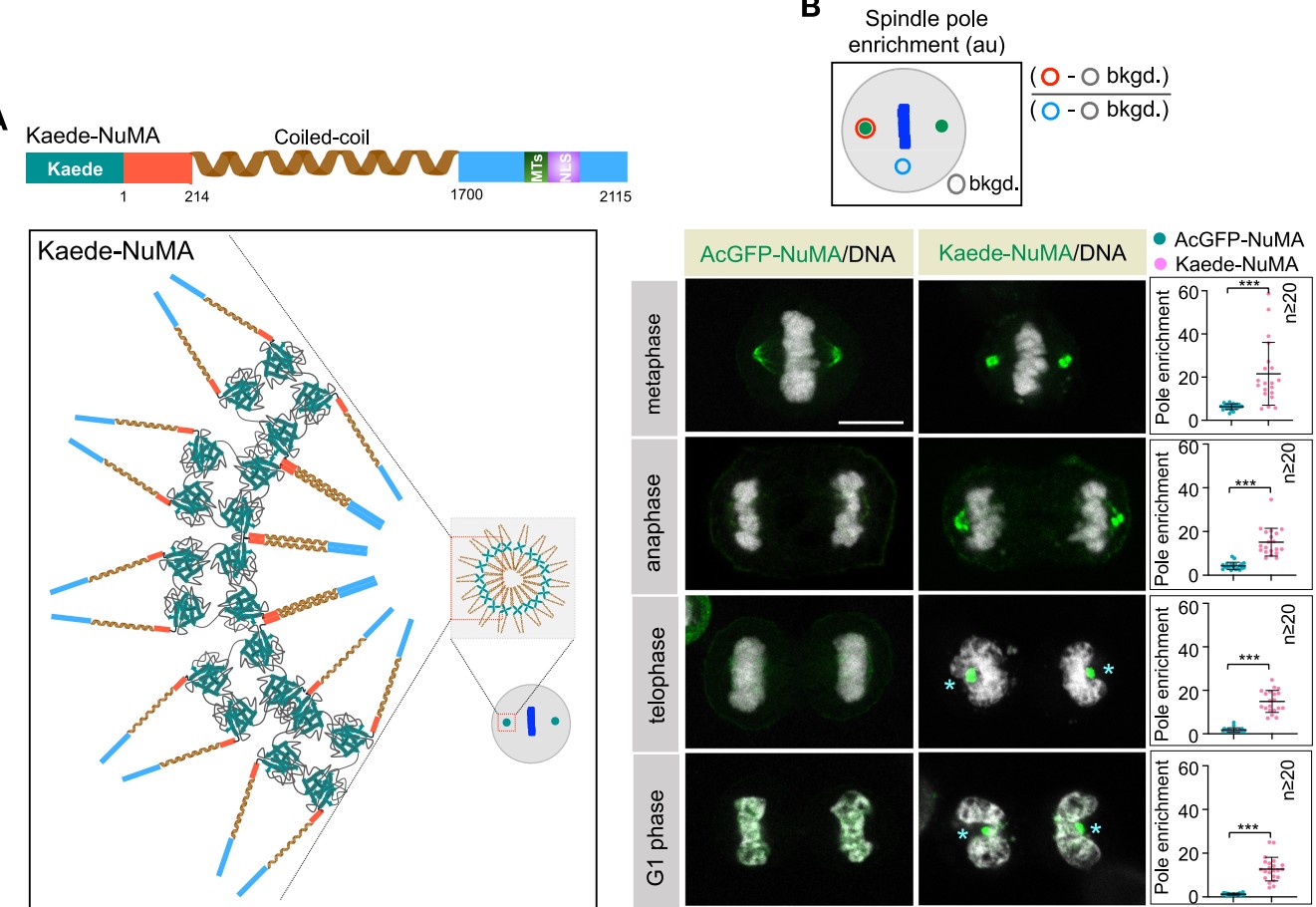

**Figure 6. Forcing multivalent interactions are sufficient to mimic Aurora A inactivation.**

(A) Schematic depiction of Kaede-tagged full-length NuMA and hypothesized oligomeric assemblies via NuMA's ability to assemble into dimers through its coiled-coil and Kaede's ability to assemble into tetramers. (B) Schematic of the method to quantify spindle pole intensity (in au) at the poles, and the representative images of HeLa cells expressing AcGFP-NuMA or Kaede-NuMA (in green) during various stages of the cell cycle, as indicated. The spindle pole accumulation of AcGFP-NuMA and Kaede-NuMA is quantified on the right for all these stages. DNA is shown in gray (also see Movies EV5 and EV6). Note that Kaede-NuMA is significantly enriched at the poles and does not dissolve at the poles post-mitosis. Asterisks in Kaede-NuMA-expressing cells depict the bending of the chromosome ensemble mass around the position where pole-localized Kaede-NuMA protein is present. Error bars: mean ± SD. Exact $p$ values from top to bottom panels are ***$p < 0.001$, ***$p < 0.001$, ***$p < 0.001$, ***$p < 0.001$. $p$ value is denoted as follows: ***-$p < 0.001$ as determined by two-tailed unpaired Student's $t$-test. Scale bar in (B) represents 10 µm. Source data are available online for this figure.

cells are deficient in accumulating active Aurora A at the centrosomes. Indeed, we discovered that T288$^P$ is absent in cells lacking centrosomes, assessed by the loss of the centriole-specific marker SAS6 (Fig. 7E). This observation allowed us to create an experimental scenario where the majority of the cells assemble a bipolar spindle with one centrosome by treating cells for 2 days with centrinone (Appendix Fig. S4B; Watanabe et al, 2020; Chinen et al, 2021). In such cells, the T288$^P$ signal was present at the pole with SAS6 signal and absent on the other pole (Fig. 7E). Importantly, in such a condition, NuMA was significantly enriched at a pole lacking T288$^P$, and this enrichment of NuMA at the pole without centrioles/SAS6 correlates with its significantly weak dynamicity in the FRAP analysis (Fig. 7F,G; Appendix Fig. S4C–E). Next, we checked if this asymmetry in NuMA accumulation at the poles could force the segregated chromosomes ensemble to bend around the pole with abnormal NuMA accumulation. As expected, the segregated chromosomes ensemble bent around the

pole with abnormal NuMA enrichment, in contrast to a pole containing wild-type weak levels of NuMA (Fig. 7H; Appendix Fig. S4F). Altogether, these results indicate that localized Aurora A activity at the centrosome is critical to keep NuMA in a dynamic state at the poles, and inducing asymmetry in active Aurora A at centrosomes is sufficient to induce asymmetry in the material state of NuMA and thus the organization of nascent nuclei at the mitotic exit.

## Proper nuclear morphology at mitotic exit is crucial for nucleoli organization and chromosome arm organization

The impact of Aurora A inactivation on nuclear morphology during mitotic exit may influence interphase genome organization. To evaluate this, we first analyzed the arrangements of nucleoli after nuclear envelope reformation. Nucleoli are composed of the nucleolar organizing regions, which are composed of the tandem

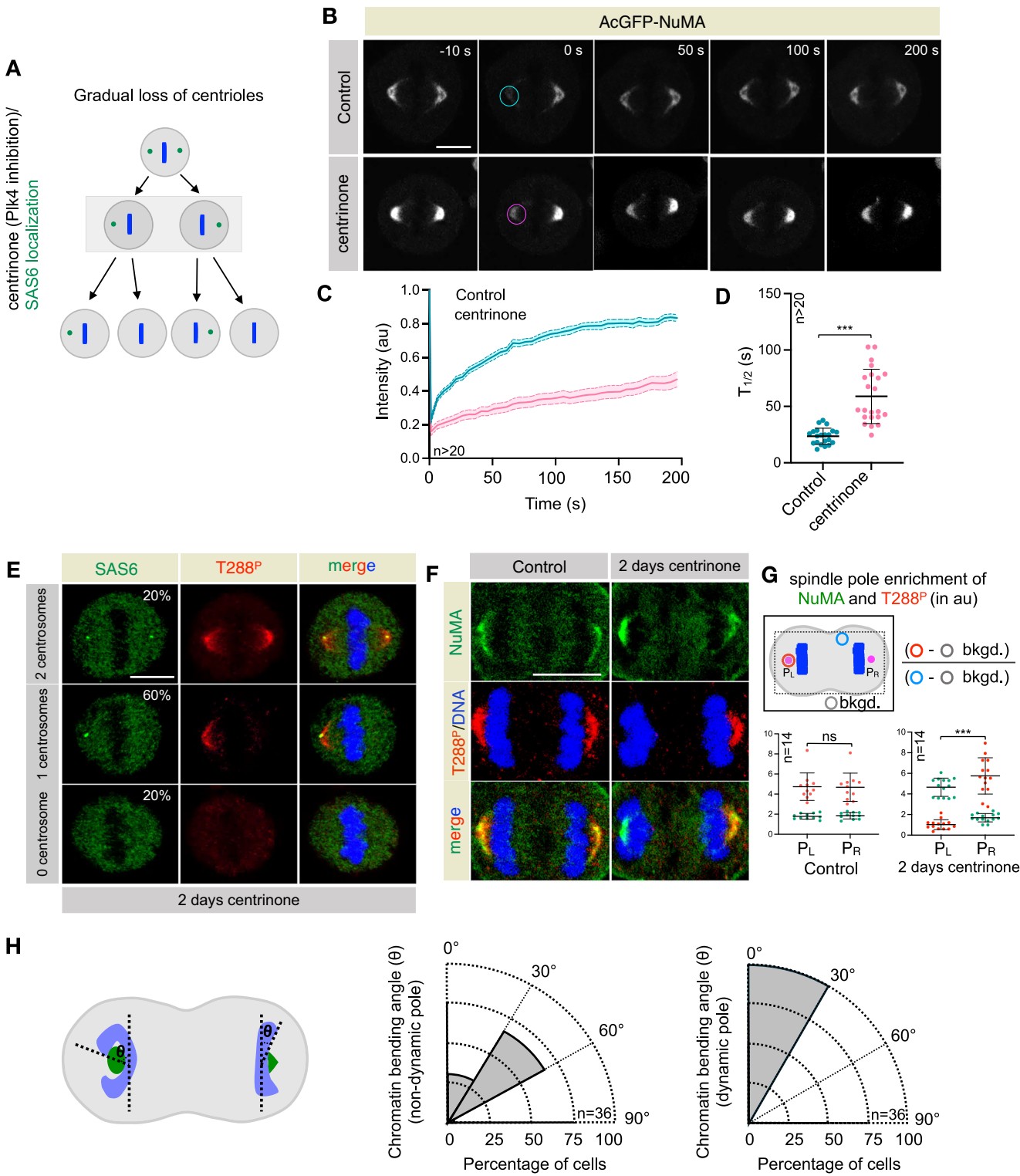

repeats of ribosomal DNA sequences (rDNA) on the short arms of the five acrocentric human chromosomes. We reasoned that Aurora A inactivation could lead to the repositioning of these DNA loci within the nucleus. We followed nucleolus dynamics in cells that are coexpressing the nucleolus marker AcGFP-Fibrillarin (Dundr et al, 2000; Lafontaine et al, 2021), mCherry-H2B, and SNAP-NuMA. In untreated control cells, AcGFP-Fibrillarin spots are evenly distributed along the long axis of the newly formed nuclei (Fig. 8A,B). In contrast, in MLN8237-treated cells, the nucleoli are considerably more unevenly distributed (Fig. 8A,B)

To scrutinize this redistribution of nucleoli upon acute Aurora A inhibition in more detail, we determined their genomic

**Figure 7.   Asymmetric localization of Aurora A is sufficient to generate asymmetry in NuMA accumulation.**

(A) Schematic depiction of centrinone-based centriole elimination. (B, C) FRAP analysis of HeLa cells stably expressing AcGFP-NuMA (gray) in the absence or presence of centrinone. Time is indicated in (s). The blue and magenta circles show the bleached poles for control and centrinone-treated cells, respectively (B). The AcGFP recovery profile of the bleached area corrected for photobleaching is plotted for 200 s (C). Curves and shaded areas indicate mean ± SEM. Note the substantial decrease in the FRAP recovery profile of centrinone-treated cells. (D) The half-time of recovery [$T_{1/2}$] of AcGFP-NuMA in control and centrinone-treated metaphase cells. Error bars: mean ± SD. Exact *p* value is ***$p < 0.001$. (E) Representative images from the IF analysis of HeLa cells during metaphase that are treated for 2 days with centrinone, as indicated. These cells were stained with anti-SAS6 (centriolar marker; green) and anti-T288$^P$ (red) antibodies. DNA is shown in blue. The % of cells with either two, one or no centrosomes is indicated (also see Appendix Fig. S4). (F) Representative images from the IF analysis of HeLa cells during anaphase that are treated for 2 days with centrinone, as indicated. These cells were stained with anti-NuMA (green) and anti-T288$^P$ (red) antibodies. DNA is shown in blue. (G) Schematic of the method to quantify spindle pole intensity (in au) of NuMA (green) and T288$^P$ (red) at the pole during anaphase for control and asymmetric centrosome-containing cells after 2 days of incubation with centrinone, as indicated. $P_L$ and $P_R$ represent left and right poles, respectively. In such cells, T288$^P$ containing poles are kept towards the right for representation and quantification analysis. Error bars: mean ± SD. Exact *p* values for control are $p = 0.6510$ (ns) for NuMA, $p = 0.9280$ (ns) for T288$^P$ and that of 2-days centrinone are ***$p < 0.001$ for NuMA, $p < 0.001$ for T288$^P$. (H) Schematic of the method to quantify the bending angle of segregated mitotic chromosome ensemble around the abnormal and normal NuMA poles during anaphase at the time of cleavage furrow formation. The data is obtained by confocal live-cell imaging of cells stably expressing AcGFP-NuMA in the presence of SiR-DNA and are treated with centrinone for 2 days (see Appendix Fig. S4F for representative images). *p* values are denoted as follows: ns- $p \geq 0.05$; ***$p < 0.001$ as determined by two-tailed unpaired Student's *t*-test. Scale bars in (B, E, F) represent 10 μm. Source data are available online for this figure.

surroundings using high-resolution 4C-seq (Circular Chromosome Conformation Capture followed by sequencing; Simonis et al, 2006). Cells were synchronized in prometaphase using nocodazole, followed by synchronized release into early G1 in the presence or absence of MLN8237 (Fig. 8C). Taking profit from the recently released telomere-to-telomere human genome assembly (Nurk et al, 2022), we designed a 4C-seq viewpoint that mapped >200 times to the rDNA tandem repeats on the short arms of the acrocentric chromosomes. This setup thus allows us to identify the combined chromosome contacts for all five nucleolar organizing regions in a single experiment (Fig. 8D).

Mapping of 4C-seq data revealed that MLN8237-treated and control cells similarly interacted with the non-acrocentric chromosomes that do not contain rDNA clusters (Fig. 8E and Appendix Fig. S5A,B; intra-chromosomal versus inter-chromosomal 4C-seq signal). We thus conclude that the uneven distribution of nucleoli along the long axis of the newly formed nucleus does not noticeably reorganize the interactions of the nucleolar organizing regions with genomic regions on other chromosomes. In contrast, we did observe more prominent changes in intra-chromosomal contacts. Focusing on the short arm (i.e., intra-arm) versus long arm (i.e., inter-arm) contacts, the MLN8237-treated rDNA clusters increased their contacts with the long arms (i.e., the arms that do not carry the nucleolar organizing regions) (Fig. 8D,F,G; Appendix Fig. S5C–F). To extend these observations beyond the rDNA clusters, we repeated these experiments using the promoter of the active *FLT3* gene as a viewpoint (Diamant et al, 2025). *FLT3* is located close to the centromere of acrocentric chromosome 13, but on the long arm (Appendix Fig. S5G). Like the rDNA clusters, *FLT3* does not noticeably change its inter-chromosomal contacts. Yet, increased short arm contacts are observed, although contacts across the centromere of this single-copy gene remain scarce (Appendix Fig. S5G). The perturbed nuclear organization and more random distribution of nucleoli upon inactivation of Aurora A activity is therefore accompanied by an increase in inter-arm chromosome contacts, reflecting a less efficient separation of the chromosome arms. Whereas this increase (in the order of 10–50%) may appear minor, previous work reported that contacts within the 10–100 Mb range from early to late G1 changed by a maximum of twofold (Zhang et al, 2019). These experiments thus demonstrate that disrupted Aurora A function during mitotic exit and the

associated perturbed nuclear morphology lead to altered reestablishment of interphase chromosome organization.

## Discussion

In this study, we show that centrosomal Aurora A activity during anaphase maintains NuMA in a dynamic state at the spindle poles (Fig. 9A). In the absence of Aurora A activity, NuMA undergoes a material state transition to a solid state and abnormally accumulates at the poles. These aberrant poles interfere with proper chromosome segregation by causing the segregated chromosomes to wrap around them, thereby disrupting the organization of the developing nuclei at the mitotic exit (Fig. 9B).

To understand the molecular mechanisms underlying NuMA accumulation and its transition from a dynamic to a solid state at the poles upon Aurora A inhibition, we performed extensive deletion and mutagenesis analyses. Previous studies using dynein/dynactin-specific antibodies or dynein light chain knockout cells have highlighted the role of dynein/dynactin in NuMA accumulation at the poles (Merdes et al, 2000; He et al, 2023). Consistently, our data indicate that the N-terminus of NuMA, which directly interacts with dynein/dynactin (Kotak et al, 2012; Okumura et al, 2018; Renna et al, 2020), is essential for its proper enrichment at the poles. In human cells, NuMA is present at sub-micromolar concentrations (<1 μM; Hein et al, 2015), and thus, for efficient pole localization, it initially relies on motor dynein/dynactin (Fig. 9A). Once delivered, homotypic multivalent interactions mediated by NuMA coiled-coil region and its C-terminal intrinsically disordered region (IDR) ensure its accumulation at the poles (Fig. 9A). Because IDR-IDR interactions are typically weak and transient (Banani et al, 2017; Holehouse and Kragelund, 2024), we propose that the coiled-coil domain structurally stabilizes weak cation-π interactions, specifically, between arginine and aromatic residues within the IDR sequence. Supporting this notion, our Corelet assay reveals that condensates formed by NuMA construct containing both the coiled-coil and the C-terminal IDR are substantially more stable than those formed by the IDR alone. We speculate that this might represent a broader mechanism by which weakly interacting IDRs achieve multivalent stabilization in cellular contexts.

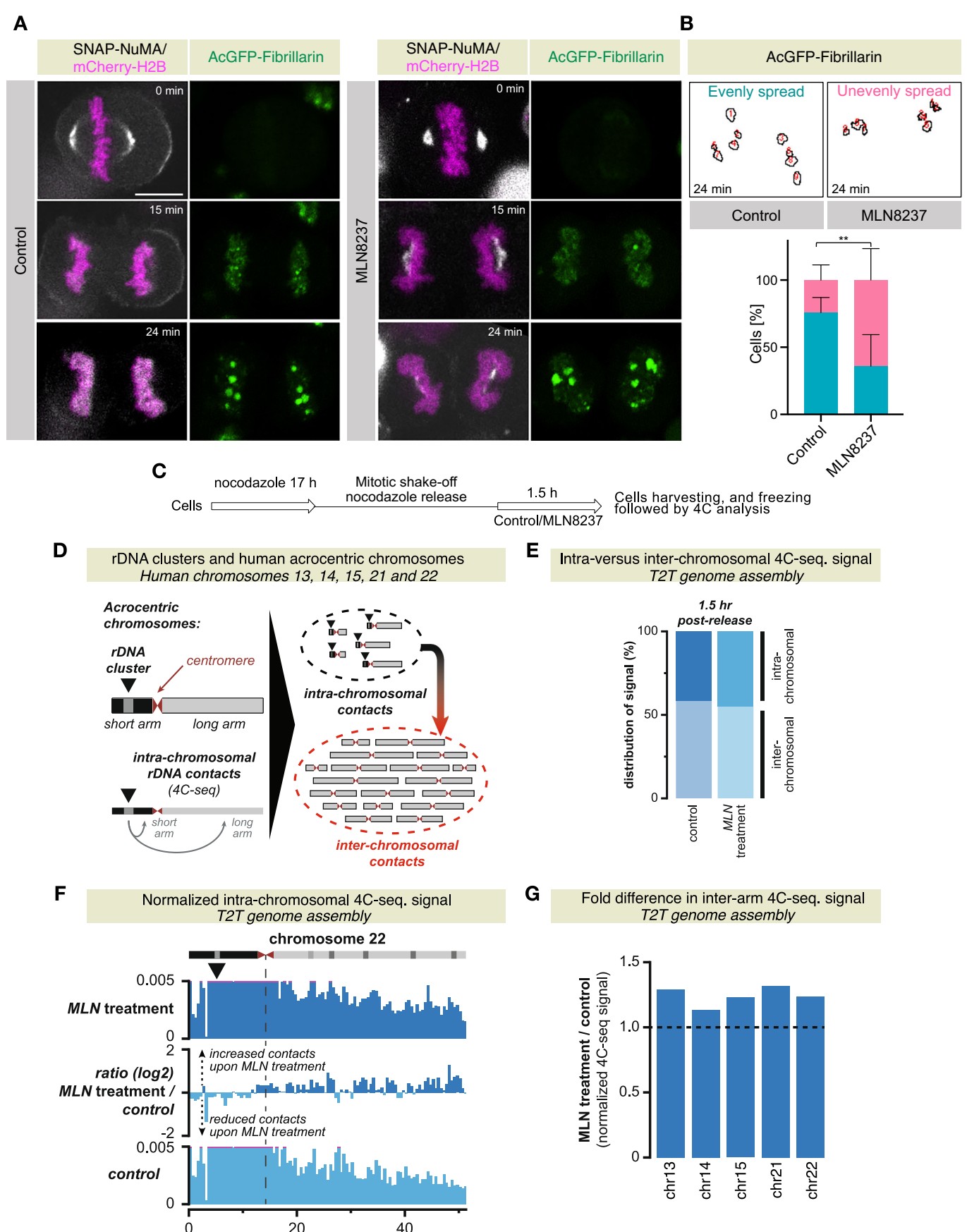

**Figure 8. Abnormal accumulation of NuMA at the poles impacts the organization of nucleoli and rDNA chromosome contacts.**

(A) Confocal live-cell imaging of HeLa cells stably coexpressing AcGFP-Fibrillarin (green), and mCherry-H2B (magenta) and are transiently transfected with SNAP-NuMA (gray) and are either treated with DMSO (control) or acutely treated with Aurora A inhibitor MLN8237. The recording was started 20 min post-incubation with DMSO or MLN8237. Timepoint $t = 0$ min was set to the metaphase-to-anaphase transition. (B) Analysis of AcGFP-Fibrillarin foci, which were either categorized as evenly spread (turquoise) or unevenly spread (pink) along the long axis of the segregated chromosomes. Error bars: mean ± SD from three independent experiments ($n > 18$ each). Exact p value is **$p = 0.0051$. (C) Method for the synchronization of HeLa cells at the mitotic exit while untreated or treated with MLN8237 for 4C-seq analysis. (D) Left: schematic outline of the acrocentric human chromosomes and the location of the rDNA clusters on the short (left) chromosomal arms. Below, the analysis of intra-chromosomal short- and long arm contacts using the rDNA clusters as 4C-seq viewpoints. Right: intra-chromosomal versus inter-chromosomal contacts of the rDNA clusters using 4C-seq. (E) Quantification of the combined 4C-seq signal on the five acrocentric chromosomes (intra-chromosomal signal) versus signal on the other chromosomes (inter-chromosomal signal) for cells with and without Aurora A inhibition (MLN8237 treatment) at 1.5 h after nocodazole release. (F) Examples of normalized intra-chromosomal 4C-seq signal for the rDNA clusters on the acrocentric chromosomes 13 and 22. The top panels show the signal upon MLN8237 treatment, and the bottom panels signal in control cells. The difference between MLN8237 treatment and controls is indicated in-between (log2 ratio). Black arrows below the chromosome ideograms indicate the centre of the rDNA clusters where the 4C-seq viewpoints are located. (G) Quantification of contacts with the long chromosome arm upon MLN8237 treatment or control conditions for individual chromosomes. Long arm contacts are consistently increased upon MLN8237 treatment after nocodazole release, indicative of a reduction in chromosome arm separation. p-values are denoted as **$p < 0.01$ as determined by two-tailed unpaired Student's *t*-test. Scale bar in (A) represents 10 µm. Source data are available online for this figure.

NuMA contains two microtubule-binding domains within its IDR: MTBD1 (aa 1914–1985; Du et al, 2002; Haren and Merdes, 2002) and MTBD2 (aa 2002–2115; Gallini et al, 2016). A possible explanation for the failure of NuMA$_{(R>G)}$ or NuMA$_{(Aro>A)}$ mutants to localize at the poles could be compromised microtubule binding. However, deletion of MTBD1 impairs spindle pole focusing but not NuMA localization (Haren and Merdes, 2002; Silk et al, 2009; Tsuchiya et al, 2021), and similarly, deletion of MTBD2 does not affect NuMA localization to the poles (Tsuchiya et al, 2021). These findings suggest that the loss of pole localization in NuMA$_{(R>G)}$ or NuMA$_{(Aro>A)}$ mutants is unlikely to stem from impaired microtubule binding. Based on Corelet assays with NuMA C-terminal fragments bearing mutations in either arginine [NuMA$_{(R>G)}$] or aromatic [NuMA$_{(Aro>A)}$] residues, we propose that these residues are important for promoting multivalent interactions required for NuMA accumulation at the poles. However, we cannot fully exclude the alternative possibility that these mutations may also disrupt NuMA interactions with specific mitotic partners, thereby contributing to the observed defects in pole accumulation.

Notably, Aurora A phosphorylates NuMA at serine 1969 within its IDR. Replacing this serine with a non-phosphorylatable alanine causes abnormal NuMA accumulation at poles similar to what is observed with Aurora A inactivation. Because post-translational modifications are implicated in modulating the material properties of several proteins (Zbinden et al, 2020; Babinchak and Surewicz, 2020), we hypothesize that the conformational changes in NuMA IDR facilitated by S1969 phosphorylation blocks strong multivalent interactions between NuMA IDRs at the poles, which otherwise could be pathogenic. Supporting this hypothesis, we show that artificially modulating NuMA multimerization by fusing it with a tetrameric fluorescent protein Kaede is sufficient for NuMA accumulation at the poles, similar to what is seen upon Aurora A inactivation. Also, we failed to observe enrichment in some of the essential centrosomal components upon Aurora A inactivation during anaphase, indicating that the solid-to-dynamic state transition of NuMA during mitosis is directly linked to S1969 phosphorylation rather than changes in the NuMA interactome at the pole. Altogether, our work suggests that Aurora A phosphorylation at S1969 maintains NuMA in a dynamic state by preventing strong multivalent interactions between NuMA IDR. Loss of this phosphorylation forces NuMA to adopt compact solid assemblies that abnormally accumulate at the poles.

What amino acids are engaged in strong multivalent interaction, which are normally suppressed in the presence of active Aurora A kinase at the centrosome? Our work reveals the relevance of 24 glutamine residues in the IDR in promoting the hardening (solid-state) of NuMA in cells lacking Aurora A activity. Understanding how Aurora A phosphorylation blocks glutamine-mediated inter-actions and thereby keeping NuMA in a dynamic state warrants further investigation. However, one possibility is that glutamine residues in NuMA IDR are involved in forming labile cross-beta sheets at the poles, similar to FUS (Murray et al, 2017; Babinchak and Surewich, 2020), and Aurora A phosphorylation at S1969 prevents the formation of such assemblies and thus hardening. Moreover, arginine and aromatic residues present in NuMA IDR may also promote its transition from dynamic to solid upon Aurora A inhibition. However, since these residues are also needed for NuMA accumulation at poles, we could not test their relevance in promoting solid-state transition.

Super-resolution imaging of NuMA reveals that it assembles into a network at the poles, which we refer to as meshwork. However, when Aurora A is inhibited, these assemblies become more compact. This behavior is reminiscent of the Drosophila pericentriolar material (PCM) protein Centrosomin (Cnn) and its *C. elegans* homolog SPD-5. During mitosis, Cnn (or SPD-5) is phosphorylated by Plk1 at the centrosome, promoting scaffold assembly around centrioles and facilitating centrosome maturation (Conduit et al, 2014; Rios et al, 2024). Thus, we propose that eukaryotic cells have evolved multiple centrosome/pole-localized proteins with an inherent ability to form scaffolds at the centrosome/poles, ensuring proper microtubule nucleation, spindle assembly, and spindle integrity.

Interestingly, in Aurora A-inhibited cells, in addition to abnormal NuMA accumulation at the poles, we observed that NuMA is only weakly localized in the developing nuclei, particularly during the early stages of mitotic exit. Therefore, the impact of Aurora A inhibition on nuclear organization and on chromatin contacts measured by 4 C analysis could stem from its (1) abnormal retention at the poles, (2) reduced nuclear accumulation, or (3) a combination of both. However, since cells expressing NuMA$_{\Delta NLS}$ do not exhibit nuclear shape organization defects, we infer that at least nuclear shape defects are primarily associated with abnormal NuMA accumulation at the poles rather than insufficient NuMA levels in the nucleus.

Aurora A regulates several mitotic processes, and since it is often amplified in cancers with poor prognoses, it represents a promising target for cancer therapy. Over the past decade, several highly specific Aurora A inhibitors (e.g., alisertib [MLN8237]) have been developed and tested in clinical trials, but with limited success (O'Connor et al, 2019). Our research highlights a "new" post-mitotic role for Aurora A in nuclear organization, suggesting that Aurora A inhibitors may have unforeseen post-mitotic effects if not carefully assessed. Therefore, it is essential to gain a deeper fundamental understanding of Aurora A (and other mitotic kinases) before these inhibitors are used in clinical settings to ensure they are both effective and safe for patients.

# Methods

### Reagents and tools table

| Reagent/resource | Reference or source | Identifier or catalog number |
|---|---|---|
| **Experimental models** | | |
| *E. coli* DH5α | | |
| Human: HEK293 | | |
| Human: U2OS | | |
| Human: hTERT-RPE1 | | |
| Human: HeLa "Kyoto" cells | | |
| AcGFP (*Aequorea coerulescens* GFP)-Fibrillarin; mCherry-H2B | This study | |
| AcGFP-LaminB; mCherry-NuMA | This study | |
| AcGFP-LaminB; mCherry-H2B | This study | |
| AcGFP-NuMA$^r$; H2B-mCherry | This study | |
| AcGFP-Nup107 | This study | |
| Aurora A$^r$ -AcGFP | This study | |
| CycB-AcGFP | This study | |
| CycB-Aurora A$^r$-AcGFP | This study | |
| CycB-Aurora A$^r$-AcGFP; mCherry-NuMA | This study | |
| **Recombinant DNA (Plasmids)** | **This study** | Appendix Table S1 |
| **Antibodies** | | |
| Anti-Aurora A (1:200) | Cell Signaling Technology | Cat# 4718, RRID:AB_2061482 |
| Anti-CDK5RAP2 (1:500) | Merck Millipore | Cat# 06-1398, RRID:AB_11203651 |
| Anti-CEP152 (1:200) | Bethyl Laboratories | Cat# A302-480A, RRID:AB_1966084 |
| Anti-CEP192 (1:1000) | Bethyl Laboratories | Cat# A302-324A, RRID:AB_1850234 |
| Anti-CyclinB1 (GNS1) (1:1000) | Santa Cruz Biotechnology | Cat# sc-245, RRID:AB_627338 |
| Anti-GFP (1:20,000) | Invitrogen | Cat# A-11122, RRID:AB_221569 |
| Anti-GFP (1:500) | Developmental Studies Hybridoma Bank | Cat# 8H11, RRID:AB_2617423 |
| Anti-mAB414 (1:1000) | BioLegend | Cat# 902907, RRID:AB_2734672 |
| Anti-mCherry (1:500) | Developmental Studies Hybridoma Bank | Cat# 3A11, RRID:AB_2617430 |
| Anti-NuMA (1:300) | Santa Cruz Biotechnology | Cat# sc-365532, RRID:AB_10846197 |
| Anti-NuMA (1:200) | Santa Cruz Biotechnology | Cat# sc-48773, RRID:AB_2154276 |
| Anti-pAurora A (1:200) | Cell Signaling Technology | Cat# 3079, RRID:AB_2061481 |

| Reagent/resource | Reference or source | Identifier or catalog number |
|---|---|---|
| Anti-PCNT (1:500) | Abcam | Cat# ab4448, RRID:AB_304461 |
| Anti-SAS6 (1:200) | Santa Cruz Biotechnology | Cat# sc-81431, RRID:AB_1128357 |
| Anti-α-tubulin (1:1000) | Sigma-Aldrich | Cat# T6199, RRID:AB_477583 |
| Anti-β-actin-Peroxidase (1:20,000) | Sigma-Aldrich | Cat# A3854, RRID:AB_262011 |
| Anti-β-tubulin (1:500) | Abcam | Cat# ab6046, RRID:AB_2210370 |
| Anti-Υ-tubulin (1:1000) | Sigma-Aldrich | Cat# T5326, RRID:AB_532292 |
| Secondary: Alexa488-conjugated anti-mouse (1:500) | Molecular Probes | Cat# A-21121, RRID:AB_2535764 |
| Secondary: Alexa488-conjugated anti-rabbit (1:500) | Molecular Probes | Cat # A-11008, RRID:AB_143165 |
| Secondary: Alexa568-conjugated anti-mouse (1:500) | Molecular Probes | Cat# A-11004, RRID:AB_2534072 |
| Secondary: Alexa568-conjugated anti-rabbit (1:500) | Molecular Probes | Cat# A-11011, RRID:AB_143157 |
| Secondary: Goat anti-mouse IgG antibody, HRP conjugate (1:5000) | Bethyl Laboratories | Cat# A90-116P, RRID:AB_67183 |
| Secondary: Goat anti-rabbit IgG antibody, HRP conjugate (1:5000) | Bethyl Laboratories | *Cat# A120-101P, RRID:*AB_67264 |
| **Oligonucleotides and other sequence-based reagent** | | |
| **siRNAs** | | |
| siNuMA (sense: CCAGACAGCGCCAACUCAUCGUUCU) | Eurogentec (mentioned in- Sana et al, 2022) | |
| siAurora A (sense: AGAAUCCAUUACCUGUAAAU) | Eurogentec | |
| siAurora A | Qiagen | S102223305 |
| **Primers** | **This study** | Appendix Table S2 |
| **Chemicals, enzymes, and other reagents** | | |
| BSA | HiMedia | Cat# MB083 |
| Centrinone | MedChemExpress | Cat# HY-18682 |
| Clarity Western ECL Substrate | Bio-Rad | Cat# 170-5060 |
| DMEM | HiMedia | Cat# AL007A |
| DMEM | MP Biomedicals | Cat# 091233354 |
| DMSO | Sigma-Aldrich | Cat# D8418 |
| DTT | Sigma-Aldrich | Cat# D9163 |
| Dulbecco's Phosphate Buffered Saline | HiMedia | Cat# TL1006 |
| EDTA | Sigma-Aldrich | Cat# E5134 |
| EGTA | Sigma-Aldrich | Cat# E3889 |
| Expand Long Template PCR system | Roche Diagnostics | Cat# 11681842001 |
| FBS | Gibco | Cat# 10270106 |
| Fluoromount-G | SouthernBiotech | Cat# 0100-01 |
| Geneticin | Life Technologies | Cat# 10131-035 |
| Glycerol | Sigma-Aldrich | Cat# G5516 |
| Glycine | SRL | Cat# 66327 |
| Hoechst 33342 | Sigma-Aldrich | Cat# B2261 |
| Immobilon Forte Western HRP Substrate | Merck Millipore | Cat# WBLUF0500 |
| jetPRIME transfection reagent | Polyplus | Cat# 101000046 |
| Lipofectamine 2000 | Life Technologies | Cat# 11668019 |
| *MboI* restriction enzyme | New England Biolabs | Cat# R0147 |
| Methanol | Merck | Cat# 34860 |
| MK5108 (IC50-0.064 nM) | TargetMol | Cat# T6068 |
| MLN8237/Alisertib (IC50-1.2 nM) | Selleckchem | Cat# S1133 |
| NaCl | Merck | Cat# 106404 |

| Reagent/resource | Reference or source | Identifier or catalog number |
|---|---|---|
| Nitrocellulose membrane | Bio-Rad | Cat# 1620112 |
| *NlaIII* restriction enzyme | New England Biolabs | Cat# R0125 |
| Nocodazole | Sigma-Aldrich | Cat# M1404 |
| NP-40 (Tergitol) | Sigma-Aldrich | Cat# NP40S |
| PMSF | Sigma-Aldrich | Cat# P7626 |
| Protease Inhibitor Cocktail Set III, EDTA-Free | Merck Millipore | Cat# 539134 |
| Puromycin Dihydrochloride | Life Technologies | Cat# A11138-03 |
| Qubit ds DNA BR kit | Thermo Fisher Scientific | Cat# Q32850 |
| RNAiMAX | Invitrogen | Cat# 13778150 |
| RO-3306 | Sigma-Aldrich | Cat# SML0569 |
| SDS | Qualigens | Cat# Q27825 |
| SiR-DNA | Spirochrome | Cat# SC007 |
| Skimmed milk powder | HiMedia | Cat# GRM1254 |
| SNAP-Cell 647-SiR | New England Biolabs | Cat# S9102S |
| Sodium deoxycholate | Sigma-Aldrich | Cat# D6750 |
| STLC | Sigma-Aldrich | Cat# 164739 |
| TapeStation Genomic DNA reagents | Agilent | Cat# 5067-5366 |
| TapeStation Genomic DNA ScreenTape | Agilent | Cat# 5067-5365 |
| TEMED | Sigma-Aldrich | Cat# T9281 |
| Thymidine | Sigma-Aldrich | Cat# T1895 |
| Tris | SRL | Cat# 71033 |
| Triton X-100 | Bio-Rad | Cat# 1610407 |
| Trypsin | HiMedia | Cat# TCL007 |
| Tween-20 | Bio-Rad | Cat# 1706531 |
| UltraPure Distilled Water | Invitrogen | Cat# 10977-015 |
| **Software** | | |
| GraphPad Prism | https://www.graphpad.com/ | RRID:SCR_002798 |
| ImageJ/Fiji | http://fiji.sc/ | RRID:SCR_002285 |
| Imaris | http://www.bitplane.com/imaris/imaris | RRID:SCR_007370 |
| IUPred | https://iupred3.elte.hu/ | RRID:SCR_014632 |
| PLAAC | http://plaac.wi.mit.edu/ | RRID:SCR_024973 |
| C4ctus | https://github.com/NoordermeerLab/c4ctus | - |
| SnapGene | http://www.snapgene.com/ | RRID:SCR_015053 |

## Method details

### Cell lines and cell culture

The cell lines used in this study were HeLa Kyoto, hTERT-RPE1, U2OS and HEK293. All these cell lines were cultured in high glucose Dulbecco's Modified Eagle Medium (DMEM) supplemented with 10% heat-inactivated fetal bovine serum (FBS) and 1X antibiotic solution containing penicillin and streptomycin at 37 °C in a humidified 5% $CO_2$ incubator. All the stable cell lines used in this study were generated in HeLa Kyoto cells (a kind gift from Daniel Gerlich, IMBA, Vienna). HEK293 cells were kindly provided by Dr. Amit Singh (Indian Institute of Science, Bangalore).

### Generation of stable cell lines

HeLa Kyoto cells stably expressing the specified recombinant constructs were generated by transfecting with 4 µg of plasmid DNA suspended in 500 µl of jetPRIME buffer. The cells were then incubated for 5 min, followed by the addition of 8 µl of jetPRIME transfection reagent and given an additional 20 min of incubation. The mixture was added to the cells plated at 80% confluency in a 10 cm dish. The medium was replaced after 12 h of transfection. After 36 h of transfection, DMEM containing puromycin (400 ng/mL) and/or G418 sulfate (0.4 mg/mL) was added for selection. The clones were isolated, analyzed, and confirmed by immunofluorescence and immunoblotting analysis.

### Transient transfection

For transient plasmid transfection, 2–4 µg of plasmid DNA was mixed with 400 µl of serum-free DMEM and incubated for 5 min. After incubation, 6 µl of Lipofectamine 2000 was added to the mixture and incubated for 15–20 min. This mixture was added to cells plated at 80% confluency on coverslips or the imaging dish. The medium was replaced after 12 h of transfection. The cells were fixed or used for immunostaining or live imaging after 24–36 h of transfection.

### Small-interfering RNA transfection

For siRNA transfection, 9 µl of 20 µM of the siRNA and 4 µl of Lipofectamine RNAiMAX reagent were mixed with 100 µl of nuclease-free water side-by-side. After 5 min of incubation, these solutions were mixed and incubated for another 20 min. This mixture was added to the cells at 30–40% confluency on coverslips in 35 mm dish or imaging dish. The medium was replaced after 12 h of siRNA transfection. The cells were grown for a total of 60–72 h, followed by fixation in methanol and immunostaining or live-imaging analysis.

### Drug treatments

All the cell lines used in this study were treated with 50–250 nM of MLN8237 for Aurora A inhibition for 15 min–2 h. For obtaining cells with asymmetric NuMA accumulation through Plk4 inhibition by centrinone (Fig. 7F–H; Appendix Fig. S4), cells were treated with 100 nM centrinone for 2 days. About 500–2000 nM nocodazole treatment was given for 17 h for microtubule depolymerization experiments. Following the drug treatments, the cells were analyzed using immunofluorescence, live imaging, or immunoblot analysis.

### Indirect immunofluorescence and live imaging

The cells were grown on autoclaved coverslips, and after the transfection with siRNA/ plasmids or drug treatments, they were fixed by cold methanol (−20 °C) and incubated at −20 °C for 8–10 min. They were washed with PBS and permeabilized with PBST (PBS with 0.05% Triton X-100) for 5 min, followed by blocking with 1% BSA (in PBST) for 1 h. Then, the cells were incubated with appropriate primary antibodies, diluted in 1% BSA for 4 h and washed thrice with PBST at 5 min intervals. Following this, the cells are incubated with fluorophore-labeled secondary antibody for 1 h and washed thrice with PBST at 5 min intervals. Further, the cells were incubated with 1 µg/µL Hoechst 33342 for 5 min, followed by three PBST washes. After that, the coverslips were mounted using Fluoromount-G. The images were acquired by an Olympus FV 3000 confocal laser scanning microscope using a

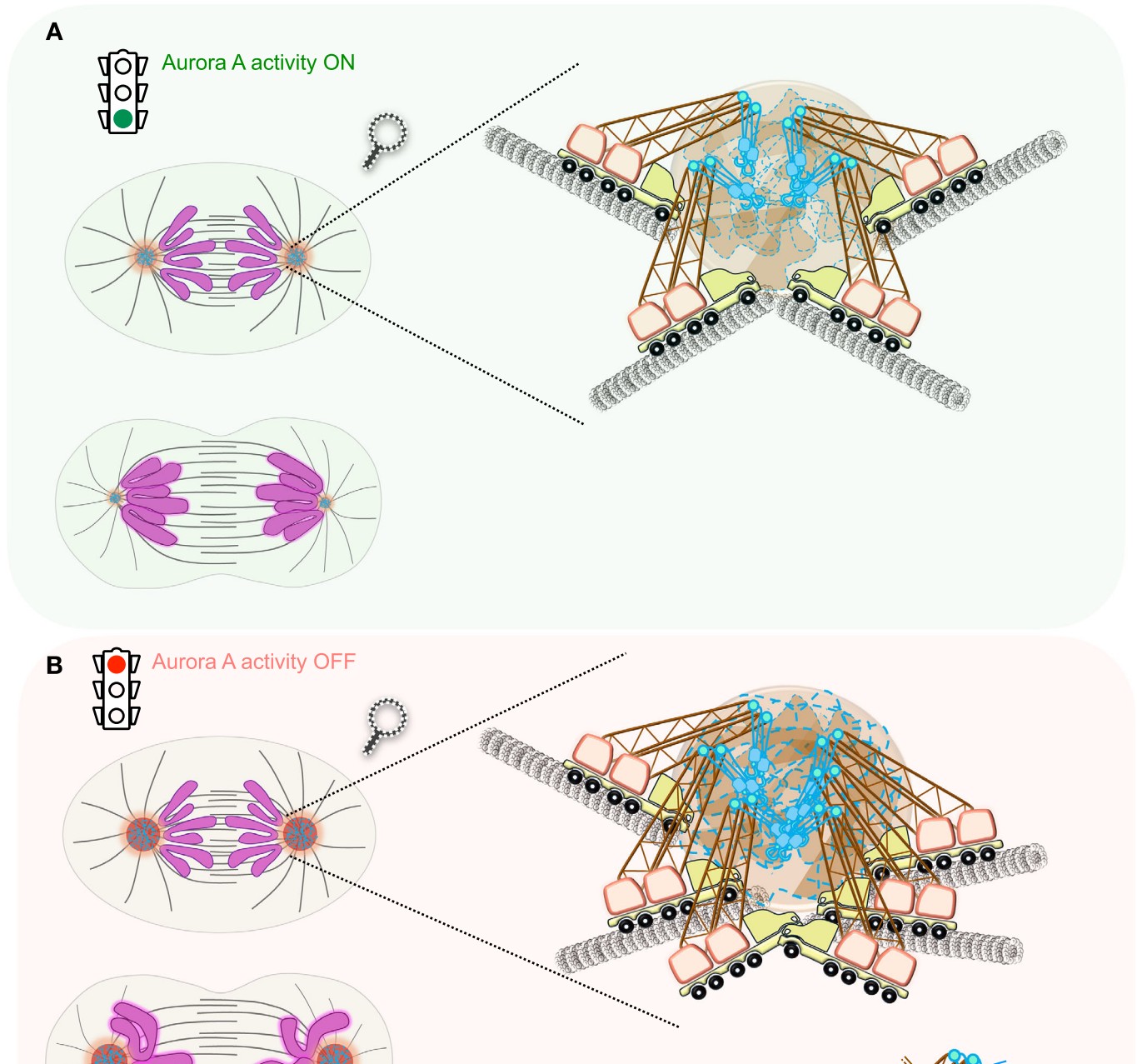

**A** Aurora A activity ON

**B** Aurora A activity OFF

Coiled-coil

IDR

N-ter N-ter

Dynein/dynactin    NuMA (dimer)

60X (NA 1.4) oil immersion objective. The acquired images were analyzed and processed using ImageJ software, preserving the relative intensities.

The time-lapse live imaging of cells was performed on an Olympus FV 3000 confocal laser scanning microscope using a 40X (NA 1.3) oil immersion objective (Olympus Corporation). The images were captured by the inbuilt FV 3000 software at 1, 2, 3, and 5 min intervals, with 9–11 optical sections (3-μm apart). During the imaging, the cells expressing AcGFP/mCherry/SNAP/ mEOS tags were maintained at optimal growth conditions (37 °C,

**Figure 9.  Aurora A regulates the material properties of NuMA-based poles to ensure organization of chromosome ensemble at the mitotic exit.**

(A, B) Model for the coordination between accumulation of NuMA at the poles and chromosome organization at the mitotic exit. In unperturbed control cells, centrosomal Aurora A activity keeps NuMA at the pole in a dynamic state to ensure that NuMA levels at the poles are gradually decreased so that the mitotically segregated chromosome ensemble can form nascent nuclei without any physical hindrance by the pole in late anaphase (A). Inset highlights that dynein/dynactin (depicted as trucks) carry a dimer of NuMA (shown as cranes) to the poles where the intrinsically disordered regions (IDR; represented as hook and slings) present in the C-terminus of NuMA are engaged in weak multivalent interactions, ensuring its localization and dynamicity. On the contrary, in the absence of centrosomal Aurora A activity, NuMA undergoes a material state transition from dynamic to solid and fails to dissolve at the poles. This leads to the wrapping of chromosome ensemble around the non-dissolved NuMA poles and, therefore, change in nuclear organization post-mitosis (B). Inset highlights that in the absence of Aurora A activity, NuMA (cranes) are engaged in strong multimeric interactions at the poles via their IDR sequences, leading to a change in the material property of NuMA from dynamic to solid. This work reveals that the coordination between poles (membraneless organelle) and developing nuclei (membrane organelle) is essential for proper nuclear shape organization.

5% $CO_2$ and 90% relative humidity) by a Tokai Hit STR Stage Top incubator.

The super-resolution imaging of NuMA and PCNT was done using a 3D-lattice-structured illumination super-resolution microscope (3D-SIM[2]) from Zeiss Elyra 7 with Lattice SIM[2] model, with a 63X oil immersion objective with 1.4 NA. As NuMA is accumulated significantly more at the spindle poles following Aurora A inhibition on MLN8237 treatment, NuMA structures were visualized accurately without overexposing the spindle poles. As a result, control and MLN8237-treated samples were imaged using different laser intensities as represented in Fig. 3G.

For quantitative analysis of the NuMA signal at the poles, the non-saturated NuMA intensity in MLN8237-treated/or CycB-Aurora A expressing cells was used to capture images, and similar laser intensities were utilized to capture images of control cells before quantification.

### Preparation of cell extracts and immunoblotting analysis

The cells were synchronized at prometaphase using 100 nM nocodazole for 17–20 h. For anaphase synchronization, 10 μM of RO-3306 was added to the nocodazole-treated cells for 15–20 min. MLN8237 is added if required. The cells were collected and suspended in a lysis buffer containing Tris pH 7.4 (50 mM), NP-40 (1% v/v), sodium deoxycholate (0.25%), NaCl (150 mM), SDS (0.01%), PMSF (0.1 mM), and protease inhibitor cocktail (complete, EDTA-free, 1:1000). The cells were then incubated on ice for 1.5 h and centrifuged at 13,000 rpm for 10 min at 4 °C, the supernatant was collected, and protein concentration was quantified by Bradford assay.

The protein samples were normalized to 2 μg/μL using 2X SDS loading dye containing β-mercaptoethanol (4.9%), and the samples were heated at 98 °C for 10 min. About 20–50 μg of the protein samples were loaded onto 6–10% SDS PAGE gels (according to the size of the protein of interest) and resolved in SDS PAGE running buffer (25 mM Tris (pH = 8), 192 mM glycine and 0.1% SDS). After electrophoresis, the proteins were transferred to a nitrocellulose membrane using a wet-transfer apparatus at 250 mA for 1.5 h with cold transfer buffer (25 mM Tris (pH = 8), 192 mM Glycine). Then, the nitrocellulose membrane was blocked with 5% skimmed milk in 1X PBST (0.05% Tween-20) for 1 h, followed by one PBST wash. The membrane was incubated in primary antibody (in 5% BSA or 1% skimmed milk) overnight at 4 °C. The blot was washed thrice with PBST at 5-min intervals, and then a secondary antibody (in 1% skimmed milk) conjugated with HRP was added and incubated at room temperature for 1 h. Further, the blots were washed thrice with PBST at 5-min intervals, and the blots were developed by Bio-Rad/Merck HRP substrate.

### Fluorescence recovery after photobleaching (FRAP)

FRAP experiments were performed in HeLa cells stably expressing AcGFP-NuMA[r], which were treated with DMSO, MLN8237 (30 min), MK5108 (30 min), or centrinone (2 or 4 days), as indicated. For FRAP shown in Fig. EV5G–J, endogenous NuMA was depleted, and cells were transiently transfected with either AcGFP-NuMA[r] or AcGFP-NuMA[r]$_{Q>G}$, followed by 17 h of STLC (7.5 μM) treatment to induce monopolar spindle poles before FRAP analysis. For all FRAP experiments, a pre-bleach image was acquired, followed by a brief pulse of 488 nm laser (40% laser intensity) to bleach the GFP signal from one of the two spindle poles at the metaphase or anaphase stage of the cell cycle, at an approximate area of 10–15 μm². Post-bleach images were taken every 5 s for a total of 50 cycles to monitor fluorescence recovery. To assess the fluorescence loss due to photobleaching, the intensities at the unbleached reference region (usually the other spindle pole) were simultaneously recorded. The intensity values were measured, corrected for the background, and the curves were then normalized using the following equation: (Phair and Misteli, 2000)

$$I = (I_t T_i)/(I_i T_t)$$

Here, $I_t$ is the intensity at the bleached ROI at a given time t, $I_i$ is the pre-bleach intensity at the bleached ROI, $T_i$ is the pre-bleach intensity at the unbleached region and $T_t$ is the intensity at the unbleached region at a given time $t$.

For the calculation of half-time of recovery ($T_{1/2}$) and the mobile fraction, full scale normalization of the FRAP intensities was done by the equation: $I(t) = (I_t - I_0)/(I_i - I_0)$ and the curves were fit in the exponential equation: $I(t) = I_0 + (I_r - I_0)(1-e^{-bt})$ (Raja et al, 2025) in OriginLab (https://www.originlab.com/). In the above equations, I(t) is the normalized intensity at a given time t, $I_t$ is the intensity at bleached ROI at a given time t, $I_i$ is the pre-bleach intensity, $I_0$ is the intensity immediately after photobleaching, $I_r$ is the maximum recovered fluorescence and b is the rate constant. The mobile fraction equals $I_r - I_0$, and the half-time is calculated by $T_{1/2} = \ln 2/b$.

### Photoconversion experiment

Photoconversion experiments were conducted in the metaphase stage of HeLa cells transiently transfected with mEOS-NuMA[r] or mEOS-NuMA[r]$_{S1969A}$ in cells depleted for endogenous NuMA. A circular ROI (~10 μm²) was drawn around a spindle pole and illuminated with a 405 nm laser set at a power of 5% for ~5 s. Following this, the images were acquired at 10 s intervals for 50 cycles. For analysis, line-scan measurements (3 μm width, 4 μm length for one spindle pole) were performed for both spindle poles using ImageJ in the mEOS (green) and mEOS* (red) channels. Fluorescence intensities were normalized by dividing the raw

intensity values by the maximum intensity value across all time points for each channel separately. The resulting normalized intensities were then plotted against distance for the selected timepoint.

### Quantification and statistical analysis

Fiji/ImageJ (https://fiji.sc), GraphPad Prism and Imaris (Bitplane Inc.) were used for all the quantifications. For intensity measurements, the "integrated density" measurement in arbitrary units (au) was used, which is defined as "(sum of pixel values in selection) x (area of one pixel)". All the quantifications in a given experiment were measured by a constant ROI, and the integrated density was measured from the maximum intensity projected images. The intensity values are plotted by normalizing with the cytoplasmic intensities and corrected for the background, as mentioned in the figure.

The level of significance between two mean values was calculated by a two-tailed unpaired Student's $t$-test. If the $P$ value is less than 0.05, it is considered to be statistically significant. It was calculated and confirmed using GraphPad Prism 8 (http://www.graphpad.com/scientific-software/prism/). The levels of significance are mentioned as n.s. (not significant) if $p \geq 0.05$, $*p < 0.05$, $**p < 0.01$, and $***p < 0.001$.

Circularity, solidity, and aspect ratio were also measured in ImageJ.

Spindle poles were considered dissolved when no residual NuMA signal was visually detectable at the spindle poles in time-lapse movies acquired using confocal microscopy. For quantitative analysis of the NuMA signal at the poles shown in Figs. 2A,B,G,H and EV2F–H, the non-saturated NuMA intensity in MLN8237-treated/or CycB-Aurora A expressing cells was used to capture images, and similar conditions were utilized to capture images of control cells before such analysis.

The chromatin bending angle (Fig. 7H) was measured 15 min from metaphase-to-anaphase transition in a single z-plane.

### Assigning time "0"

In almost all experiments, time, $t = 0$, is assigned as the metaphase-to-anaphase transition. This is the frame at which a proper metaphase plate is formed, leading to anaphase entry in the upcoming time frame.

### Optogenetic activation of the corelet system

HEK293 cells were transiently transfected with 2 µg each of iLid-GFP-FTH1 and IDR(s)-mCherry-SspB at 70% confluency plated on coverslips in 35 mm dish. Cells were imaged 24 h after plasmid transfection using the mCherry (561 nm) channel for the visualization of the behavior of IDR components. For global or local activation, the cells were illuminated with a 3–5% blue laser (488 nm) for 1–20 s in a circular area of 3000–4000 µm² and imaged for 2 min with 2 s intervals with a single z-plane. The puncta intensities were measured using ImageJ (ROI of area 0.893 µm²) by taking the ratios of fluorescent intensities at the puncta with respect to the cytoplasm, corrected for the background, and plotted.

### Sample preparation for 4C-seq experiments

For synchronization of HeLa Kyoto cells in the early G1 phase for 4C analysis, 24 h after plating the cells, they were synchronized in prometaphase with 100 nM nocodazole for 17 h. The synchronized cells were collected by mitotic shake-off and washed thrice with 1X

PBS to remove nocodazole. Cells were plated in media containing DMSO or MLN8237 for 1.5 h, followed by harvesting for 4C analysis. Briefly, $10–15 \times 10^6$ cells were collected, washed once with 1X PBS, and cross-linked using freshly prepared 2% formaldehyde solution by tumbling for 10 min at room temperature. The fixation was stopped by adding glycine and immediately transferring the tubes to ice. The cross-linked cells were collected at $300 \times g$ for 5 min at 4 °C and washed once with 1X PBS. After removing the supernatant, the pellet was snap-frozen in liquid nitrogen and stored at −80 °C.

### 4C-seq experiments and analysis

Chromatin fixation, cell lysis, and 4C-seq procedure were done as previously described using $10–15 \times 10^6$ cells per cell experiment (Matelot and Noordermeer, 2016). *MboI* (New England Biolabs) was used as the primary restriction enzyme and *NlaIII* (New England Biolabs) as the secondary restriction enzyme.

4C-seq sequencing libraries were generated using a two-step amplification approach with reduced amounts of input, as described in (Haarhuis et al, 2017; Krijger et al, 2020). For the rDNA viewpoint, which is present in the human genome in several hundred copies, a total of 120 ng of 4C material was amplified in a first step, using the Expand Long Template PCR System (Roche Diagnostics) in 12 reactions in parallel for 12 cycles (see Appendix Table S2 for rDNA viewpoint primer sequences). For the *FLT3* promoter viewpoint, which is a single-copy gene, a total of 900 ng of 4C material was amplified in a first step, using the Expand Long Template PCR System (Roche Diagnostics) in 12 reactions in parallel for 15 cycles (see Appendix Table S2 for *FLT3* viewpoint primer sequences). All reactions were pooled, followed by clean-up of 1/6th of the material using Agencourt Ampure XP Beads (Beckman Coulter) to remove fragments smaller than 200 bp. In the second step, 20% of the purified PCR products were further amplified in two reactions using the expand long template PCR system and universal index adapters for 20 cycles (see Appendix Table S2 for universal 4C-seq primer sequences). Amplified 4C-seq libraries were again purified with Agencourt Ampure XP Beads to remove fragments smaller than 200 bp. Quality and size distribution of the PCR libraries was verified by Qubit ds DNA BR kit (Thermo Fisher Scientific) and Tapestation Genomic DNA reagents (Agilent). Amplified 4C-seq libraries from the cells with or without MLN8237 treatment were mixed in an equimolar amount, followed by sequencing using 86 bp single-end reads on the Illumina Next-seq 550 device at the I2BC High-throughput sequencing facility (Gif-sur-Yvette, France).

4C-seq datasets were processed using the c4ctus pipeline (Miranda et al, 2022), available at https://github.com/NoordermeerLab/c4ctus]. Mapping was done on the T2T-CHM13 genome assembly (Nurk et al, 2022). For visualization, 4C-seq tracks were binned to 500 kb resolution, and then each bin was normalized to the total signal for each chromosome individually.

### Sequence analysis for protein disorder

IUPred program (https://iupred3.elte.hu/) was used to analyze the disordered regions of NuMA.

### Cloning of different constructs

Different truncated constructs of NuMA (AcGFP-NuMA$^r_{(1–1699)}$, AcGFP-NuMA$^r_{\Delta DBD}$, AcGFP-NuMA$^r_{(1700–2115)}$) were generated by

amplifying the regions (Fig. 4A) from AcGFP-NuMA$^r$ by respective primer pairs and subcloning them with AgeI and NotI restriction sites inside the pIRES-AcGFP-FLAG vector.

AcGFP-NuMA$^r_{NC}$ was generated by sequential cloning where initially, amino acids 1–705 of NuMA were subcloned using AgeI and EcoRI sites. After that, amino acids 1700–2115 were subcloned using EcoRI and NotI sites inside the pIRES-AcGFP-FLAG vector. As a result of sequential cloning, the EF linker was incorporated at the junction of these two fragments.

AcGFP-NuMA$^r_{(Control),(R>G),(Q>G),(Aro>A)}$ constructs were also made by sequential cloning, where in the first round 1–1699 region of NuMA was subcloned using AgeI and EcoRI sites and in the next round 1700–2115 regions of control and mutated NuMA (R > G, Q > G, Aro>A) were subcloned using EcoRI and BamHI sites inside pIRES-AcGFP-FLAG vector. AcGFP-NuMA$^r_{\Delta NLS}$ was also cloned in a similar strategy, where the 1700–2115 region without the NLS region was amplified by overlap extension PCR using overlapping primer pairs mentioned in the primer list, followed by sequential cloning with EcoRI and BamHI restriction sites. While generating the AcGFP-NuMA$^r_{(S1969A;Q>G)}$ construct, the process was similar to earlier, but the mutation within the 1700–2115 Q > G region was incorporated by PCR using a primer bearing altered bases at the 1969 position. The mutated 1700–2115 fragments were synthesized by Twist Bioscience (https://www.twistbioscience.com/) and were subcloned by amplifying with proper primer pairs. AurA$^r$-AcGFP construct was generated by amplifying AurA$^r$-AcGFP region from CycB-AurA$^r$-AcGFP plasmid, followed by subcloning it into the same plasmid using EcoRV and NotI enzyme sites.

While generating the optogenetic constructs of NuMA$^S_{C-ter}$, a common vector system was essential, so we made an in-house vector by subcloning mCherry-SspB using AgeI and NotI sites inside the H2B-mCherry encoding plasmid. In this vector, H2B was substituted with NuMA$_{C-ter}$ using XhoI and AgeI sites, whereas the NuMA$_{1-2057}$ was generated by substituting H2B with the NuMA$_{(1-2057)}$ region using EcoRI and AgeI sites. In this vector, H2B was substituted with NuMA$_{C-ter}$ using XhoI and AgeI sites. Other mutated constructs were made using a similar strategy. HNRNPA1c-mCherry-SspB construct was generated by amplifying HNRNPA1c-mCherry-SspB from the addgene plasmid #122668 and subcloning it into the Addgene plasmid #21044 using NheI and NotI sites.

The information on all the plasmids and primers utilized in this study can be found in Appendix Tables S1, S2, respectively.

## Data availability

The unprocessed and processed 4C-seq data from this publication have been deposited in the GEO repository (https://www.ncbi.nlm.nih.gov/geo/) with the identifier GSE277941.

The source data of this paper are collected in the following database record: biostudies:S-SCDT-10_1038-S44318-025-00564-4.

## Peer review information

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

## Acknowledgements

We thank Arshad Desai, Jonathon Pines, Martin Lowe, Paul Guichard, Virginie Hamel, Gautam Dey, K Subramanian, Sukriti Kapoor, Umesh Varshney, and all the members of the Kotak lab for their discussion and suggestions. We thank Daniel Gerlich, Mark Petronczki, Deepak Nair, Amit Singh, and Anthony Hyman (MPI-CBG, Dresden) for providing plasmids and cell lines. We thank Srividya Sana, Riya Keshri, Arka Ghosh, and Sukriti Kapoor for their initial contribution to this project. We acknowledge the help of Sukriti Kapoor with the working model. We acknowledge the help of DST-FIST, and IISc for infrastructure support. We acknowledge the sequencing and bioinformatics expertise of the I2BC High-throughput sequencing facility, supported by France Génomique (funded by the French National Program "Investissement d'Avenir" ANR-10-INBS-09). This work is supported by the Institute of Eminence fund (IE/REDA-23-2061, IE/REDA-24-2061, IE/REDA-25-2061), the DBT grant (BT/PR36084/BRB/10/1857/2020), the DST-SERB grant (CRG/2022/005151) to S. Kotak, and Indo-French Centre for the Promotion of Advanced Research (CEFIPRA) grant (7103-4) to S. Kotak and D. Noordermeer.

## Author contributions

**Ashwathi Rajeevan**: Data curation; Formal analysis; Validation; Investigation; Visualization; Methodology. **Vignesh Olakkal**: Data curation; Formal analysis; Supervision; Validation; Investigation; Visualization; Methodology; Writing—review and editing. **Madhumitha Balakrishnan**: Data curation; Formal analysis; Validation; Investigation; Visualization; Methodology; Writing—review and editing. **Dwaipayan Chakrabarty**: Data curation; Formal analysis; Validation; Investigation; Visualization; Methodology; Writing—review and editing. **François Charon**: Formal analysis; Validation; Investigation; Methodology. **Daan Noordermeer**: Data curation; Formal analysis; Supervision; Funding acquisition; Validation; Investigation; Visualization; Methodology; Project administration; Writing—review and editing. **Sachin Kotak**: Conceptualization; Resources; Data curation; Formal analysis; Supervision; Funding acquisition; Validation; Investigation; Visualization; Methodology; Writing—original draft; Project administration; Writing—review and editing.

Source data underlying figure panels in this paper may have individual authorship assigned. Where available, figure panel/source data authorship is listed in the following database record: biostudies:S-SCDT-10_1038-S44318-025-00564-4.

## Disclosure and competing interests statement

The authors declare no competing interests.

# Expanded View Figures

**Figure EV1. Anaphase-specific Aurora A degradation tool to study Aurora A function at mitotic exit.**

(A) Immunofluorescence (IF) analysis of anaphase cells stained with phospho-specific anti-Aurora A antibody against T288 (T288$^P$; green). DNA is shown in blue. (B) Schematic of the method to quantify the T288$^P$ enrichment (in au) at poles in cells stained with anti-T288$^P$ antibody during anaphase. In this and subsequent Fig. panels bkgd., representing background intensity. Note that T288$^P$ is significantly enriched at the poles in anaphase, indicating the presence of active Aurora A pool at the poles during anaphase. Error bars: mean ± SD. (C) Immunoblot analysis of protein extracts prepared from mitotically synchronized control HeLa or HeLa cells stably expressing CycB-AcGFP. Extracts were probed with antibodies directed against CyclinB and β-actin. Endogenous CyclinB and CycB-AcGFP bands are indicated. In this and other immunoblot panels, the molecular mass is indicated in kilodaltons (kDa) on the left. (D) Schematic of the method and the quantification of the chromosomal intensity of CycB-AcGFP (in au) during the metaphase-to-anaphase transition (time 0). The graph illustrates the mean, and the shaded region indicates ± SEM. For representative images, see 1 G. (E) IF analysis of control and monoclonal cell line expressing CycB-Aurora A$^r$-AcGFP. These cells were stained with anti-Aurora A (red; endogenous protein detection) and anti-GFP (green; exogenous protein detection) antibodies upon transfection with control and Aurora A siRNA for 60 h. DNA is shown in blue. (F) Quantification of the fraction of mitotic cells (mitotic index) in control cells and monoclonal cell line expressing CycB-Aurora A$^r$-AcGFP. Cells were analyzed by IF 60 h after transfection with control and Aurora A siRNA. Error bars: mean ± SD from two independent experiments ($n > 1000$ cells each). Exact $p$ values from left to right are **$p = 0.0049$, $p = 0.4392$ (ns). (G) Schematic representation of chromosome instability defects and the quantification of such defects in control cells and monoclonal cell line expressing CycB-Aurora A$^r$-AcGFP upon transfection with control and Aurora A siRNA for 60 h, as indicated. Error bars: mean ± SD from two independent experiments ($n > 120$ cells each). Exact $p$ value is ***$p < 0.001$. (H) Schematic representation and the estimation of chromosome separation kinetics of monoclonal cell line expressing CycB-Aurora A$^r$-AcGFP in control or upon transfection with control and Aurora A siRNA for 60 h. Values represent mean ± SD. Exact $p$ values from left to right are ***$p < 0.001$, ***$p < 0.001$, ***$p < 0.001$. (I) Confocal live-cell imaging of HeLa cells stably expressing AuroraA$^r$-AcGFP (green) and probed for silicon-rhodamine DNA (SiR-DNA; magenta) to visualize chromosomes ensemble and nuclear shape post-mitosis in control and upon transfection with Aurora A siRNA for 60 h. Also, see nuclear shape analysis of cells expressing CycB-AuroraA$^r$-AcGFP in control and upon Aurora A depletion in Fig. 1I–K. (J, K) Nuclear shape analysis [circularity (J) and solidity (K)] from the confocal live-cell imaging of cells, as mentioned in (I). Exact $p$ values are $p = 0.4962$ (ns) (J) and $p = 0.0813$ (ns) (K). $p$ values are denoted as follows: ns-$p \geq 0.05$; **$p < 0.01$; ***$p < 0.001$ as determined by two-tailed unpaired Student's $t$-test. Scale bars in (A, E, I) represent 10 μm.

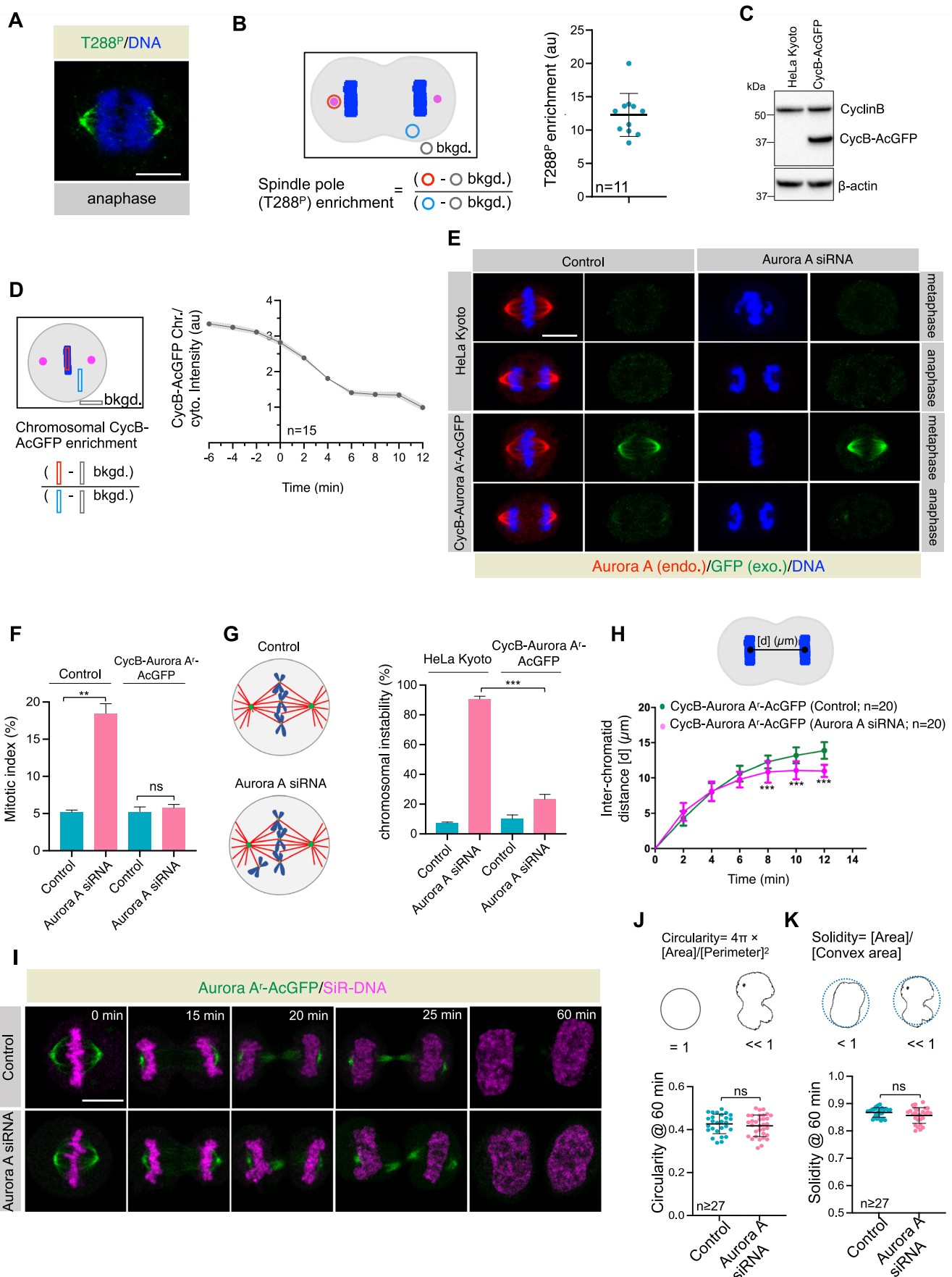

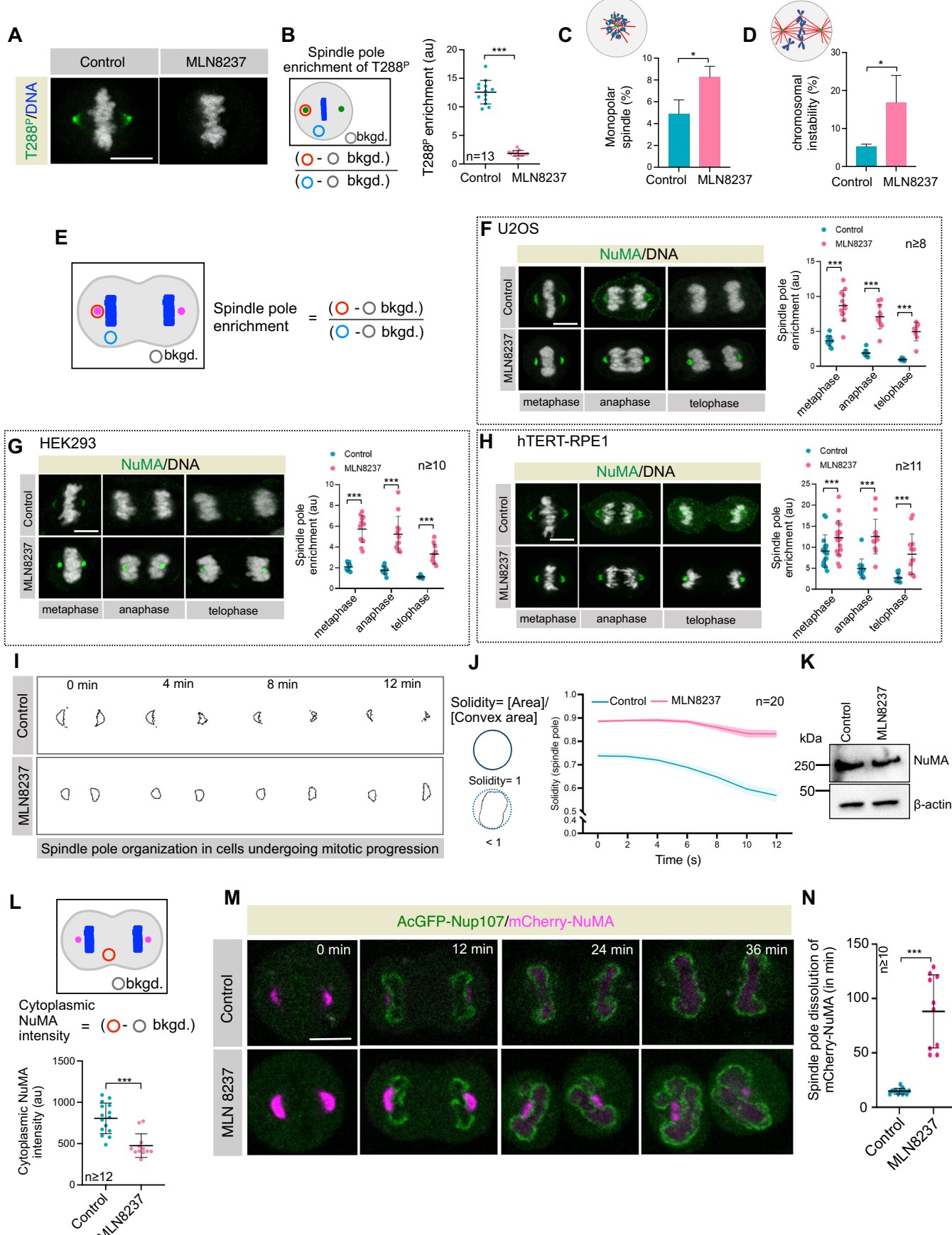

◀ **Figure EV2. NuMA is abnormally accumulated at the poles upon acute Aurora A inhibition at the mitotic exit.**

(A) IF analysis of HeLa cells during metaphase in control and upon acute Aurora A inhibition (2 h) with MLN8237 (50 nM). Cells were stained for T288$^P$ (green). DNA is shown in gray. (B) Schematic representation of the quantification method and the quantification of T288$^P$ at poles in cells stained with anti-T288$^P$ antibody in metaphase. Note that T288$^P$ enrichment is significantly affected in cells treated with Aurora A inhibitor MLN8237. Error bars: mean ± SD. Exact $p$ value is ***$p < 0.001$. (C, D) Quantification of the fraction of monopolar spindle (C), and chromosomal instability defects (D) in control and upon acute Aurora A inhibition (2 h) with MLN8237 (50 nM). Error bars: mean ± SD from three independent experiments ($n > 300$ cells each). Note that cells without any sign of chromosome instability were utilized for further analysis. Exact $p$ values are *$p = 0.0113$ (C) and *$p = 0.0266$ (D). (E–H) Schematic of the method to quantify spindle pole enrichment of NuMA (in au) (E) and the outcome of such analysis by performing IF in various cell lines such as U2OS (F), HEK293 (G), and hTERT-RPE1 (H) in control and upon acute Aurora A inhibition with MLN8237 (50 nM) for 2 h. For this analysis, cells were stained with anti-NuMA (green) antibody. DNA is shown in gray. The quantification on the right represents mean ± SD. Exact $p$ values from left to right in (F–H) are ***$p < 0.001$, ***$p < 0.001$, ***$p < 0.001$. (I, J) The representative geometry of NuMA organization at the pole in control and MLN8237-treated HeLa cells. Timepoint $t = 0$ min was set to the metaphase-to-anaphase transition (I). The measurement of the solidity of NuMA-based poles in control and MLN8237-treated cells is shown in (J). Curves, and shaded areas indicate mean ± SEM. The p-values for the solidity of NuMA-based poles for all the time points is <0.001. (K) Immunoblot analysis of protein extracts prepared from anaphase-synchronized control and MLN8237-treated cells. Extracts were probed with anti-NuMA and anti-β-actin antibodies. (L) Schematic of the method to quantify cytoplasmic NuMA intensity during anaphase and the outcome of such analysis (in au). Bars indicate mean ± SD. Exact $p$ value is ***$p < 0.001$. (M, N) Confocal live-cell imaging of HeLa cells stably expressing nucleoporin marker AcGFP-Nup107 (green) and were transiently transfected with mCherry-NuMA (magenta) in the absence (control) or upon acute inhibition of Aurora A kinase using MLN8237 (M). Timepoint $t = 0$ min was set to the metaphase-to-anaphase transition. Quantification of the dissolution time (in min) of mCherry-NuMA at the poles in control and MLN8237-treated cells with respect to (w.r.t.) metaphase-to-anaphase transition (N). Error bars: mean ± SD. Exact $p$ value is ***$p < 0.001$. $p$ values are denoted as follows: *$p < 0.05$; ***$p < 0.001$ as determined by two-tailed unpaired Student's $t$-test. Scale bars in (A, F, G, H, M) represent 10 μm.

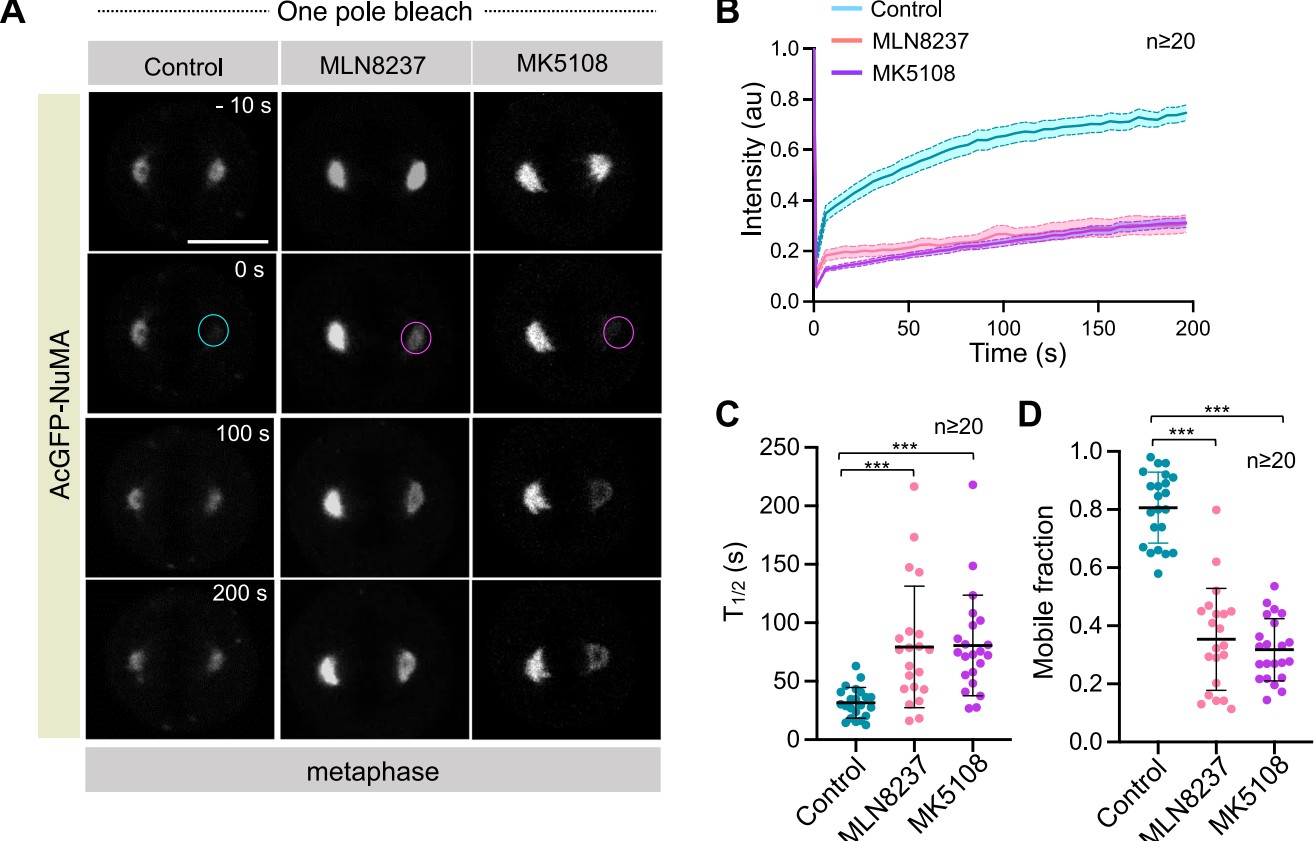

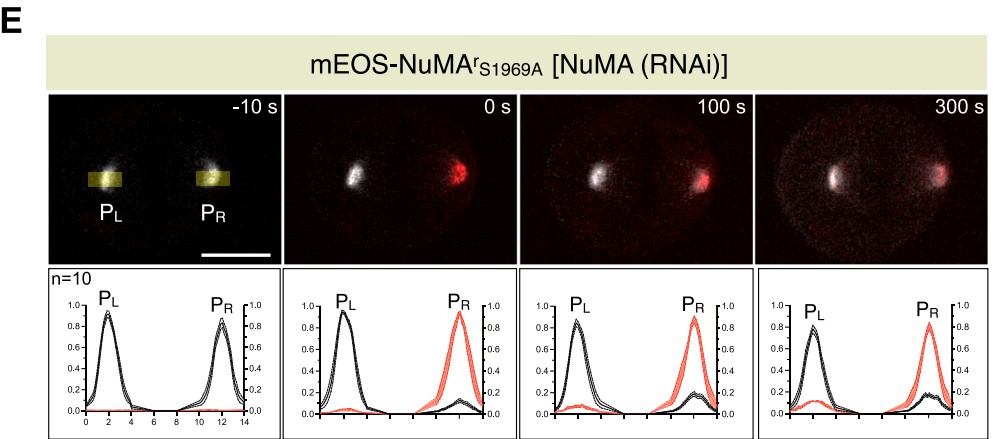

◀ **Figure EV3. NuMA changes its material properties from dynamic-to-solid upon Aurora A inhibition.**

(A) FRAP analysis of metaphase cells stably expressing AcGFP-NuMA (gray) that are either left untreated or treated with specific Aurora A inhibitors — MLN8237 or MK5108, as indicated. Time is indicated in seconds (s). Blue and magenta circles show the bleached regions of control and MLN8237 or MK5108-treated cells, respectively. (B) The AcGFP recovery profile of the bleached area is plotted for 200 s for control and MLN8237 or MK5108-treated cells. Curves and shaded areas indicate mean ± SEM. Note the remarkably slow recovery of pole-localized AcGFP-NuMA signal in MLN8237 or MK5108-treated cells. (C, D) The half-time of recovery [$T_{1/2}$] (C) and the mobile fraction (D) of untreated and MLN8237 or MK5108-treated metaphase cells. Error bars: mean ± SD. See Methods for details. Exact *p* values from left to right are \*\*\**p* < 0.001, \*\*\**p* < 0.001 for (C) and \*\*\**p* < 0.001, \*\*\**p* < 0.001 for (D). See Methods for details. (E) One-pole photoconversion and the dynamics of photoconverted mEOS-NuMA$^r_{(S1969A)}$ signal in HeLa cells depleted for endogenous NuMA after 72 h of transfection with NuMA siRNA. Timepoint "0" indicates the time when mEOS-NuMA$^r_{S1969A}$ was photoconverted at one pole. A line-scan with a line thickness of 3 µm (roughly similar to the size of the poles) was performed for the unperturbed left pole ($P_L$) or perturbed right pole ($P_R$). Ten cells were analyzed (see Methods). Error bars: mean ± SEM. *p* values are denoted as \*\*\**p* < 0.001 as determined by two-tailed unpaired Student's *t*-test. Scale bars in (A, E) represent 10 µm.

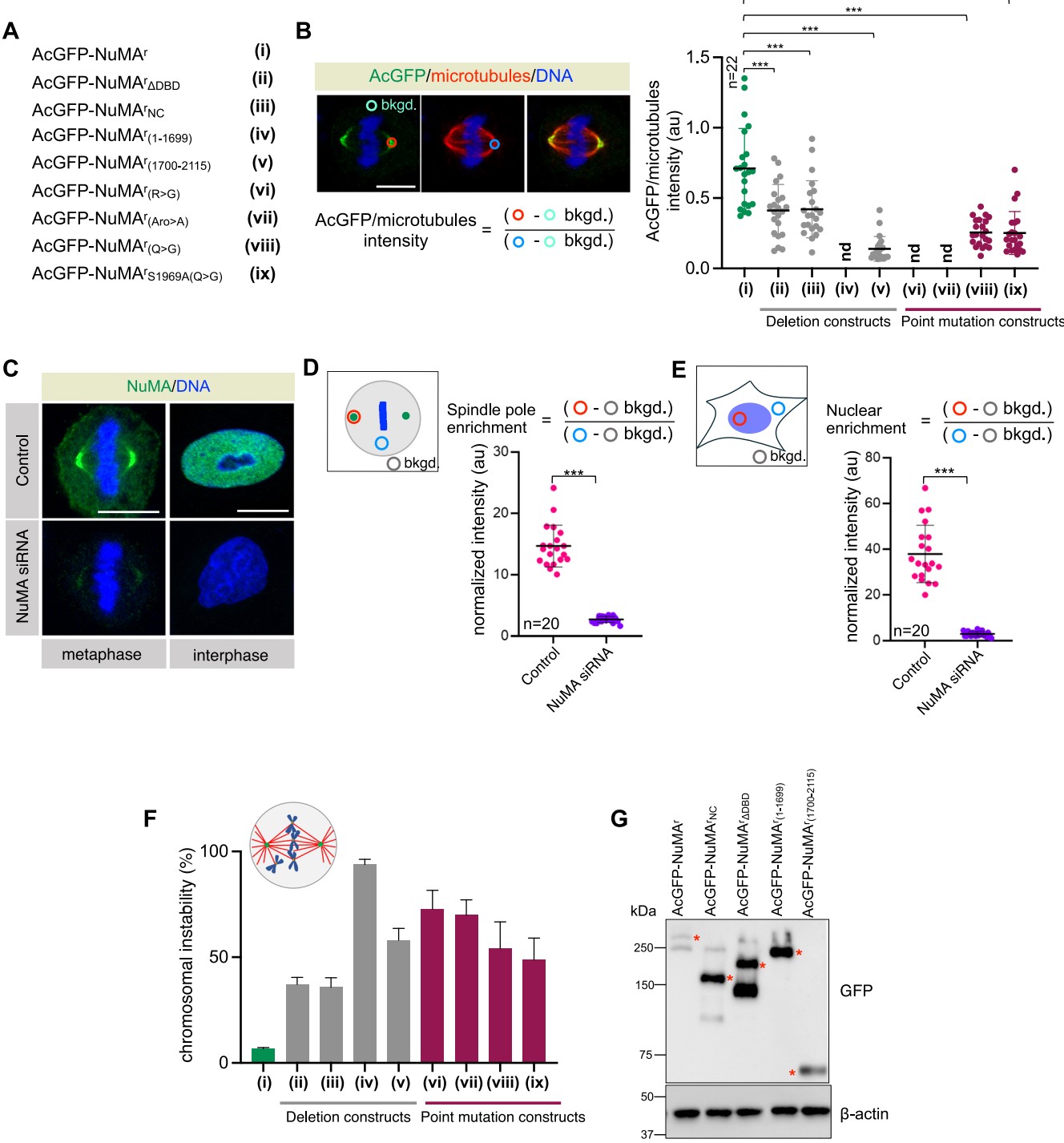

**Figure EV4.    Expression of various NuMA mutant or deletion constructs lead to significant chromosomal instability.**

(**A**) Nomenclature of siRNA-resistant wild-type full-length NuMA (**i**) and various deletion and mutant constructs (**ii–ix**), as indicated. See Figs. 4, 5, for detail. (**B**) Schematic representation of the method, and the quantification of spindle pole intensity in terms of AcGFP/microtubules intensity (in au) in HeLa cells expressing either wild-type (**i**) or deletion/mutant NuMA constructs (**ii–ix**), as mentioned in (**A**), 72 h after transfections with NuMA siRNA. These cells were stained with anti-GFP (green), and anti-α-tubulin (red) antibodies. The DNA is shown in blue. Please note, that (**iv**), (**vi**), and (**vii**) fail to localize at the poles, and therefore their ratio with the microtubules cannot be determined (nd). Exact $p$ values from left to right are ***$p < 0.001$, ***$p < 0.001$, ***$p < 0.001$, ***$p < 0.001$, ***$p < 0.001$. (**C–E**) IF analysis to assess the efficiency of NuMA depletion after 72 h of transfection with NuMA siRNA. IF analysis shows NuMA localization during mitosis and in interphase (**C**) and NuMA quantification at the poles (**D**) and in the nucleus (**E**). NuMA was stained using anti-NuMA antibodies (green), and DNA is shown in blue. Error bars: mean ± SD. Exact $p$ values are ***$p < 0.001$ (**D**) and ***$p < 0.001$ (**E**). (**F**) Quantification of the fraction of chromosomal instability in HeLa cells expressing either wild-type (**i**) or deletion/mutant NuMA constructs (**ii–ix**; see **A**), 72 h after transfections with NuMA siRNA. Error bars represent mean ± SD of two experiments ($n > 100$ cells each). Note a direct correlation between failure in pole localization by these mutants and an increase in chromosomal instability. Also note that the ability of these cells to organize the quasi-bipolar spindle could be due to the residual endogenous NuMA levels in our experimental conditions. (**G**) Immunoblot analysis of protein extracts prepared from prometaphase synchronized cells expressing either wild-type or various deletion constructs of NuMA tagged with AcGFP, as indicated. Extracts were probed with anti-GFP and anti-β-actin antibodies. Asterisk represents the correct size of the AcGFP-tagged protein. $p$ values are denoted as ***$p < 0.001$ as determined by two-tailed unpaired Student's $t$-test. Scale bars in (**B**, **C**) represent 10 µm.

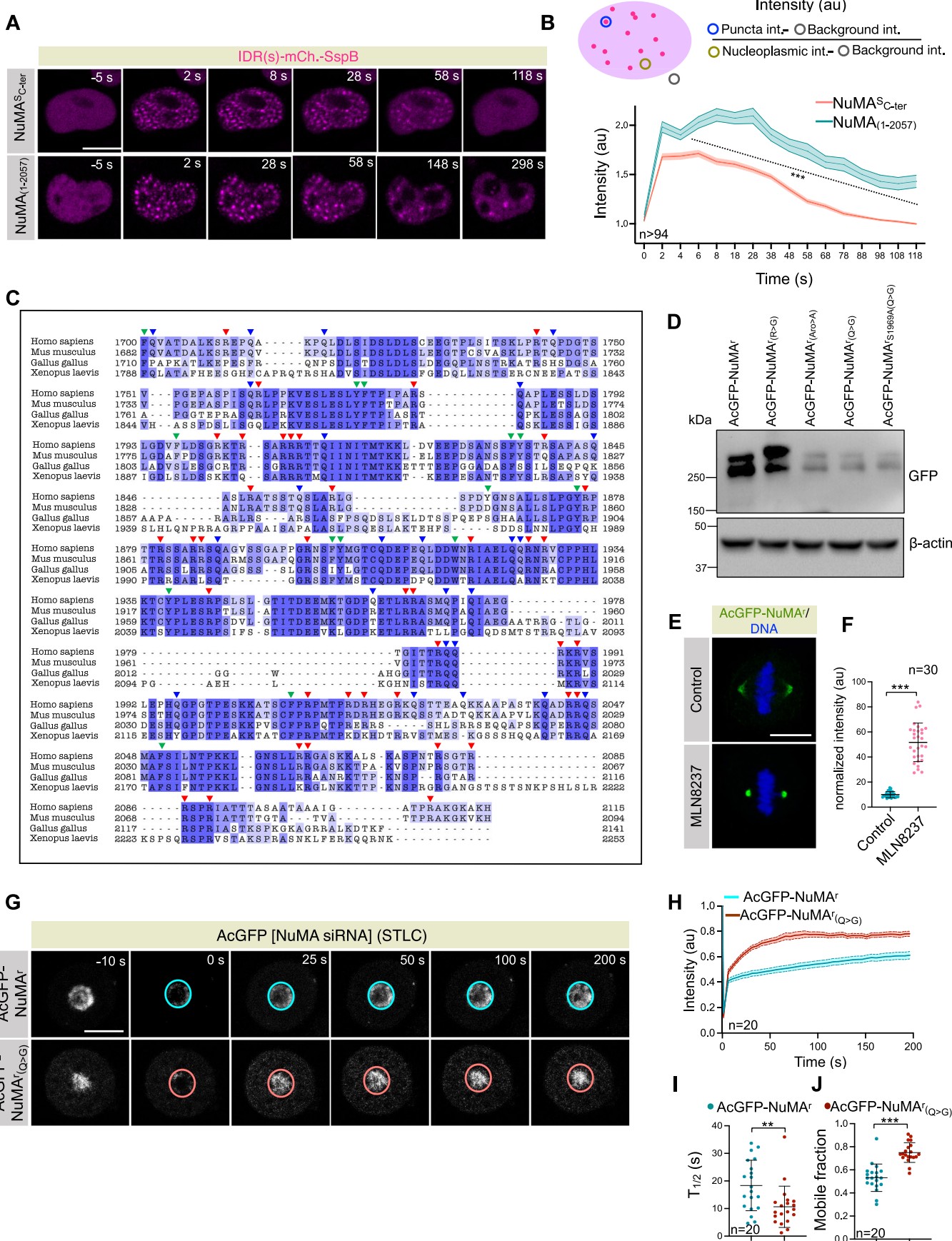

**Figure EV5. Glutamine residues in the NuMA C-terminus provide hardening to the poles.**

(A) Confocal live-cell imaging of Corelet-expressing HEK293 cells with wild-type NuMA$^S_{C-ter}$ or full-length NuMA lacking its chromatin binding domain [NuMA$_{(1-2057)}$]. Note that the condensates of NuMA$_{(1-2057)}$ are significantly slower in dissolving compared to NuMA$^S_{C-ter}$ IDR. Please note the re-use of the image (Fig. 5C) revealing NuMA$^S_{C-ter}$ expression. (B) Schematic representation of the quantification method, intensity (in au), and the dynamics of Corelet-based condensates —NuMA$^S_{C-ter}$, or NuMA$_{(1-2057)}$. Curves and shaded areas indicate mean ± SEM. As indicated the behavior of more than 94 clusters were analysed from a minimum of 20 cells in each condition. (C) Sequence alignment of the NuMA C-terminus (1700–2115 aa) of different NuMA orthologs. Red, green, and blue arrowheads represent arginine (R), aromatic (Aro: W, Y, F), and glutamine (Q) residues, respectively. (D) Immunoblot analysis of protein extracts prepared from prometaphase synchronized cells expressing either wild-type or various AcGFP-tagged NuMA mutants, as indicated. Extracts were probed with anti-GFP and anti-β-actin antibodies. Since NuMA is highly phosphorylated in mitosis, the two bands detected by anti-GFP antibodies could be phosphorylated and non-phosphorylated NuMA bands. (E, F) Representative images of HeLa cells stably expressed low levels of siRNA resistant AcGFP-tagged NuMA (in green) that were depleted for endogenous NuMA by siRNA. These cells are either treated with DMSO (control) or treated with MLN8237, as indicated (E). DNA is shown in blue. Normalized spindle pole intensity (in au) on the right of AcGFP-NuMA$^r$ at the poles in control and MLN8237-treated cells (F). Error bars ± SD. Exact $p$ value is ***$p < 0.001$. (G) FRAP analysis of HeLa cells which are either expressing AcGFP-NuMA$^r$ (gray) or AcGFP-NuMA$^r_{(Q>G)}$ upon endogenous protein depletion for 72 h. These cells were treated with $S$-trityl-ʟ-cysteine (STLC)—kinesin 5 inhibitor for 17 h before imaging to block bipolar spindle assembly. Time is indicated in seconds (s). Blue and orange circles show the bleached regions of AcGFP-NuMA$^r$ and AcGFP-NuMA$^r_{(Q>G)}$ expressing cells, respectively. Note the significantly faster recovery of AcGFP signal in cells expressing AcGFP-NuMA$^r_{(Q>G)}$. (H) The AcGFP recovery profile of the bleached area corrected for photobleaching is plotted for 200 s for AcGFP-NuMA$^r$ and AcGFP-NuMA$^r_{(Q>G)}$ expressing cells. Curves and shaded areas indicate mean ± SEM. (I, J) The half-time of recovery [T$_{1/2}$] in (s) (I) and the mobile fraction (J) of AcGFP-NuMA$^r$ and AcGFP-NuMA$^r_{(Q>G)}$ expressing cells, as indicated. Error bars: mean ± SD. Exact $p$ values are **$p < 0.0044$ (I) and ***$p < 0.001$ (J). $p$ values are denoted as follows: **$p < 0.01$; ***$p < 0.001$ as determined by two-tailed unpaired Student's $t$-test. Scale bars in (A, E, G) represent 10 μm.

