## [Peer Review File · The EMBO Journal]

Aurora A regulates the material property of spindle poles to orchestrate nuclear organization at mitotic exit

Ashwathi Rajeevan, Vignesh Olakkal, Madhumitha Balakrishnan, Dwaipayan Chakrabarty, François Charon, Daan Noordermeer, and Sachin Kotak

Corresponding author(s): Sachin Kotak (sachinkotak@iisc.ac.in)

Review Timeline:

Submission Date:	7th Mar 25
Editorial Decision:	7th Apr 25
Revision Received:	7th Jul 25
Editorial Decision:	8th Aug 25
Revision Received:	14th Aug 25
Accepted:	26th Aug 25

Editor: Hartmut Vodermaier

Transaction Report:

Prof. Sachin Kotak
Indian Institute of Science Bangalore
Microbiology and Cell Biology
C V Raman Avenue
Bangalore, Karnataka 560012
India

7th Apr 2025

Re: EMBOJ-2025-120705
Phase transition of spindle pole localized protein orchestrates nuclear organization at mitotic exit

Dear Sachin,

Thank you again for submitting your study on NuMA spindle pole dynamics at late mitotic stages to The EMBO Journal. We have now received reports from three expert referees, copied below for your information. As you will see, all referees clearly appreciate the importance of the subject, and the potential interest of your findings and model. However, they at the same time feel that several key conclusions would need to be strengthened with additional evidence, and raise a number of presentational concerns as well as various other specific issues to be clarified before the study may become suitable for publication.

Should you be able to comprehensively address the referees' points, we would be happy to pursue a revised version further for The EMBO Journal. Please be reminded, however, that our single-major-revision-round policy makes it important to diligently respond to each referee point at the time of resubmission; therefore, please do not hesitate to contact me well ahead of resubmission with any questions you may have in this regard. We would also be open to extension of the regular three-months revision period if needed; our 'scooping protection' (meaning that competing work appearing elsewhere in the meantime will not affect our considerations of your study) would of course remain valid also throughout such an extension.

Further information on preparing, formatting and uploading a revised manuscript can be found below and in our Guide to Authors. Thank you again for the opportunity to consider this work for The EMBO Journal, and I look forward to receiving your revised manuscript in due time.

With kind regards,

Hartmut

9) To facilitate reproducibility and cross-laboratory adoption of methodologies, please structure the Materials & Methods section as outlined in our guide to authors, including a completed Reagents and Tools Table that can be downloaded from our author guidelines as well (<https://www.embopress.org/page/journal/14602075/authorguide#structuredmethods>).

10) Digital image enhancement is acceptable practice, as long as it accurately represents the original data and conforms to community standards. If a figure has been subjected to significant electronic manipulation, this must be clearly noted in the figure legend and/or the 'Materials and Methods' section. The editors reserve the right to request original versions of figures and the original images that were used to assemble the figure. Finally, we generally encourage uploading of numerical as well as gel/blot image source data; for details see: embopress.org/page/journal/14602075/authorguide#sourcedata

At EMBO Press, we ask authors to provide source data for the main manuscript figures. Our source data coordinator will contact you to discuss which figure panels we would need source data for and will also provide you with helpful tips on how to upload and organize the files.

In the interest of ensuring the conceptual advance provided by the work, we recommend submitting a revision within 3 months (6th Jul 2025). Please discuss the revision progress ahead of this time with the editor if you require more time to complete the revisions. Use the link below to submit your revision:

Link Not Available

Referee #1:

Phase transition of spindle pole localized protein orchestrates nuclear organization at mitotic exit

In this study, Rajeevan et al. investigate the organization of mitotic spindle poles during late mitosis, focusing on how NuMA pole localization is resolved and coordinated with nuclear envelope reformation. The authors begin by analyzing mitotic exit in the presence of the Aurora A inhibitor MLN8237 and observe that its inhibition leads to the formation of misshapen nuclei. To specifically examine the late mitotic function of Aurora A, the authors ingeniously constructed a version of Aurora A fused to the D-box of Cyclin B. This design allows Aurora A to be degraded at the metaphase-anaphase transition via the ubiquitin/proteasome system while still fulfilling its early mitotic functions. Notably, a significant number of these cells exhibit aberrantly shaped nuclei, with chromosomes wrapped around the spindle poles-suggesting a critical late mitotic role for Aurora A kinase.

Interestingly, the authors find that NuMA localization at spindle poles persists longer than usual during mitotic exit when Aurora A is inhibited. They also note that Aurora A inhibition increases NuMA retention at spindle poles in metaphase. This regulation is likely mediated through phosphorylation at the S1969 site of NuMA. To further explore this, the authors analyze the dynamic properties of NuMA at spindle poles in both control and MLN8237-treated cells, concluding that NuMA is less mobile in the absence of Aurora A activity (see my comment below). This is a crucial experiment, and I would recommend moving the

quantifications from Fig. S5C,D into the main Figure 3 for greater emphasis.

In the final part of the manuscript, the authors examine Aurora A-guided NuMA phase transition at spindle poles. They highlight that previous studies on NuMA phase transition relied on artificial conditions, such as crowding agents, whereas their work demonstrates that microtubules are necessary for the accumulation of NuMA into larger assemblies at the spindle poles. Using the Corelet system, they investigate the intrinsically disordered region (IDR) of NuMA and its role in homotypic interactions, showing that residues 1700-2057 are sufficient for forming light-induced condensates. Further analysis reveals that arginine and aromatic amino acids within the IDR are crucial for these interactions, and mutations in these residues disrupt NuMA localization at spindle poles in mitotic cells. Moreover, they conclude that glutamine residues within the IDR contribute to the hardening of NuMA assemblies in the absence of Aurora A activity.

To further probe this mechanism, the authors increase NuMA multivalency using Kaede to determine whether this mimics the effects of Aurora A inhibition. Additionally, in a well-designed experiment, they introduce Aurora A asymmetry at spindle poles and assess its impact on NuMA dynamics. Their findings demonstrate that NuMA behaves differently on each spindle pole depending on the presence or absence of active Aurora A kinase. Finally, the authors explore whether Aurora A-dependent spindle pole regulation influences interphase genome organization (see point 1 below).

Overall, I find this manuscript interesting as it sheds light on a largely overlooked aspect of mitosis—the disassembly of spindle poles at the end of mitosis. When cells enter mitosis, NuMA relocates from chromatin to spindle poles under the regulation of CDK1. From prometaphase until the end of mitosis, Aurora A controls the amount of NuMA at spindle poles. As mitosis progresses, decreasing CDK1 activity and increasing PP1/PP2A activity likely dephosphorylate CDK1-phosphorylated NuMA, allowing it together with Aurora A activity to transition from the poles back to chromatin. If this process is disrupted, nuclear and chromatin organization is compromised. Given this, I am surprised that CDK1 is not discussed in more detail in the manuscript's Discussion section.

Major points

1. The authors have shown in Rajeevan et al. (MBoC 2020) that NuMA interacts with chromatin in interphase and prophase via its C-terminus. Expression of a DNA-binding-deficient mutant of NuMA affects chromatin decondensation at the mitotic exit and nuclear shape in interphase. Is it possible that AuroraA inhibition impacts NuMA binding/loading to chromatin? In this case, the nuclear shape phenotype and the defects described in Fig. 7 would not be (or only partly) caused by NuMA at the spindle poles. The authors should discuss this possibility.
2. p. 7: "Intriguingly, a significant number of CycB-AuroraA-AcGFP expressing cells" What is this number? It is not given in Fig. 1J or K.
3. Fig. 1I: as control, it would be better to express an siRNA resistant AuroraA.
4. Fig. 3B: the pole of the control spindle is bleached and the signal recovers to ca. 30% within 3-5 min and then remains stable (blue line). This happens in all 14 cells. Why is there no further recovery? Why is NuMA at spindle poles in control cells during anaphase (see Fig. 2G)? I assume that the authors picked relatively rare anaphase cells with NuMA at spindle poles - please indicate this in the text.
5. Fig. S5E : I do not see recovery in the Control and the MLN8237 cells. How was the intensity quantified in Fig. S5F? It basically starts at 0.0. I guess that only part of the spindle pole was quantified. Please explain.
6. Fig. S5C, G: the spread of the data points is quite high in the MLN8237 cells - why? Number of MLN8237 should be increased. Since it is a key experiment, the 2nd Aurora A inhibitor should be used to confirm data.
7. Fig. 3E is a nice experimental set up. However, the line scan is risky because a 3D object is quantified by a 1D analysis. I suggest measuring the entire pole signal. In addition, spindle pole localization of NuMA in control and MLN8237 cells is quite different (more extended in the Control - why? Should be less). More than 30 cells were analyzed. The authors should find a way to summarize the behavior of these cells in one graph.
8. Fig. 3F: Why does NuMA have a similar signal intensity at spindle poles in control anaphase and MLN8237 treated cells? Is it possible to perform time resolved 3D-SIM to see how the fibers change over time?
9. Do mutations that were introduced in Fig. 4A affect MT binding?
10. Measuring the dynamic properties of Q>G (Fig. 5) by FRAP would be an important experiment.
11. Fig. 5G (Ac-GFP-NuMA) looks different to Fig. 2A (antibodies, no tag). Does this mean that the tag impacts NuMA at spindle poles? This would be concerning. Y-axis in Fig. 5G? Was the entire poly signal measured or only a circular area of ? nm?
12. Fig. 6F needs FRAP analysis (Sas-6-mCherry as centrosome marker and Ac-GFP-NuMA). Comment on line scan analysis see 7.

Minor points

1. Fig. 1B-D: y-axis of the graphs needs description.
2. p. 8: As similar observation was made in other cell lines - please name them in the result part.
3. Fig. 2E: the line scan intensity does not fit with the images above. For example, control at 9 min signal intensity is max and as high as after 15 min although in the image the mCherry-NuMA signal after 15 min is much lower than after 9 min.
4. p. 10: "Additionally, we show that AuroraA-mediated phosphorylation at S1969 of NuMA dynamically regulates pole-localized NuMA...." This was not shown!
5. p. 10: "massively accumulates" suggests a recruitment mechanism. However, it just stays and is not disassembled.
6. p. 10: NuMA is not a PCM component. Thus, other PCM localized proteins is incorrect. MTs "remained unaltered at the poles

- in AuroraA inhibited anaphase cells". Unaltered compared to wild type or metaphase...? Be more specific.
7. Fig. S4 - remove (A - only one pannel). Would be nice to do double staining with NuMA.
 8. Fig. 3F: the order of the staining in the enlargements should be the same: NuMA/PCNT/Merge.
 9. p. 13: (>1.5 mM) - in figure S6: 1.5 μ M (this is the correct value).
 10. Authors have to check expression levels by IB of the constructs in Fig. 4.
 11. The authors should indicate how many R>G, Aro>A, Q>G mutations were introduced in the IDR. 39 R? 14 Y/F/W?
 12. p. 13: telophase
 13. p.14: NuMA C-terminus is predicated to be largely disordered ...
 14. What is the meaning of -(i)/+(ii) and so on in Fig. 5A? Does this refer to Fig. 5D: + (ii)/- (i) MLN8237 of full length NuMA? Explain better.
 15. Fig. 5C: n>100; what does this mean? n>100 cells or dots?
 16. Fig. 6B, C: different time scales (90 s versus 200 sec) that can cause confusion.

Referee #2:

The authors study here the interplay between the roles of NuMA at the spindle pole and in the reforming nucleus after mitosis, with a particular focus on consequences of NuMA phosphorylation by Aurora A. There is quite a body of literature on NuMA's role for spindle assembly and on its nuclear function, for example Aurora A inhibition is known to lead to NuMA overaccumulation at poles. Here, the authors dissect in more detail the mechanism of this overaccumulation, NuMA's dynamic properties at poles and the consequences of NuMA overaccumulation on nuclear shape and aspects of the internal organization of the nucleus in the absence of Aurora A activity. The study impresses by a large number of often technically advanced and elegant experiments using new cell lines allowing more targeted perturbations and observations than in previous studies, for example a cell line in which Aurora A is degraded pre-maturely in anaphase. The main conclusion is that Aurora A prevents overaccumulation of NuMA at poles by weakening phase separation, in line with previous work; overaccumulation is suggested to then have downstream effects for chromosome decondensation/nuclear reformation, possibly because not sufficient NuMA can be recruited to chromosomes on time.

Major comments:

- (1) Despite the large body of carefully performed experiments presented in this manuscript, it can be difficult for the reader to pin-point what's really the major novelty of this study. Unfortunately the Introduction does not provide sufficient specific background which would allow putting the results presented here into the context of the quite extensive literature already existing on NuMA's and Aurora A's roles for spindle/pole assembly and for nucleus reassembly after mitosis. This prevents the authors from clearly stating in specific terms which key open questions their study addresses, given the existing literature, and then to focus on the key experiments addressing these questions. By not providing this background upfront, the reader is left wondering to which extent the findings are rather confirmatory versus novel.
- (2) Aurora A inhibition has been reported to lead to various spindle defects, most prominently monopolar spindles, observed in various organisms. The authors present here always bipolar spindles after Aurora A inhibition that appear to be a little shorter than wildtype spindles. Could they please clarify if they observed a mix of monopolar and bipolar spindles and then selected only bipolar spindles for analysis or whether they used a particular protocol, for example Aurora A inhibition after centrosome separation has already occurred, to ensure that bipolar spindles always form.
- (3) The authors point out several times that chromosomes are 'bent' around poles with overaccumulated NuMA, for example as shown in Fig. 2E, suggesting that this bending may cause defects when nuclei reform. Why does overaccumulation at poles cause chromosome bending? Is this because microtubules are too short, maybe due to an effect of Aurora A on the activity of other microtubule associated proteins? Or does NuMA start to interact with chromosomes before the nuclear envelope reforms?
- (4) In the Discussion the authors state that overaccumulation of NuMA at poles prevents correct subsequent nuclear localization of NuMA - this seems to be a suggestion for a different type of mechanism for the downstream effects of the overaccumulation of NuMA at poles than the 'bending of the nuclei', and indeed in Fig. 2E, NuMA appears to localize earlier to the inside of reforming nuclei than when Aurora A is inhibited. This phenomenon is however not quantified, despite the prominent statement in the Discussion. Does eventually all NuMA enter the nucleus even when Aurora A is inhibited and NuMA has become a 'solid'?
- (5) Fig. S4A: in the α -tubulin images, it seems that the MLN8237-treated spindle has less visible astral microtubules. Is this a random feature of this one spindle or is it something representative of the phenotype? And how does this connect with the overaccumulation of NuMA at the poles? (Fig. 4F and 5B)
- (6) In Fig. 3F, the authors show super-resolution images of NuMA around centrosomes and talk about a meshwork. Is that justified based on what can be observed in the images? This reviewer sees dots whose density and spread away from the centrosome vary depending on Aurora A activity, but not a "meshwork".
- (7) The data shown in Fig. S6A appear to be important and could be shown in a main figure, because they seem to demonstrate

clearly that condensates only form when NuMA is locally concentrated in cells in the presence of microtubules apparently by a mechanism involving dynein activity.

(8) In the experiments shown in Fig. 4B, it appears that endogenous NuMA was knocked down and replaced by variants with truncations. In all cases, bipolar spindles appear to form. Given the literature, is this expected? Or is bipolar spindle formation due to residual endogenous NuMA? Given that there may be fewer/varying numbers of microtubules in the spindles when endogenous NuMA is knocked down, it would be important to report also the NuMA fluorescence intensity at poles divided by the microtubule intensity at poles. And, Fig. 4B (v) shows a peculiar staining the authors do not comment on. Has this been seen before? Is it expected?

(9) In Fig. 4 and 5, several NuMA mutants are generated in each of which many amino acids in the C-terminal part are mutated (R>G, Aro>A, Q>G). All these mutants show reduced pole binding (Fig. 4B and 5D). The authors have very specific interpretations based on expectations of what the different kinds of amino acids that were replaced are expected to do in condensates. Is this justified? The mutations may also have considerable effects on microtubule binding given that microtubule binding parts have been described to be present in the NuMA C-terminus. Moreover the C-terminal part has recently been shown to bind with preference to microtubule minus ends. Can the authors exclude that these properties are affected by the mutations and that the observed reduction in pole localization is simply due to reduced microtubule (minus end) binding? Controls appear to be missing showing that mutating similar numbers of other amino acids not expected to affect microtubule binding or phase separation can still localize to spindle poles. In the absence of clear separation-of-function mutants either only affecting condensation or microtubule binding, conclusions could be stated a bit more cautiously.

(10) The authors conclude that glutamine residues in the C-terminus of NuMA are responsible for "hardening" of condensates based on localization of mutants in which these glutamines have been removed (Fig. 5D). It seems that FRAP measurement with the various "phase separation mutants" mutants would be required to really make that point.

(11) The manuscript appears at times overly loaded with data, distracting sometimes from the key experiments. For example, does the half-bleach experiment in Fig. S5E provide any additional information compared to the other bleaching experiments? Do the experiments shown in Fig. 6 address the key question of the study? The experiments shown in Fig. 7 appear interesting, but what would be the appropriate controls to put the results of the 4C-seq experiments into context?

Minor comments:

(12) Language: some statements appear to lack clarity/precision. Some examples:

- Abstract: what does "...material properties...organizing the nucleus and genome..." mean?
- Introduction: the impression is given that spindles in all eukaryotic cells have focused poles
- end of first paragraph of Introduction: "...coordination ... is essential; however, the consequences of failed coordination remain unclear" can sound like a contradiction.
- "poles orchestrate mitotic spindle assembly by directing microtubule convergence around centrosomes" is unclear - aren't poles the consequence of a combination of molecular activities coming together?

(13) page 7: "this phenotype was independent of the previously defined role of Aurora A in spindle assembly" - missing reference.

(14) page 10: "We detected no enrichment in other centrosomal proteins, such as γ -tubulin and Cep152 (Fig. S4A)". The levels of these two proteins actually decreased significantly, as shown in the figure. It is not clear why the decrease of PCNT was mentioned, while the decrease of these other two proteins was called "no enrichment". And could the authors speculate why such proteins decrease in MLN8237-treated cells? And possibly revise the related statement in the Discussion?

(15) page 13: typo: 1.5 microM instead of mM

(16) Is the scheme in Fig. 8 really helpful in its current form?

Minor Figure and Movie comments

(17) Fig. 3D: Given that two distinct microtubule binding regions have been identified in NuMA, in the legend it would be best to specify that the "MTs" domain refers to what is commonly known in the literature as MTBD1, and to add the appropriate reference.

(18) Fig. S1B: in the legend, the meaning of "bkgd" is introduced only later for panel D.

(19) Fig. S1F: The horizontal "Control" label in the left upper corner should be "HeLa Kyoto" like in panel G.

(20) Fig. S1H: the legend states "monoclonal cell line expressing CycB-Aurora Ar-AcGFP in control or upon transfection with control and Aurora A siRNA" -should the first in control be removed.

(21) Movie S1 and S2 legends lack the reference to the related figures.

Comments on Methods

(22) How was the "spindle pole dissolution" measured? Specifically, what NuMA intensity was considered to correspond to a dissolved pole?

(23) How were the half-time for recovery determined given that there were mobile and immobile fractions? How were the two fractions quantified?

(24) Is there a reason why for some datasets the SEM was plotted, while for some other the SD?

Referee #3:

This study explores the coordination between nuclear envelope (NE) dynamics and spindle pole organization during mitosis. It reveals that Aurora A kinase maintains the spindle pole protein NuMA in a dynamic state during anaphase. Without Aurora A, NuMA solidifies and accumulates at the poles, causing mispositioning of segregated chromosomes. NuMA localization depends on dynein/dynactin interactions, its coiled-coil domain, and its intrinsically disordered region (IDR). Key molecular interactions within the IDR regulate this process, with cation- π interactions maintaining NuMA at the poles and glutamine residues triggering its solidification when Aurora A is inhibited. The findings highlight the importance of spindle pole material properties in nuclear and genome organization post-mitosis. Overall, the experiments are well-executed, and the figures are well-prepared. I am happy to support publication in the EMBO Journal provided that the authors can address the following points.

The authors observed that the inhibition of Aurora A kinase activity during mitosis leads to misshaped nuclei in newly formed cells. To investigate this effect, they used an AcGFP-tagged, CycB-dependent Aurora A construct that degrades during the metaphase-to-anaphase transition, allowing them to study Aurora A's role in anaphase without interfering with its function during other phases of the cell cycle. In Fig. 1F, they demonstrate the degradation of this construct during anaphase. It would be beneficial to include a comparison showing the behavior of wild-type Aurora A during this phase. While they discuss the advantages of their construct on pages 6-7, it remains an artificial system, which could have implications for protein folding and phase transition.

The study reports NuMA accumulation at the poles during anaphase in cells treated with an Aurora A inhibitor and in cells expressing the modified Aurora A construct. Based on these observations, the authors conclude that Aurora A activity during anaphase is required for NuMA dissolution at the poles (Fig. 2).

Photobleaching experiments revealed that NuMA mobility depends on Aurora A activity. This dependency also applies to a phosphorylation-resistant NuMA variant, providing strong evidence that phosphorylation at S1969 is necessary for NuMA mobility (Fig. 3).

Figs. 4 and 5 focus on the C-terminal intrinsically disordered region (IDR). The authors employ the Corelet system to investigate phase transition, a well-established approach in this context (>100 orders at Addgene). However, the division of mutations across two figures is suboptimal. It might be clearer to group all immunostaining data together and present the Corelet data separately.

The labeling of constructs is somewhat unclear. The use of labels such as (i, ii, etc.) could be replaced with construct names for improved clarity. Additionally, the intensity of immunostainings in Fig. 4B does not appear to correlate with the quantifications in Fig. 4C. In Fig. 4D, "Glu" should be replaced with "Gln." Furthermore, there is no graphical representation of the NuMA-Cterm construct in Fig. 4E, which would be helpful for visualization.

A limitation of the study is that the authors do not demonstrate full-length NuMA accumulation in condensates. Instead, they use a construct consisting of amino acids 1700-2057 (IDR region), which lacks the N-terminal region and 58 C-terminal residues.

The authors then use a Kaede-based system to artificially increase NuMA multivalency, leading to its accumulation in a "non-dynamic pool." However, it is unclear whether this experiment is meant to compensate for the absence of a full-length NuMA experiment using the Corelet system. Additionally, the term "non-dynamic pool" needs clarification-does it correspond to "light-induced condensates"?

In Fig. 6, centrinone treatment was used to generate cells with a single centrosome. In these cells, phosphorylated Aurora A localized exclusively at the pole containing SAS6, while NuMA preferentially accumulated at the pole lacking Aurora A. This further supports the conclusion that Aurora A is required for NuMA release during anaphase.

Fig. 7A/B focuses on nucleoli positioning in newly formed nuclei of Aurora A-inhibited cells. The authors categorize nucleolar distributions as "uniformly spread" or "randomly spread." However, the representative image does not clearly illustrate this distinction, and a statistical evaluation of these distributions is missing.

To assess genome organization in newly formed cells, the authors employed high-resolution 4C-seq. They observed that MLN8237-treated cells exhibited an increased tendency for rDNA clusters to contact the intramolecular arm, leading them to conclude that chromosomes remain more compact. However, the effect appears small and should be discussed in the context of other studies utilizing this technology.

Minor Points:

The study investigates Aurora A function using an artificial construct or Aurora A inhibitors. While the discussion mentions the relevance of Aurora A inhibitors for cancer treatment, the physiological significance of the observed condensates (i.e., whether they exist in vivo) remains unclear.

The authors acknowledge that phase transitions are concentration-dependent. However, they should further discuss this in relation to their experimental setup (stable vs. transient transfections) and the impact of mutations (R/Q/Aro) on expression levels.

The title includes "phase transition," yet this term is not used in the abstract and is rarely mentioned in the main text. Instead, the abstract refers to "dynamic state" or "solid state," while the main text uses terms like "non-dynamic pool," "accumulation," or "condensates." It remains unclear whether the observed phenomenon represents true phase separation or merely a protein assembly based on phosphorylated NuMA. This should be explicitly clarified.

The point-by-point response to reviewers**Reviewer #1**

We sincerely thank the reviewer for their thoughtful and generous assessment of our work, especially for noting that *"this manuscript sheds light on a largely overlooked aspect of the disassembly of spindle poles at the end of mitosis."* We greatly appreciate the reviewer's insightful comments, which have helped us strengthen the manuscript. We have addressed all concerns through additional experiments and text revisions. As requested, we have now expanded the Introduction to include a more detailed discussion on the role of Cdk1 in regulating NuMA dynamics during mitosis.

Major points:

1. The authors have shown in Rajeevan et al. (MBoC 2020) that NuMA interacts with chromatin in interphase and prophase via its C-terminus. Expression of a DNA-binding-deficient mutant of NuMA affects chromatin decondensation at the mitotic exit and nuclear shape in interphase. Is it possible that AuroraA inhibition impacts NuMA binding/loading to chromatin? In this case, the nuclear shape phenotype and the defects described in Fig. 7 would not be (or only partly) caused by NuMA at the spindle poles. The authors should discuss this possibility.

Response: We thank the reviewer for raising this point. We agree that the chromatin defects described in Fig. 7 (now Fig. 8) could result from (1) significantly reduced nuclear NuMA levels, (2) persistent accumulation of NuMA at the spindle poles, or (3) a combination of both. Reviewer 2 also raised a similar concern (see their major concern #4).

To address this thoroughly, we have now quantified NuMA localization in newly formed nuclei under control conditions and following Aurora A inhibition. As expected, our data show a marked reduction in nuclear NuMA levels upon Aurora A inhibition (see new Fig. S3A and S3B). As suggested by the reviewer, in the discussion section of the revised manuscript (p. 28), we have discussed the possibility that reduced nuclear NuMA and its persistent pole-associated accumulation could both contribute to the observed chromatin contact defects (as revealed by 4C analysis).

However, we believe the nuclear shape defects are primarily due to NuMA accumulation at the spindle poles. To test this directly, we have now analyzed nuclear morphology in cells expressing AcGFP-NuMA^{r_{ΔNLS}} (a NuMA mutant that fails to localize to the nucleus) following depletion of endogenous NuMA. This analysis reveals that abnormal NuMA accumulation at the poles, but not its reduced levels in the nucleus, predominantly underlie the nuclear shape defects observed upon mitotic exit after Aurora A inhibition. These new results are shown in Fig. S3 and discussed on p. 10 of the revised manuscript.

2. "Intriguingly, a significant number of CycB-AuroraA-AcGFP expressing cells" What is this number? It is not given in Fig. 1J or K.

Response: Thanks, we have now slightly revised this sentence on p. 8 and directed the readers to the nuclear shape defects presented in Fig. 1J and 1K, where we have utilized more than 26 cells, as indicated.

3. Fig. 1I: as Control, it would be better to express an siRNA resistant Aurora A.

Response: We thank the reviewer for asking for this control. A similar point was raised by Reviewer 3 (see their major concern #1). We have now examined nuclear organization in cells stably expressing an siRNA-resistant Aurora A-AcGFP construct. As shown in the new Fig. S1I-S1K, HeLa cells expressing Aurora A-AcGFP do not exhibit any significant difference in nuclear shape upon treatment with either control or Aurora A siRNA. These new findings are discussed in the revised manuscript on p. 8.

4. Fig. 3B: the pole of the control spindle is bleached and the signal recovers to ca. 30% within 3-5 min and then remains stable (blue line). This happens in all 14 cells. Why is there no further recovery? Why is NuMA at spindle poles in control cells during anaphase (see Fig. 2G)? I assume that the authors picked relatively rare anaphase cells with NuMA at spindle poles - please indicate this in the text.

Response: We thank the reviewer for this valuable point. NuMA intensity naturally decreases at the poles as cells progress from metaphase to anaphase (Kotak et al., 2013; *EMBO J*). Because of

this reason, we performed the FRAP analysis specifically in early anaphase cells, before cleavage furrow formation. During this time, low levels of NuMA at the poles can be seen, for instance, see Fig. 2E and 2G.

In response to the reviewer's suggestion, we have now substantially revised the FRAP dataset by analyzing a larger number of anaphase cells and including the $T_{1/2}$ and mobile fraction values. As expected, this new data reveal that the NuMA pool is significantly dynamic at the poles in anaphase, in contrast to Aurora A-inhibited cells. Please note that the non-bleached pole also shows a marked decrease in intensity as mitosis progresses, which may partly explain the limited recovery seen in bleached control samples. We have now clarified this in the revised Fig. legend.

5. Fig. S5E : I do not see recovery in the Control and the MLN8237 cells. How was the intensity quantified in Fig. S5F? It basically starts at 0.0. I guess that only part of the spindle pole was quantified. Please explain.

Response: Although we initially included the half-bleach analysis, we found it somewhat redundant given the full FRAP experiments conducted in metaphase and anaphase, as well as the mEOS-based photoconversion experiments. In response to a reviewer 2 comment (major concern #11), we have removed the half-bleach analysis from the revised manuscript. Instead, we have strengthened the full FRAP dataset by increasing the number of metaphase and anaphase cells analyzed and by improving the quantitative evaluation of the mEOS-based photoconversion experiments (see below). We hope the reviewer agrees with this revision and supports our decision to omit the half-bleach data.

6. Fig. S5C, G: the spread of the data points is quite high in the MLN8237 cells - why? Number of MLN8237 should be increased. Since it is a key experiment, the 2nd Aurora A inhibitor should be used to confirm data.

Response: As suggested by the reviewer, we have increased the number of cells analyzed (both control and MLN8237-treated) and included additional data using a second Aurora A-specific inhibitor: MK5108. These new results are presented in the revised Fig. S5 and discussed on p. 12-13 of the revised manuscript. We agree with the reviewer that the data show greater variability in metaphase and anaphase cells that are treated with Aurora A inhibitor, and this variability may reflect differences in the extent of Aurora A inhibition across cells during the 60-min treatment window.

7. Fig. 3E is a nice experimental set up. However, the line scan is risky because a 3D object is quantified by a 1D analysis. I suggest measuring the entire pole signal. In addition, spindle pole localization of NuMA in Control and MLN8237 cells is quite different (more extended in the Control - why? Should be less). More than 30 cells were analyzed. The authors should find a way to summarize the behavior of these cells in one graph.

Response: We agree with the reviewer that 1D line scan analysis may overlook spatial nuances in protein distribution across the entire spindle pole. To address this, we have now increased the thickness (3 μm) of the line used for analysis to match the approximate width of the spindle pole. In the manuscript, we have consistently measured spindle pole intensity using a constant ROI from the maximum intensity projected metaphase or anaphase images. In line with this approach, based on the reviewer's comment, we ensured that the line scan covered the full area of interest rather than a single narrow line through the pole. Additionally, as suggested, we have now included the quantification from 10 cells in a consolidated graph.

Also, as noted by the reviewer, control cells have more spread/extended NuMA localization, which can be seen in Fig. 2A or 2E, and this is much more compact when cells are treated with Aurora A inhibitor. Please see also Fig. S2I.

8. Fig. 3F: Why does NuMA have a similar signal intensity at spindle poles in Control anaphase and MLN8237 treated cells? Is it possible to perform time-resolved 3D-SIM to see how the fibers change over time?

Response: We apologize for not providing sufficient detail in the Methods regarding our 3D-SIM² analysis. Due to the pronounced accumulation of NuMA at the spindle poles following Aurora A inhibition (MLN8237 treatment), it was essential to avoid overexposure to accurately visualize NuMA structures. Consequently, control and MLN-treated samples were imaged using different laser intensities. Normalizing NuMA intensity to the MLN8237 condition would have compromised the accurate assessment of NuMA organization in control cells, which explains why the pole intensities appear similar. We have now explicitly clarified this point in the revised Methods.

3D-SIM² analysis was performed on fixed samples using a primary NuMA antibody and a stable Alexa Fluor 488-conjugated secondary antibody. Although we attempted to perform live-imaging-based super-resolution analysis using AcGFP-NuMA^r, it was not feasible as the signal was too dim and exhibited rapid photobleaching, preventing us from doing time-resolved imaging of NuMA organization at the poles.

9. Do mutations that were introduced in Fig. 4A affect MT binding?

Response: We thank the reviewer for raising this point. A similar point was also raised by reviewer 2 (please see their major concern #9). In the past, several investigators have analyzed NuMA localization where they have perturbed its both microtubule-binding domains: MTBD1 (amino acids 1914–1985; Du et al., 2002, *Curr. Biol.*; Haren and Merdes, 2002, *JCS*) and MTBD2 (amino acids 2002–2115; Gallini et al., 2016, *Curr. Biol.*). MTBD1 harbors the conserved NuMA–LIN-5–Mud (NLM) motif found in NuMA orthologs, whereas MTBD2 is not a conserved feature across NuMA-related proteins. Importantly, deletion of a large portion of MTBD1 does not abolish spindle pole localization of NuMA in HCT116 cells (Tsuchiya et al., 2021, *Curr. Biol.*; see Fig. 1). Instead, it was shown that MTBD1 is needed for spindle pole focusing in HCT116 and mouse fibroblast cells (Tsuchiya et al., 2021, *Curr. Biol.*; Silk et al., 2009, *JCB*). Surprisingly, in mouse keratinocytes, MTBD1 is dispensable for both pole localization and pole focusing (Seldin et al., 2016). Given that deletion of either MTBD1 or MTBD2 generally does not severely impact NuMA pole localization, we are of the opinion that the failure of NuMA mutants: NuMA_(R>G) or NuMA_(Aro>A) to accumulate at the pole cannot be due to impaired microtubule binding. However, considering the reviewers' concerns, we have now tested the ability of NuMA_(Aro>A) to colocalize with microtubules in the cytoplasm during interphase. This was achieved by deleting NLS in addition to having Aro>A mutations [referred to as AcGFP-NuMA_(ΔNLS; Aro>A)]. Notably, AcGFP-NuMA_(ΔNLS; Aro>A) colocalizes with the microtubules in interphase, despite failing to localize at the pole in mitosis. Thus, we think that the inability of Aro>A, or R>G can't stem from failure to associate with the microtubules. Since the previous findings have emphasized that MTBD1 and MTBD2 are not key for pole localization in various cells, we have decided not to add this data to the manuscript; however, we have discussed this point in the revised manuscript on p. 26.

Immunofluorescence analysis of HeLa cells (#1, and #2) expressing AcGFP-NuMA^r(Aro>A;ΔNLS). These cells were stained with anti-GFP (green) and anti- α -tubulin (red) antibodies. DNA is shown in blue. The line-scan analysis of the indicated region (rectangular box in merge panel) is shown at the bottom. This analysis indicates colocalization between AcGFP-NuMA^r(Aro>A;ΔNLS) and microtubules.

10. Measuring the dynamic properties of Q>G (Fig. 5) by FRAP would be an important experiment.

Response: We thank the reviewer for raising this important point. In response (and in line with reviewer 2 major concern #10), we have now assessed the dynamic behavior of the AcGFP-NuMA_(Q>G) mutant. This analysis reveals that AcGFP-NuMA_(Q>G) is highly dynamic compared to full-length NuMA. Due to the very weak pole localization of AcGFP-NuMA_(Q>G) mutant, we conducted the FRAP experiment under monopolar spindle conditions induced by Kinesin-5 inhibition-STLC [S-Trityl-L-cysteine (STLC)], which helped concentrate the AcGFP-NuMA_(Q>G) signal at a single pole, thereby improving the feasibility and accuracy of the recovery measurements.

This new dataset is now included in Fig. S7G-S7J and discussed in the revised manuscript on p. 19.

11. Fig. 5G (AcGFP-NuMA) looks different to Fig. 2A (antibodies, no tag). Does this mean that the tag impacts NuMA at spindle poles? This would be concerning. Y-axis in Fig. 5G? Was the entire poly signal measured or only a circular area of ? nm?

Response: We apologize for previously overlooking this point. AcGFP-NuMA is a functional construct that faithfully recapitulates the localization pattern of endogenous NuMA, as demonstrated in our earlier studies (Rajeevan et al., 2020, *MBoC*; Sana et al., 2022, *JCB*), in

which we generated monogenic cell lines expressing this fusion protein. The AcGFP-NuMA expressing cell shown in the earlier version of the manuscript was highly overexpressing the construct, resulting in a strong GFP signal visible behind the segregated chromosomes at the spindle poles. We had deliberately chosen this example to highlight that even under conditions of NuMA overexpression, its dynamic nature at the poles does not induce chromosome bending. Prompted by the reviewer's comment, we have now revised Fig. 6B to include images of AcGFP-NuMA expressing cells with expression levels comparable to those of endogenous NuMA.

We would also like to clarify our intent with Fig. 2A, where we aimed to quantify the NuMA signal localized at the poles in control and MLN8237-treated cells. To avoid signal saturation, we set the fluorescence detection threshold based on the MLN8237-treated cells, which display abnormally high NuMA accumulation at the poles. As a result of this thresholding, the lower-intensity endogenous NuMA signal in control cells, typically visible until mid-anaphase (Kotak et al., 2013, *EMBO J*; Rajeevan et al., 2020, *MBoC*), is not readily apparent in the image.

As suggested, we have now added Y-axis labeling to the revised Fig. 6B. Consistent with the analysis in other panels, pole-associated signal intensity was quantified using a maximum intensity projected image using a circular region of interest with a diameter of 2.8 μm .

12. Fig. 6F needs FRAP analysis (Sas-6-mCherry as centrosome marker and Ac-GFP-NuMA). Comment on line scan analysis see 7.

Response: As requested, we have now analyzed the AcGFP-NuMA FRAP recovery profiles at both poles under the 2-day centrinone treatment condition. As shown in the new Fig. S8C-S8E, NuMA at the pole with centriole/SAS6 signal recovers significantly faster than at the pole lacking centriole/SAS6. Please note that this analysis was performed in metaphase cells because: (1) the number of metaphase cells is considerably higher than anaphase cells, and (2) AcGFP-NuMA accumulates more robustly at the control pole (or SAS6-positive pole) during metaphase compared to anaphase, facilitating a more reliable assessment of recovery profiles.

Minor points:

1. B-D: y-axis of the graphs needs description.

-Added, thanks

2. p. 8: As similar observation was made in other cell lines - please name them in the result part.

-Included, thanks

3. Fig. 2E: the line scan intensity does not fit with the images above. For example, Control at 9 min signal intensity is max and as high as after 15 min although in the image the mCherry-NuMA signal after 15 min is much lower than after 9 min.

-We thank the reviewer for highlighting this discrepancy, which arose from differing Y-axis intensity values that were not indicated in the graph. Since, we have now measured nuclear and pole intensity of NuMA in control and MLN8237-treated cells, we have removed the graphs from Fig. 2E.

4. p. 10. "Additionally, we show that AuroraA-mediated phosphorylation at S1969 of NuMA dynamically regulates pole-localized NuMA...." This was not shown!

-In the previous version of the manuscript, we demonstrated (see Fig. S5I) that photoconverted mEOS-tagged NuMA mutated at S1969 fails to accumulate at the non-photoconverted pole, indicating that phosphorylation at S1969 by Aurora A kinase dynamically regulates NuMA localization at the poles. This data is now shown in the revised Fig. S5E.

5. p. 10: "massively accumulates" suggests a recruitment mechanism. However, it just stays and is not disassembled.

-We respectfully disagree with the reviewer. The term "massive accumulation" reflects not only recruitment but also maintenance, or a combination of both processes. Therefore, we believe that "massive accumulation" is a neutral term that simply denotes the abundance of a macromolecule, regardless of the underlying mechanisms.

6. p. 10: NuMA is not a PCM component. Thus, other PCM localized proteins is incorrect. MTs "remained unaltered at the poles in AuroraA inhibited anaphase cells". Unaltered compared to wild type or metaphase...? Be more specific.

-Apologies, we have fixed these errors in the revised text.

7. Fig. S4 - remove (A - only one pannel). Would be nice to do double staining with NuMA.

-As requested, we have now removed panel 'A' from Fig. S4. Regarding the double-staining point, we did not perform it because both the NuMA antibody and the antibodies against a few PCM-localized proteins were raised in mouse, making it technically challenging. However, to ensure the efficacy of Aurora A inhibition, it is standard practice in our laboratory to perform NuMA staining in parallel as a control.

8. Fig. 3F: the order of the staining in the enlargements should be the same: NuMA/PCNT/Merge.

-Thanks, fixed.

9. p. 13: (>1.5 mM) - in figure S6: 1.5 μ M (this is the correct value).

-Thanks, it is now corrected.

10. Authors have to check expression levels by IB of the constructs in Fig. 4.

-We have now included immunoblots (IB) for all constructs in the revised Fig. S6G and S7D. As the reviewer is aware, divergent constructs often show variable expression levels in transient transfections, which also vary between cells. IB provides an outcome that averages across all transfected cells, some with very high and others with very low protein levels. Therefore, we generally avoid relying solely on immunoblotting to assess expression, unless we study monogenic stable lines. Instead, for quantification in our microscopy analysis, we focus on cells exhibiting relatively uniform expression across constructs, and we think this kind of analysis is more precise for quantification.

11. The authors should indicate how many R>G, Aro>A, Q>G mutations were introduced in the IDR. 39 R? 14 Y/F/W?

-This information is now added.

12. p. 13: telophase

-Corrected, thanks.

13. p.14: NuMA C-terminus is predicated to be largely disordered ...

-Corrected, thanks.

14. What is the meaning of -(i)/+(ii) and so on in Fig. 5A? Does this refer to Fig. 5D: + (ii)/- (i) MLN8237 of full length NuMA? Explain better.

-Thanks, -/+ referred to in the absence or presence of MLN8237; we have now improved this Fig. and the text in the related Fig. legend.

15. Fig. 5C: n>100; what does this mean? n>100 cells or dots?

-We have now revised the Fig. legend and have added this information. For this analysis, we analyzed at least 20 cells in each condition and studied the behavior of 5 condensates/cells.

16. Fig. 6B, C: different time scales (90 s versus 200 sec) that can cause confusion.

-Thanks, we have now improved this data (now Fig. 7) by adding data covering more time points, and we hope that this will not create confusion for the readers.

Reviewer #2

We are grateful to this reviewer for acknowledging that “*The study impresses by a large number of often technically advanced and elegant experiments using new cell lines allowing more targeted perturbations and observations than in previous studies.*” At the same time, the reviewer raised several important concerns, which we have addressed systematically, as detailed below. We sincerely thank the reviewer for their constructive feedback, which has significantly helped us strengthen the manuscript.

Major comments:

(1) Despite the large body of carefully performed experiments presented in this manuscript, it can be difficult for the reader to pin-point what's really the major novelty of this study. Unfortunately the Introduction does not provide sufficient specific background which would allow putting the results presented here into the context of the quite extensive literature already existing on NuMA's and Aurora A's roles for spindle/pole assembly and for nucleus reassembly after mitosis. This prevents the authors from clearly stating in specific terms which key open questions their study addresses, given the existing literature, and then to focus on the key experiments addressing these questions. By not providing this background upfront, the reader is left wondering to which extent the findings are rather confirmatory versus novel.

Response: We thank the reviewer for highlighting this important point. As detailed in the revised manuscript, we have now included a more comprehensive background with citations to relevant literature on NuMA localization at spindle poles. In addition, we have clarified key unresolved questions and emphasized the novelty of our work. Below, we outline a few key features that we believe make our study unique:

1. We have demonstrated that Aurora A activity during anaphase is critical for maintaining proper three-dimensional nuclear organization. To probe this, we developed a novel genetic tool allowing for rapid depletion of Aurora A (within ~10 minutes) at the metaphase-to-anaphase transition.
2. Previous studies have shown that Aurora A regulates NuMA levels at the poles in metaphase (Gallini et al., 2016, *Curr. Biol.*; Kotak et al., 2016, *JCS*). However, whether Aurora A activity is required continuously during anaphase to preserve NuMA dynamics at the poles was unknown. Our work addresses this gap. Notably, our work reveals the importance of dynamic NuMA-based poles for nuclear organization post-mitosis.
3. While earlier studies using dynein/dynactin-specific antibodies or dynein light chain knockout cells established a role for dynein/dynactin in NuMA pole accumulation (Merdes et al., 2000; He et al., 2023), a detailed domain-level understanding was lacking. We show that NuMA accumulation at spindle poles requires both its interaction with dynein/dynactin (via the N-terminal region) and multivalent interactions mediated by its central coiled-coil and C-terminal intrinsically disordered region (IDR). Notably, our findings challenge recent reports suggesting that NuMA accumulation at the poles is solely driven by its C-terminus via LLPS (Sun et al., 2021, *Nat. Comm.*; Ma et al., 2022, *JMGB*).
4. Through extensive mutagenesis, we reveal that cation- π interactions between arginine and aromatic residues in the IDR are essential for NuMA pole localization. We also show that glutamine residues promote the transition of NuMA from a dynamic to a solid state upon Aurora A inhibition. As the reviewer will note, we have further strengthened these data.

(2) Aurora A inhibition has been reported to lead to various spindle defects, most prominently monopolar spindles, observed in various organisms. The authors present here always bipolar spindles after Aurora A inhibition that appear to be a little shorter than wildtype spindles. Could they please clarify if they observed a mix of monopolar and bipolar spindles and then selected only bipolar spindles for analysis or whether they used a particular protocol, for example Aurora A inhibition after centrosome separation has already occurred, to ensure that bipolar spindles always form.

Response: As noted, the long-term (6-24 h) inhibition of Aurora A with MLN8237 leads to monopolar spindles in mitotic cells, thereby activating the spindle assembly checkpoint (SAC). This

prevents the investigation of Aurora A function during anaphase. Therefore, for most of our analyses with fixed samples, we used acute Aurora A inhibition (2 h) at 50 nM, and chiefly focused on metaphase and anaphase cells. Under these conditions, we significantly lose active Aurora A marks at the poles, suggesting our experimental regimen is potent to remove active Aurora A pools in the cells (see new Fig. S2A and S2B). While these conditions do induce minor spindle defects (for instance, ~16% of cells show chromosome instability, and ~8% of cells show a monopolar spindle), the majority of the cells progress to metaphase and subsequently to anaphase. We have now included the information on the percentage of cells with chromosome instability and monopolar spindles in new Fig. S2C and S2D and clarified that only metaphase and anaphase cells without signs of chromosome instability were analyzed to examine the Aurora A-NuMA axis.

(3) The authors point out several times that chromosomes are 'bent' around poles with overaccumulated NuMA, for example as shown in Fig. 2E, suggesting that this bending may cause defects when nuclei reform. Why does overaccumulation at poles cause chromosome bending? Is this because microtubules are too short, maybe due to an effect of Aurora A on the activity of other microtubule associated proteins? Or does NuMA start to interact with chromosomes before the nuclear envelope reforms?

Response: We thank the reviewer for this intriguing point. At this stage, we do not know precisely—why, upon Aurora A inhibition, the chromosomes bend around the NuMA-based poles. Interestingly, while imaging the control cells at 1-2 min time intervals in the context of our cytokinesis work, we have observed a slight bending of chromosomes around mid-anaphase, which appears to coincide with the initiation of cytokinesis (please see below the representative images from a time-lapse movie). A similar observation has been reported previously (e.g., see Fig. S3 in Samwer et al., 2017, *Cell*), but not formally documented.

Confocal live-imaging analysis of HeLa cells stably expressing mCherry-H2B (DNA). Cell shape can be viewed in Differential Interference Contrast (DIC) images. Please note slight bending of the segregated chromosome ensemble at the time of cleavage furrow formation (indicated by asterisks).

We suspect that cytokinetic furrow-induced cytoplasmic flow, along with mitotic cell remodeling (from a round to an oval shape), might exert some mechanical forces that bend chromosome edges. Under control conditions, this bending does not cause chromosome bending/wrapping, likely because NuMA-based spindle poles have diminished by this stage. In contrast, Aurora A inhibition results in persistent NuMA accumulation with altered material properties at the poles, potentially creating a mechanical barrier. This may cause pronounced chromosome bending and nuclear distortion in these cells. While this is an intriguing possibility, we have opted not to elaborate on this hypothesis in the manuscript to maintain focus on the central narrative.

An alternative explanation, as the reviewer suggests, is that Aurora A inactivation affects the localization or function of other microtubule-associated proteins, which, together with NuMA altered material properties, contribute to the observed phenotype. Nevertheless, given that artificially increasing NuMA multivalency using Kaede-NuMA mimics the MLN8237-induced phenotype (Fig. 6), we believe that NuMA phase transition upon Aurora A inactivation plays a direct and central role in driving this defect. Additionally, if NuMA were to interact with chromosomes before nuclear envelope reformation, we would expect to observe its localization at chromosomes during early or mid-anaphase in Aurora A-inhibited cells, which we do not.

(4) In the Discussion, the authors state that overaccumulation of NuMA at poles prevents correct subsequent nuclear localization of NuMA - this seems to be a suggestion for a different type of mechanism for the downstream effects of the overaccumulation of NuMA at poles than the 'bending of the nuclei', and indeed in Fig. 2E, NuMA appears to localize earlier to the inside of

reforming nuclei than when Aurora A is inhibited. This phenomenon is however not quantified, despite the prominent statement in the Discussion. Does eventually all NuMA enter the nucleus even when Aurora A is inhibited and NuMA has become a 'solid'?

Response: We thank the reviewer for bringing up this point, a similar concern was also raised by reviewer 1 (see their major concern #1). As shown in Fig. 2E, acute Aurora A inactivation leads to NuMA accumulation at spindle poles. Concurrently, we observe that when NuMA remains sequestered at the poles, its nuclear enrichment is significantly delayed. These findings have now been incorporated into the revised Fig. S3A and S3B and discussed on p. 10.

As the reviewer will note, NuMA eventually accumulates to substantial levels in the nucleus. Interestingly, the nuclear shape defects observed during mitotic exit under Aurora A inhibition are possibly not because of weak NuMA levels inside the nucleus. This is supported by new data showing that cells expressing AcGFP-NuMA $_{\Delta NLS}$, which lacks a nuclear localization signal, do not exhibit bending of the segregated chromosome mass near the poles. This indicates that the chromosome bending phenotype specifically depends on persistent NuMA accumulation at the poles.

However, based on our previous work (Rajeevan et al., 2020, *MBoC*), which demonstrated that NuMA nuclear entry at mitotic exit is essential for proper chromosome decondensation, it is plausible that the chromosome contact defects observed in our 4C analysis (Fig. 8) could result from (1) significantly reduced nuclear NuMA levels, (2) persistent accumulation of NuMA at the spindle poles, or (3) a combination of both. We have now explicitly discussed this possibility in the revised manuscript on p. 28.

(5) Fig. S4A: in the α -tubulin images, it seems that the MLN8237-treated spindle has less visible astral microtubules. Is this a random feature of this one spindle or is it something representative of the phenotype? And how does this connect with the over-accumulation of NuMA at the poles? (Fig. 4F and 5B)

Response: We have now thoroughly analyzed MLN8237-treated cells for astral microtubules and found that the observed feature was a random occurrence in a single spindle. We have replaced that image with a more representative example in the revised Fig. S4.

(6) In Fig. 3F, the authors show super-resolution images of NuMA around centrosomes and talk about a meshwork. Is that justified based on what can be observed in the images? This reviewer sees dots whose density and spread away from the centrosome vary depending on Aurora A activity, but not a "meshwork".

Response: In response to this concern, we acquired additional super-resolution SIM² images of NuMA during both metaphase and anaphase and performed line-scan analyses across multiple z-planes in cells, as shown below. These analyses reveal that NuMA accumulates in a continuous, extended pattern near the spindle poles, which we describe as a 'meshwork.' While the structure may appear punctate (doty)—likely reflecting local differences in density—the line-scan profiles consistently show a contiguous signal without sharp intensity drops to zero, which would be expected if NuMA were organized into discrete puncta (or dots). Thus, based on spatial continuity, we refer to this organization as a meshwork.

Different z-plane from images obtained by SIM² analysis of NuMA either in metaphase or in anaphase cells. The line-scan profile of NuMA (in aqua) and cytoplasm (cyto.-in brown) is shown below the images.

Our use of the term 'meshwork' is also informed by historical context. As the reviewer may recall, NuMA is a large coiled-coil protein known to form multi-armed oligomers, as demonstrated

in early *in vitro* work using purified NuMA from *E. coli* (Harborth et al., 1999, *EMBO J*). These higher-order assemblies depend on both the C-terminal region and the large coiled-coil domain of NuMA. Unfortunately, our current super-resolution analysis does not have sufficient resolution to definitively resolve the molecular organization of NuMA at the poles. However, based on the observed patterns and previous biochemical insights, we speculate that NuMA may exist in higher-order assemblies in this region. In the revised manuscript, we have used the term 'meshwork' cautiously and with appropriate qualification.

(7) The data shown in Fig. S6A appear to be important and could be shown in a main figure, because they seem to demonstrate clearly that condensates only form when NuMA is locally concentrated in cells in the presence of microtubules apparently by a mechanism involving dynein activity.

Response: Thanks for this suggestion. We have now moved panel S6A to the new Fig. 4A and 4B.

(8) In the experiments shown in Fig. 4B, it appears that endogenous NuMA was knocked down and replaced by variants with truncations. In all cases, bipolar spindles appear to form. Given the literature, is this expected? Or is bipolar spindle formation due to residual endogenous NuMA? Given that there may be fewer/varying numbers of microtubules in the spindles when endogenous NuMA is knocked down, it would be important to report also the NuMA fluorescence intensity at poles divided by the microtubule intensity at poles. And, Fig. 4B (v) shows a peculiar staining the authors do not comment on. Has this been seen before? Is it expected?

Response: We thank the reviewer for raising this important point. For the analyses presented in Fig. 4 and Fig. 5 (control and mutant conditions), we assessed NuMA intensity at the spindle poles in cells that had assembled bipolar spindles. As these experiments were performed on fixed samples, we are unable to determine whether bipolar spindle assembly occurred with normal timing or was delayed. However, based on our observation of substantial chromosome instability in cells expressing mutant forms of NuMA (see below), it is likely that while cells do manage to form bipolar spindles, proper spindle assembly may be delayed.

(A-C) Different NuMA constructs (A), and IF analysis of HeLa cells expressing all these NuMA constructs (B) at 72- h after transfections with NuMA siRNA. These cells were stained with anti-GFP (green) antibody. The DNA is shown in magenta. As shown in B, many cells expressing either NuMA deletion, or mutated constructs show significant chromosome instability, which is measured in C, and added to the Fig. S6F.

Our primary objective in these experiments was to evaluate and quantify NuMA localization at the poles across various deletion/mutant constructs. Therefore, for illustrative purposes, we selected representative cells in which chromosomes were mostly aligned at the metaphase plate. In response to the reviewer's suggestion, we have now clearly indicated in the revised figure legends that cells expressing mutant NuMA constructs show significant chromosome instability. We refer to revised Fig. S6A and S6F for these data. As the reviewer correctly noted, the bipolar spindles observed in these cells could also be supported by residual endogenous NuMA.

In addition, as suggested, we have now quantified NuMA intensity at the poles normalized to microtubule intensity. These results are presented in the new panels in Fig. S6A–S6C. The

distinct localization pattern of the NuMA_(1700–2115) construct on the chromatin has been previously reported by us and others during metaphase, when cells are fully congressed on the metaphase plate, and not before chromosomes alignment on the metaphase plate (Rajeevan et al., 2020, *MBoC*; Serra-Marques et al., 2020, *JCB*). This information has now been included in the revised legend for Fig. 4D.

(9) In Fig. 4 and 5, several NuMA mutants are generated in each of which many amino acids in the C-terminal part are mutated (R>G, Aro>A, Q>G). All these mutants show reduced pole binding (Fig. 4B and 5D). The authors have very specific interpretations based on expectations of what the different kinds of amino acids that were replaced are expected to do in condensates. Is this justified?

Response: We appreciate the reviewer’s thoughtful comment. NuMA localizes to the spindle and spindle poles during mitosis and interacts with numerous proteins during this cell cycle phase. To assess whether the NuMA C-terminus can engage in multivalent interactions critical for its accumulation at the poles, we first employed the Corelet assay (also see Reviewer 3, Point #4). This system enabled us to probe NuMA-driven multimeric interactions specifically within the nucleus, thereby avoiding potential confounding influences from microtubules or mitosis-specific binding partners. Using this assay, we observed that both the R>G and Aro>A mutants fail to support multivalent interactions (Fig. 5C, D). Based on these findings, we next examined the localization of full-length NuMA constructs carrying these mutations and found that, unlike wild-type NuMA, these mutants fail to localize to the spindle poles (Fig. 5E-G). Thus, the results from the interphase Corelet assay and mitotic localization analyses are complementary and consistent with our interpretation.

Furthermore, over the years, we have studied the relevance of mitotic-specific NuMA phosphorylation. Notably, we generated a NuMA mutant in which 35 phosphorylated serine/threonine residues at the C-terminus of NuMA were mutated to alanine, while preserving the known phosphorylation sites of Cdk1 (T2055), and Aurora A (1969) implicated in pole localization. Importantly, this 35A mutant still localized to the spindle poles, suggesting that the pole localization defect of the R>G and Aro>A mutant is not simply due to large-scale peptide alteration. We include this analysis below for completeness, though we chose not to incorporate it into the main manuscript to maintain a focused narrative.

(A) Domain organization of AcGFP-tagged (S/T>A) mutant NuMA construct where 35 S/T residues present at the C-terminus of NuMA are mutated to alanine.

(B) IF analysis of HeLa cells expressing either AcGFP-NuMA^r or AcGFP-NuMA^r(S/T>A) constructs (B) at 72- h after transfections with NuMA siRNA. These cells were stained with anti-GFP (green) antibodies. The DNA is shown in magenta. As shown both cells show significant localization of AcGFP-signal at the poles.

(C) Schematic representation of the quantification method and the spindle pole intensity of NuMA constructs mentioned above. Error bars ± SD.

Taken together, we propose that the failure of the R>G and Aro>A mutants to accumulate at the poles results from disrupted multivalent interactions. That said, we agree with the reviewer that diminished localization could also arise from altered interactions with specific NuMA mitotic partners. We have now acknowledged this alternative possibility in the revised Discussion (p. 26).

(9B) The mutations may also have considerable effects on microtubule binding given that microtubule binding parts have been described to be present in the NuMA C-terminus. Moreover the C-terminal part has recently been shown to bind with preference to microtubule minus ends. Can the authors exclude that these properties are affected by the mutations and that the observed reduction in pole localization is simply due to reduced microtubule (minus end) binding? Controls appear to be missing showing that mutating similar numbers of other amino acids not expected to

affect microtubule binding or phase separation can still localize to spindle poles. In the absence of clear separation-of-function mutants either only affecting condensation or microtubule binding, conclusions could be stated a bit more cautiously.

Response: This is an important point, and we would also like to refer to our response to a similar concern raised by the reviewer 1 (major concern #7). As noted earlier, we did not specifically test whether the mutant proteins that fail to localize at the spindle poles are defective in microtubule binding. This decision was based on the following rationale:

NuMA contains two microtubule-binding domains: MTBD1 (amino acids 1914–1985; Du et al., 2002, *Curr. Biol.*; Haren and Merdes, 2002, *JCS*) and MTBD2 (amino acids 2002–2115; Gallini et al., 2016, *Curr. Biol.*). Previous studies have shown that deletion of both domains does not substantially impair NuMA localization to the spindle poles in HCT116 cells (Tsuchiya et al., 2021, *Curr. Biol.*), although deletion of MTBD1 can affect spindle pole focusing and assembly. Interestingly, in mouse keratinocytes, MTBD1 is dispensable for both pole localization and focusing, and instead plays a role in spindle orientation (Seldin et al., 2016, *MBoC*).

Nonetheless, in light of the reviewer's concern, we have now examined whether the Aro>A mutant can interact with microtubules. To test this, we expressed a version of AcGFP-NuMA_(Aro>A) lacking its NLS to retain it in the cytoplasm during interphase. As shown below, this analysis revealed that AcGFP-NuMA_(Aro>A; ΔNLS) does colocalize with microtubules, despite its failure to localize to spindle poles in mitosis.

Immunofluorescence analysis of HeLa cells (#1, and #2) expressing AcGFP-NuMA^{r(Aro>A;ΔNLS)}. These cells were stained with anti-GFP (green) and anti- α -tubulin (red) antibodies. DNA is shown in blue. The line-scan analysis of the indicated region (rectangular box in merge panel) is shown at the bottom. This analysis indicate colocalization between AcGFP-NuMA^{r(Aro>A;ΔNLS)} and microtubules.

Altogether, our findings, together with the published data on MTBD1/2 suggest that the inability of Aro>A or R>G mutants to localize to spindle poles is unlikely to stem from defective microtubule association. Given that prior studies have demonstrated that MTBD1 and MTBD2 are not essential for pole localization in various systems, we have opted not to include this additional data in the manuscript. However, we now discuss these findings and the relevant rationale in the revised Discussion section on p. 26.

(10) The authors conclude that glutamine residues in the C-terminus of NuMA are responsible for "hardening" of condensates based on localization of mutants in which these glutamines have been removed (Fig. 5D). It seems that FRAP measurement with the various "phase separation mutants" mutants would be required to really make that point.

Response: In response, we have now assessed the dynamic properties of the AcGFP-NuMA_(Q>G) mutant (a similar concern was also raised by the 1st reviewer, please refer to their major concern #10). As anticipated, FRAP analysis revealed that AcGFP-NuMA_(Q>G) is highly dynamic compared to full-length NuMA.

It is important to note that this experiment was performed under monopolar spindle conditions induced by Kinesin-5 inhibition using STLC. This setup was necessary because AcGFP-NuMA_(Q>G) exhibits only weak localization at each spindle pole, making standard FRAP analysis at bipolar poles technically challenging. The monopolar setting enhanced signal intensity at the pole, enabling a reliable assessment of AcGFP recovery dynamics.

These new results are included in revised Fig. S7G-S7J, and discussed in the revised manuscript on p. 19.

(11) The manuscript appears at times overly loaded with data, distracting sometimes from the key experiments. For example, does the half-bleach experiment in Fig. S5E provide any additional information compared to the other bleaching experiments? Do the experiments shown in Fig. 6 address the key question of the study? The experiments shown in Fig. 7 appear interesting, but what would be the appropriate controls to put the results of the 4C-seq experiments into context?

Response: We agree with the reviewer that the half-bleach experiments do not provide additional insights beyond those already offered by the other bleaching assays. As noted in our response to the major concern #5 of the reviewer 1, we have therefore removed the half-bleach experiments from the revised manuscript. However, we believe that the experiments involving the Plk4 inhibitor centrinone (revised Fig. 7) remain essential to the manuscript. These experiments effectively demonstrate that abnormally accumulated NuMA at a pole lacking Aurora A is unable to exchange with the dynamic NuMA pole. These results have two messages 1) centrosome localized active Aurora A is essential for this phenotype 2) the altered material state of NuMA at a pole lacking active Aurora A has become incapable of exchanging with the dynamic NuMA pool in a cell-thus revealing its true solid state nature. To strengthen this point further, and in response to reviewer 1, major concern #12, we have now performed FRAP analysis on both poles under asymmetric Aurora A localization, where both poles show asymmetric recovery, as expected.

Similarly, the 4C-seq experiments provide important insights about perturbed chromosomal organization upon Aurora A inhibition. Our experimental design, whereby MLN-treated cells are compared to control cells (including the visualization of the differential ratio in Fig. 8F) follows standard conventions in the field. To nonetheless confirm that the observations are not unique to the rDNA loci or short chromosome arms, we have repeated the analysis on a single-copy gene on the long arm of chr. 13, which generates similar results (Fig. S9G).

Minor comments:

(12) Language: some statements appear to lack clarity/precision. Some examples:

- Abstract: what does "...material properties...organizing the nucleus and genome..." mean?

- Introduction: the impression is given that spindles in all eukaryotic cells have focused poles

- end of first paragraph of Introduction: "...coordination ... is essential;however, the consequences of failed coordination remain unclear" can sound like a contradiction.

- "poles orchestrate mitotic pindle assembly by directing microtubule convergence around centrosomes" is unclear - aren't poles the consequence of a combination of molecular activities coming together?

-We thank the reviewer for all these points. As the reviewer will note, we have thoroughly improved the manuscript based on their suggestions.

(13) page 7: "this phenotype was independent of the previously defined role of Aurora A in spindle assembly" - missing reference.

-References added, thanks

(14) page 10: "We detected no enrichment in other centrosomal proteins, such as γ -tubulin and Cep152 (Fig. S4A)". The levels of these two proteins actually decreased significantly, as shown in the figure. It is not clear why the decrease of PCNT was mentioned, while the decrease of these other two proteins was called "no enrichment". And could the authors speculate why such proteins decrease in MLN8237-treated cells? And possibly revise the related statement in the Discussion?

-We have revised this sentence in the updated manuscript to improve clarity. At this stage, the reason why some centrosomal proteins appear reduced in MLN8237-treated cells remains unclear. One possibility is that Aurora A activity helps maintain their levels; however, without further investigation, it would be premature to make a definitive claim.

(15) page 13: typo: 1.5 microM instead of mM

-Our apologies for this mistake; we have fixed this error now.

(16) Is the scheme in Fig. 8 really helpful in its current form?

*-We have discussed our current model with several colleagues. While some appreciate its simplicity and clarity, particularly for conveying the overarching concept to a broader audience, others feel it lacks detailed mechanistic insight. Given the diverse readership of *The EMBO Journal*, we believe that the simplified model is valuable for effectively communicating the central message to those who may not be specialists in the field. Therefore, we would prefer to retain the current version in the main figure (now Fig. 9). That said, since *The EMBO Journal* also includes a small graphical synopsis, we are planning to present a more mechanistic version of the model there, which will emphasize key mechanistic insights from the manuscript. We hope this dual approach is acceptable to the reviewer.*

Minor Figure and Movie comments

(17) Fig. 3D: Given that two distinct microtubule binding regions have been identified in NuMA, in the legend it would be best to specify that the "MTs" domain refers to what is commonly known in the literature as MTBD1, and to add the appropriate reference.

-Added, thank you so much.

(18) Fig. S1B: in the legend, the meaning of "bkgd" is introduced only later for panel D.

-Fixed, thank you.

(19) Fig. S1F: The horizontal "Control" label in the left upper corner should be "HeLa Kyoto" like in panel G.

-Corrected.

(20) Fig. S1H: the legend states "monoclonal cell line expressing CycB-Aurora Ar-AcGFP in control or upon transfection with control and Aurora A siRNA" -should the first in control be removed.

-Corrected.

(21) Movie S1 and S2 legends lack the reference to the related figures.

-Added, thank you.

Comments on Methods

(22) How was the "spindle pole dissolution" measured? Specifically, what NuMA intensity was considered to correspond to a dissolved pole?

-Spindle poles were considered dissolved when no residual NuMA signal was visually detectable at the spindle poles in time-lapse movies acquired using confocal microscopy (see Fig. 2F, S2N, and S3G). While this assessment involves a degree of subjectivity, multiple graduate students independently examined similar movies to arrive at consistent conclusions. For quantitative analysis of the NuMA signal at the poles, the non-saturated NuMA intensity in MLN8237-treated/or CycB-Aurora A expressing cells was used to capture images, and similar conditions were utilized to capture images of control cells before quantification (see Fig. 2A and 2B, Fig. S2). These details have been added to the Methods section.

(23) How were the half-time for recovery determined given that there were mobile and immobile fractions? How were the two fractions quantified?

-We apologize for not including the details of our FRAP analysis in the Methods section in the previous version of the manuscript. As the reviewer will note, we have now provided a comprehensive description of the FRAP analysis, including appropriate citations to previous studies that employed similar methodologies. Briefly, the $T_{1/2}$ and mobile fraction values were derived from FRAP recovery curves (the equation used to generate these curves is now included in the Methods), which plot fluorescence intensity over time following photobleaching. To correct for fluorescence loss due to imaging-associated photobleaching, we simultaneously recorded fluorescence intensities from an unbleached reference region—typically the opposite spindle pole—throughout the acquisition.

(24) Is there a reason why for some datasets the SEM was plotted, while for some other the SD?

-Thank you for raising this point. We have used SEM (standard error of the mean) for all graphs in which curves are plotted with shaded areas (e.g., Fig. 1B-1D, 1H, 5D, etc.). This choice was made to prevent overlapping of shaded regions—particularly in plots like Fig. 5D—where multiple closely spaced curves are shown with a large number of data points. Using SEM helps maintain visual clarity and distinction between datasets in such cases without impacting the statistical significance of the data.

Reviewer #3

We genuinely thank the reviewer for stating, 'Overall, the experiments are well-executed, and the figures are well-prepared. The reviewer asked us to address a few points that we have attempted, as discussed below.

Major points:

1. The authors observed that the inhibition of Aurora A kinase activity during mitosis leads to misshaped nuclei in newly formed cells. To investigate this effect, they used an AcGFP-tagged, CycB-dependent Aurora A construct that degrades during the metaphase-to-anaphase transition, allowing them to study Aurora A's role in anaphase without interfering with its function during other phases of the cell cycle. In Fig. 1F, they demonstrate the degradation of this construct during anaphase. It would be beneficial to include a comparison showing the behavior of wild-type Aurora A during this phase. While they discuss the advantages of their construct on pages 6-7, it remains an artificial system, which could have implications for protein folding and phase transition.

Response: We thank the reviewer for raising this critical point, which was also highlighted by reviewer 1 (see their major concern #3). In response, we have now examined the localization of Aurora A-AcGFP. As shown in the new Fig. S1I-S1K, HeLa cells expressing Aurora A-AcGFP do not exhibit any significant difference in nuclear shape upon treatment with either control or Aurora A siRNA. These new findings are discussed in the revised manuscript on p. 8.

2. The study reports NuMA accumulation at the poles during anaphase in cells treated with an Aurora A inhibitor and in cells expressing the modified Aurora A construct. Based on these observations, the authors conclude that Aurora A activity during anaphase is required for NuMA dissolution at the poles (Fig. 2).

3. Photobleaching experiments revealed that NuMA mobility depends on Aurora A activity. This dependency also applies to a phosphorylation-resistant NuMA variant, providing strong evidence that phosphorylation at S1969 is necessary for NuMA mobility (Fig. 3).

Response: We thank the reviewer for their recognition of these findings.

4. Figs. 4 and 5 focus on the C-terminal intrinsically disordered region (IDR). The authors employ the Corelet system to investigate phase transition, a well-established approach in this context (>100 orders at Addgene). However, the division of mutations across two figures is suboptimal. It might be clearer to group all immunostaining data together and present the Corelet data separately.

Response: As suggested, we have grouped all the Corelet experiments together. However, for comparison with the impact of these mutations (R>G; Aro>A; Q>G) on spindle pole accumulation of NuMA, we have added the immunostaining data on the same Fig (please see new Fig. 5). We are hopeful that the reviewer is satisfied with this arrangement.

5. The labeling of constructs is somewhat unclear. The use of labels such as (i, ii, etc.) could be replaced with construct names for improved clarity. Additionally, the intensity of immunostainings in Fig. 4B does not appear to correlate with the quantifications in Fig. 4C. In Fig. 4D, "Glu" should be replaced with "Gln." Furthermore, there is no graphical representation of the NuMA-Cterm construct in Fig. 4E, which would be helpful for visualization.

Response: We thank the reviewer for their helpful suggestions. We have improved the clarity of Fig. 4 and Fig. 5. Because of the space constraint and to be consistent between Fig. 4 and Fig. 5- we have not removed the labelling i, ii, etc. Though we have tried to make it more self-explanatory and added a valuable key in the corresponding Fig. legend.

For the immunostaining analysis, we examined multiple samples ($n = 51$). The intensity values shown in Fig. 4C represent the ratio of NuMA signal at the spindle poles relative to the cytoplasm. In our view, the representative images in Fig. 4B accurately reflect the quantified data shown in Fig. 4C. For example, AcGFP-tagged full-length NuMA shows robust pole localization, and this intensity is markedly reduced in constructs lacking either the dynein-binding domain (DBD)

or the coiled-coil domain. Further, as suggested, we have replaced Glu with Gln, and added the NuMA_{Cter} construct in Fig. 5A, thanks.

6. A limitation of the study is that the authors do not demonstrate full-length NuMA accumulation in condensates. Instead, they use a construct consisting of amino acids 1700-2057 (IDR region), which lacks the N-terminal region and 58 C-terminal residues.

Response: We thank the reviewer for their concern. We have now analyzed full-length NuMA lacking the C-terminal 58 amino acids using the Corelet system. This truncation was necessary because the last 58 amino acids contain a DNA-binding domain that anchors NuMA to chromatin, thereby restricting its mobility within the nucleus (Rajeevan et al., 2020; MBoC). Consistent with our previous findings using the NuMA₍₁₇₀₀₋₂₀₅₇₎ fragment, the NuMA₍₁₋₂₀₅₇₎ construct also forms condensates. Interestingly, these condensates appear more stable and less prone to dissolution than those formed by the IDR alone. This indicates that, as seen at spindle poles, the weak multivalent interactions mediated by the IDR are likely stabilized by the adjacent coiled-coil domain.

A broader insight emerging from these findings is that many cellular condensates may utilize stronger structural elements, such as coiled-coil domains, to reinforce and stabilize phase-separated assemblies *in vivo*.

7. The authors then use a Kaede-based system to artificially increase NuMA multivalency, leading to its accumulation in a "non-dynamic pool." However, it is unclear whether this experiment is meant to compensate for the absence of a full-length NuMA experiment using the Corelet system. Additionally, the term "non-dynamic pool" needs clarification-does it correspond to "light-induced condensates"?

Response: We apologize for the lack of clarity in our original writing, which may have led to this confusion. The Corelet experiments were performed in the nucleus during interphase to specifically investigate NuMA-based interactions in the absence of microtubules and other mitotic pole-associated factors. In contrast, the Kaede-based experiments were designed to test whether NuMA-dependent multivalency is a key driver of its abnormal accumulation at spindle poles, and our results support this idea. We have revised the relevant sections in the revised manuscript to better articulate these distinctions and improve the overall clarity of our rationale and experimental design.

8. In Fig. 6, centrinone treatment was used to generate cells with a single centrosome. In these cells, phosphorylated Aurora A localized exclusively at the pole containing SAS6, while NuMA preferentially accumulated at the pole lacking Aurora A. This further supports the conclusion that Aurora A is required for NuMA release during anaphase.

Response: We thank the reviewer for summarizing this part of the results.

9. Fig. 7A/B focuses on nucleoli positioning in newly formed nuclei of Aurora A-inhibited cells. The authors categorize nucleolar distributions as "uniformly spread" or "randomly spread." However, the representative image does not clearly illustrate this distinction, and a statistical evaluation of these distributions is missing.

To assess genome organization in newly formed cells, the authors employed high-resolution 4C-seq. They observed that MLN8237-treated cells exhibited an increased tendency for rDNA clusters to contact the intramolecular arm, leading them to conclude that chromosomes remain more compact. However, the effect appears small and should be discussed in the context of other studies utilizing this technology.

Response: We thank the reviewer for their valuable comments. We have now included a statistical analysis of nucleolar positioning based on three independent experiments, where we have analyzed at least 18 cells each time by confocal live-imaging analysis. As shown in the revised Fig. 8, Nucleoli marked by AcGFP-Fibrillarin staining appear evenly distributed in a significant proportion of control cells. In contrast, in Aurora A-inhibited cells, nucleolar foci show uneven distribution, with a pattern that varies from cell to cell. We now refer to these arrangements as 'unevenly' spread in MLN8237-treated samples. Due to observed variability in the pattern of nucleoli organization in Aurora A-inhibited conditions, we failed to carry out a more structured quantitative analysis, such as measuring distances between nascent nucleoli along the long axis of segregating sister chromatids.

In the context of 4C-seq. analysis, to further better understand the inter-arm contacts, we have now repeated the experiment using an active gene (*FLT3*) on the long arm of chromosome

13, which confirms our previous results about inter-arm contacts. Yet, if compaction was perturbed, we expected to see a loss of long-range long-arm contacts upon MLN-treatment, but this was not detected. We have therefore revised our interpretation accordingly and now focus specifically on changes in inter-arm contacts. Second, to address the reviewer's point about the observed effect's relatively modest nature, we have placed our findings in the broader context of published work. Few studies have employed Chromosome Conformation Capture methods, such as 4C or Hi-C, to analyze chromosome architecture specifically during mitotic exit. On page 22 of the revised manuscript, we now compare our results to the study by Zhang et al., 2019, *Nature*, where Hi-C experiments at different time points during normal mitotic exit revealed changes in the ratio between more local versus long-range contacts, but with overall relatively minor changes at the 10-100 Mb scale as well (Fig. S2 in the referred study).

Minor points:

The study investigates Aurora A function using an artificial construct or Aurora A inhibitors. While the discussion mentions the relevance of Aurora A inhibitors for cancer treatment, the physiological significance of the observed condensates (i.e., whether they exist in vivo) remains unclear.

-We agree with the reviewer that it remains unclear whether patient samples treated with Aurora A inhibitors exhibit abnormal NuMA accumulation at spindle poles during mitosis. However, our observations across multiple cancer (HeLa, U2OS) and non-cancer (HEK, RPE1) cell lines consistently demonstrate NuMA accumulation at the poles upon Aurora A inhibition. Based on these findings, we anticipate that mitotic cells in general may respond similarly to Aurora A inhibitors. Nonetheless, we acknowledge that further validation using patient-derived samples will be necessary to confirm this possibility.

The authors acknowledge that phase transitions are concentration-dependent. However, they should further discuss this in relation to their experimental setup (stable vs. transient transfections) and the impact of mutations (R/Q/Aro) on expression levels.

-This is an important point raised by the reviewer, and we agree that phase separation is concentration-dependent. To address this, in our transient transfection experiments with mutant constructs, we carefully selected cells expressing protein levels comparable to wild-type or endogenous NuMA. Notably, even cells with relatively high expression of the R/Aro mutants fail to localize to the spindle poles, indicating that this defect is due to functional disruption caused by the mutations rather than insufficient protein levels. We have extended similar analyses to the Q mutants.

As also suggested by other reviewers (see point #10 of reviewer 1), we have now included immunoblot data showing expression of the mutant constructs. However, since immunoblots represent pooled cellular material, expression levels can vary between constructs in such assays; that is why we consistently rely on image-based quantification.

The title includes "phase transition," yet this term is not used in the abstract and is rarely mentioned in the main text. Instead, the abstract refers to "dynamic state" or "solid state," while the main text uses terms like "non-dynamic pool," "accumulation," or "condensates." It remains unclear whether the observed phenomenon represents true phase separation or merely a protein assembly based on phosphorylated NuMA. This should be explicitly clarified.

-We appreciate the reviewer's point and would like to clarify our position. In our view, phase transition refers to a phenomenon where the behavior of proteins, protein complexes, or non-membrane-bound organelles changes in a measurable way—manifesting as dynamic or solid states. We fully agree that our observations may not meet the strict criteria for a 'phase separation', at least LLPS, which is why we have now chosen to use the more neutral term 'material properties' to specify the physical nature of poles.

In the context of our study, we show that Aurora A-mediated phosphorylation of NuMA at serine 1969 maintains NuMA in a dynamic state, as demonstrated by FRAP and phosphoconversion analyses. Loss of this phosphorylation alters the material properties of NuMA, leading to a transition toward a more solid-like state. As the reviewer will note, we have made a concerted effort to use consistent, neutral terminology throughout the revised manuscript to avoid confusion and overinterpretation. We hope these clarifications and revisions address the reviewer's concern.

Prof. Sachin Kotak
Indian Institute of Science Bangalore
Microbiology and Cell Biology
C V Raman Avenue
Bangalore, Karnataka 560012
India

8th Aug 2025

Re: EMBOJ-2025-120705R

Aurora A Regulates the Material Property of Spindle Poles to Orchestrate Nuclear Organization at Mitotic Exit

Dear Sachin,

Thank you for submitting your revised manuscript to The EMBO Journal. It has now been re-reviewed by the three original referees, who all on the whole appreciated the revisions and improvements in response to their original comments. Following a final round of further revision to address several remaining specific points raised in the new reports copied below, we shall therefore be happy to eventually accept the study for publication.

In addition to making these final modifications requested by the referees, please also carefully consider our guidelines for formatting and submission of revisions, in particular the following editorial issues:

- Please download (see link below) our author checklist, and upload it in completed form with the final manuscript.
- Please upload all main Figures as individual, image-only files with sufficient resolution/quality for production.
- Please adjust the order and the headers of the different manuscript sections: Title page with complete author information, Abstract, Keywords, Introduction, Results, Discussion, Methods, Data Availability, Acknowledgements, Disclosure and Competing Interests Statement, References, Main Figure Legends, Tables, Expanded Figure Legends.
- Please reduce the number of keywords on the abstract page to five (ideally choosing broad general terms).
- We do not allow referencing of unpublished data or data not shown. Therefore, please remove the reference on page 7 (or consider showing the data as an Appendix figure), as well as on page 26 (unless it was meant to refer to the present results).
- As we are switching from a free-text author contribution statement towards a more formal statement based on Contributor Role Taxonomy (CRediT) terms, please remove the present Author Contribution section and instead specify each author's contribution(s) directly in the Author Information page of our submission system during upload of the final manuscript. See <https://casrai.org/credit/> for more information.
- Please rename the Resource Availability section into "Data Availability", removing the redundant lead contact and materials availability part, and only including information relevant for datasets generated and deposited externally as part of this study, together with a direct hyperlink (for details, please refer to <https://www.embopress.org/page/journal/14602075/authorguide#dataavailability>). We suggest the following format: "The [structural coordinates | microarray | mass spectrometry ...] data from this publication have been deposited to the [name of the database] database [URL] and assigned the identifier [accession | permalink | hashtag]."
- Please remove the Reagents and Tools table from the manuscript text, and upload it separately (making sure to adhere to the table template downloadable from our Guide to Authors).
- Please carefully go through the reference list and make sure that each reference is complete with citation year, volume, and page/locator numbers (currently missing for several of them). Furthermore, please remove the numbering of the list, as explained in our Guide to Authors.
- The "supplementary figure" file should be renamed to "Appendix", uploaded as PDF, and start with a title page containing a header ("Appendix for manuscript xxx") and a brief table of content including page numbers. Also, make sure to rename the figures (both within the Appendix PDF, ToC, legends, and when referencing them in the text) as "Appendix Figure S1-S9".
- Please convert all movies into "Expanded View movies", adjusting the respective in-text callouts to "Movie EV1/2/3...", and moving each movie together with a text file containing its respective legend into a separate ZIP file before re-uploading.

- Please upload the figure source data in separate folders - one archive folder per main figure.

- Please provide suggestions for a short 'blurb' text prefacing and summing up the conceptual aspect of the study in two sentences (max. 250 characters), followed by 3-5 one-sentence 'bullet points' with brief factual statements of key results of the paper; they will form the basis of an editor-written 'Synopsis' accompanying the online version of the article. Please also upload a synopsis image, which can be used as a "visual title" for the synopsis section of your paper. The rather simple image should be in PNG or JPG format, and please make sure that it remains in the modest dimensions of (exactly) 550 pixels wide and 300-600 pixels high.

- Finally, during routine pre-acceptance checks, our data editors have raised the following queries regarding figures, data, and legends, which I would ask you to address (ideally using the Track Changes option):

- 1) Please define the annotated p values ****/***/**/* as well as provide the exact p-values for the same in the legend of figure 1J, K; 2B, F, H; 3C, D; 4B, E; 5G, 6B, 7D, G as appropriate.
- 2) Please indicate the statistical test used for data analysis in the legends of figures 1J, K; 2B, F, H; 3C, D; 4B, E; 5G, 6B, 7D, G
- 3) Please note that the error bars need to be defined in the legend of figure 7G.
- 4) Please note that the measure of center for the error bars needs to be defined in the legends of figures 4B, 8B
- 5) Please note that the scale bar needs to be defined for figures 1G, I; 2E, G; 7E, F

I am therefore returning the manuscript to you now for a final round of minor revision, to allow you to make all these adjustments and upload all modified files. Once we will have received them, we should hopefully be able to swiftly proceed with acceptance and publication of the manuscript. Please do not hesitate to get back to us in case you should have any questions in this regard.

With kind regards,

Hartmut

7) All authors listed as (co-)corresponding need to deposit, in their respective author profiles in our submission system, a unique

ORCID identifier linked to their name. Please see our Guide to Authors for detailed instructions.

9) To facilitate reproducibility and cross-laboratory adoption of methodologies, please structure the Materials & Methods section as outlined in our guide to authors, including a completed Reagents and Tools Table that can be downloaded from our author guidelines as well (<https://www.embopress.org/page/journal/14602075/authorguide#structuredmethods>).

10) Digital image enhancement is acceptable practice, as long as it accurately represents the original data and conforms to community standards. If a figure has been subjected to significant electronic manipulation, this must be clearly noted in the figure legend and/or the 'Materials and Methods' section. The editors reserve the right to request original versions of figures and the original images that were used to assemble the figure. Finally, we generally encourage uploading of numerical as well as gel/blot image source data; for details see: embopress.org/page/journal/14602075/authorguide#sourcedata

In the interest of ensuring the conceptual advance provided by the work, we recommend submitting a revision within 3 months (6th Nov 2025). Please discuss the revision progress ahead of this time with the editor if you require more time to complete the revisions. Use the link below to submit your revision:

Link Not Available

Referee #1:

Aurora A regulates the material property of spindle poles to orchestrate nuclear organization at mitotic exit

In this manuscript, the authors investigate the functions of Aurora A late in mitosis, focusing particularly on its role in regulating NuMA at the spindle poles. To distinguish early from late mitotic functions, they ingeniously construct a Cyclin B1 D-box-Aurora A fusion protein that is selectively degraded during metaphase/anaphase. This system ensures that Aurora A activity is absent as cells exit mitosis. Loss of Aurora A activity at this stage leads to misshaped nuclei and persistent association of NuMA with spindle poles. It is the persistent association of NuMA with spindle poles that causes the misshaped nuclei.

The authors further demonstrate that Aurora A enhances the dynamic behavior of NuMA through phosphorylation. Additionally, they identify that the accumulation of NuMA at spindle poles is mediated by a combination of its dynein-binding site, central coiled-coil domain, and C-terminal intrinsically disordered region (IDR). Within the IDR, the authors pinpoint arginine (R) and aromatic (Aro) residues as critical for NuMA's pole localization.

In the final section of the manuscript, the authors explore whether NuMA's mitotic activity influences interphase genome organization. Using 4C-seq targeting rDNA, they detect changes in intra-chromosomal contacts, suggesting potential long-term nuclear consequences of disrupted NuMA regulation.

Overall, the manuscript presents a compelling and previously underappreciated role for Aurora A in modulating NuMA function during late mitosis. The approaches described provide a useful framework for dissecting temporal roles of mitotic regulators in late mitosis. While the study is significantly improved and suitable for publication in EMBO Journal, the authors should further clarify the broader significance of Aurora A-mediated regulation of NuMA (e.g., Figure 8). I also recommend a few minor editorial revisions (see below).

Specific points:

p. 11: "AuroraA activity in anaphase is essential to dissolve NuMA at poles" - I suggest to change to: "AuroraA activity in anaphase is important to release NuMA from the spindle poles." Even without AuroraA NuMA slowly is released from spindle poles (Figure 2H).

Fig. 3G: Description is based on one example. It would be helpful to add an additional example for NuMA at poles in MNLN8237

treated cells.

p. 14: "undergoes phase transition" - what they have shown up to this part in the manuscript is that NuMA becomes less dynamic.

Fig. 5C: it would be helpful to show enlargement's of the 58 s timepoint.

p. 19: "did not significantly accumulate at the poles (Fig. 5E-G)". I guess the authors mean does not stronger accumulate at the poles compared to NuMA (Q>G).

Fig. 5G: significant test between (i) and (v), (vi) and (viii) should be done.

Start with a new paragraph on p 17: Next, we sought to examine the

p. 19: "Thus, we conclude that glutamine residues in the IDR of NuMA are key drivers to ensuring". I would not call them key drivers. Compared to R and Aro residues, Q residues have a relatively mild impact on the pole localization of NuMA. Please rephrase

Discussion: the authors did not discuss the role of the CDK1 phosphorylation on chromatin association in the overall context of NuMA localization to spindle poles as suggested. In the Discussion, they strongly emphasize the role of glutamine residues in the IDR without mentioning arginine and aromatic amino acids.

Referee #2:

The authors have carefully addressed all comments of the reviewers and added additional experiments and considerably improved the presentation of the data and the clarity of the text. The Introduction has also been nicely improved so that the open questions can now be clearly stated.

The main findings of the study are of interest to researchers in the cell division/spindle field, as the authors dissect in more detail the mechanism of the previously reported NuMA overaccumulation at spindle poles in the absence of Aurora A activity and some of its consequences (chromosome decondensation/nuclear reformation). This work fills some gaps left from previous work, clarifies some discrepancies in the literature which is useful, and provides more detail about the molecular mechanism of NuMA's pole accumulation and its control by Aurora A. The study is very comprehensive, the quality of the experiments is high, and the interpretations and conclusions are well-supported by the data, balanced, and the findings are appropriately put in the context of the literature.

This reviewer has only three remaining concerns that can be addressed by editing the text:

(1) On page 9, the authors state now clearly that under their Aurora A inhibition conditions they still obtain bipolar spindles. As the Introduction did not introduce that under previous conditions generally monopolar spindles were observed upon Aurora A inhibition, for clarity the authors might want to state in their Results section more explicitly why their condition leads to a different phenotype than that expected from the literature.

(2) On page 14, the authors summarize 3 conditions of recent LLPS experiments. They could be clearer here by stating clearly which of these conditions refer to experiments with purified proteins and which ones to experiments in cells. It would maybe also be helpful to indicate what the hexanediol is used for as this sounds like a very technical point now accessible only to readers of the LLPS literature. In the same paragraph, the authors state that these conditions can lead to misinterpretations - which is correct, but is also a fairly bold statement. So maybe they could indicate which sort of misinterpretations could arise and why their experimental approach avoids this problem.

(3) The authors report in their experiments shown in Fig. 4D/E to which extent a certain NuMA construct localizes to/accumulates at spindle poles. It seems important to clarify whether pole localization means also rescue of pole formation or not. This is because one expects that knockdown of endogenous NuMA will cause disruption of bipolar spindles (turbulent state described in papers from the Dumont lab, i.e. no poles). If pole localization is observed, does it mean that the construct also is able to rescue pole formation? Or is the knockdown sufficiently incomplete that poles can still form and the assay only tests whether constructs can localize to poles under conditions where levels of endogenous dynein/NuMA are sufficient to make poles?

Referee #3:

In response to the reviewer's suggestions, the authors incorporated additional controls and reorganized the figures. These

revisions enhanced the manuscript's readability and provided a more robust foundation for the conclusions. Overall, this is a well-executed study in cell biology that deepens our understanding of NuMA's function in spindle pole organization and merits publication in The EMBO Journal.

Minor points:

Fig. 3: Scale bars mm instead of micrometer

Fig. 7: "anti T288P" should probably read "anti-Aurora A T288P"

Response to the reviewers**Reviewer #1**

We thank the reviewer for supporting our work. However, the reviewer made a few specific points, which we have addressed, as explained below.

Specific points:

p. 11: "AuroraA activity in anaphase is essential to dissolve NuMA at poles" - I suggest to change to: "AuroraA activity in anaphase is important to release NuMA from the spindle poles." Even without AuroraA NuMA slowly is released from spindle poles (Figure 2H).

Response. The sentence is now corrected, thanks for this suggestion.

Fig. 3G: Description is based on one example. It would be helpful to add an additional example for NuMA at poles in MLN8237 treated cells.

Response. Thanks for this point. In the manuscript, we have presented representative images of metaphase and anaphase cells in control and MLN8237-treated conditions by analyzing more than 10 cells in each condition. Prompted by this comment, we have now included an additional example of NuMA organization at the pole in MLN8237-treated cells (shown below). This data is now added to Appendix Figure S3.

Representative images from the super-resolution 3D-SIM analysis of HeLa cells immunostained with anti-NuMA (green) and anti-PCNT (magenta) during metaphase and anaphase upon acute treatment with MLN8237. Insets on the right show the pole localized NuMA (grey) and PCNT (magenta). Related to Fig 3G. $n \geq 10$ cells were analysed for control and MLN8237 in metaphase and anaphase. Scale bars represent 5 μm for the cell and 2 μm for the insets.

p. 14: "undergoes phase transition" - what they have shown up to this part in the manuscript is that NuMA becomes less dynamic.

Response. Thanks, we have reworded the sentence.

Fig. 5C: it would be helpful to show enlargement's of the 58 s timepoint.

Response. We thank the reviewer for this suggestion. However, we believe the data are already clearly visible in the high-resolution panel of Fig. 5C, enabling readers to readily assess whether the condensates have dissolved. In addition, due to space constraints in Fig. 5 and to maintain consistency across all time points, we have chosen not to include an enlarged view of the 58 s time point in Fig. 5C.

p. 19: "did not significantly accumulate at the poles (Fig. 5E-G)". I guess the authors mean does not stronger accumulate at the poles compared to NuMA (Q>G).

Response. Yes, we were referring that AcGFP-NuMA(S1969A;Q>G) does not strongly accumulate at the poles, compared to AcGFP-NuMA(Q>G). We have now slightly reworded this sentence to make it clearer.

Fig. 5G: significant test between (i) and (v), (vi) and (viii) should be done.

Response. Added, thanks.

Start with a new paragraph on p 17: Next, we sought to examine the

Response. Thanks, done.

p. 19: "Thus, we conclude that glutamine residues in the IDR of NuMA are key drivers to ensuring". I would not call them key drivers. Compared to R and Aro residues, Q residues have a relatively mild impact on the pole localization of NuMA. Please rephrase

Response. The sentence has been rephrased as suggested, thanks.

Discussion: the authors did not discuss the role of the CDK1 phosphorylation on chromatin association in the overall context of NuMA localization to spindle poles as suggested.

Response. We thank the reviewer for raising this point, which extends their earlier comment. As per the reviewer's suggestion, we have already included in the Introduction (p. 4) the relevance of Cdk1 phosphorylation for NuMA localization at spindle poles. Specifically, we stated: "*The accumulation of NuMA at the poles is regulated by mitotic kinases: Cdk1-mediated phosphorylation promotes NuMA enrichment at the poles, while Aurora A-mediated phosphorylation maintains NuMA in a dynamic state.*" In this work, we are primarily focusing on the role of Aurora A in keeping NuMA at the poles in a dynamic state during mitotic exit. To maintain a clear and focused narrative, we have not elaborated here on the impact of Cdk1-mediated phosphorylation on releasing NuMA from the chromatin during mitotic entry. We feel that mentioning this in the discussion section could potentially distract our readers from our main focus: Aurora A-mediated regulation of NuMA at the poles during mitotic exit.

In the Discussion, they strongly emphasize the role of glutamine residues in the IDR without mentioning arginine and aromatic amino acids.

Response. We respectfully note that the revised manuscript already addresses the role of arginine and aromatic amino acid residues. Specifically, on p. 25 we state: "*the coiled-coil domain structurally stabilizes weak cation- π interactions, specifically between arginine and aromatic residues within the IDR sequence.*" In addition, on p. 27 we discuss that "*arginine and aromatic residues present in the NuMA IDR may also promote its transition from dynamic to solid upon Aurora A inhibition. However, since these residues are also required for NuMA accumulation at the poles, we could not directly assess their relevance in promoting solid-state transition.*" We hope this clarifies that these points were already included in the revised version.

Reviewer #2

We sincerely thank the reviewer for their positive feedback on our work. Below, we address the three specific points they raised in detail and have revised the manuscript accordingly to ensure these points are adequately incorporated where relevant.

(1) On page 9, the authors state now clearly that under their Aurora A inhibition conditions they still obtain bipolar spindles. As the Introduction did not introduce that under previous conditions generally monopolar spindles were observed upon Aurora A inhibition, for clarity the authors might want to state in their Results section more explicitly why their condition leads to a different phenotype than that expected from the literature.

Response. We thank the reviewer for this point. As suggested, we have added a sentence in the Results section explaining why our experimental conditions are particularly relevant for studying Aurora A's role during anaphase—a context that could not be addressed earlier using prolonged inhibition with MLN8237.

(2) On page 14, the authors summarize 3 conditions of recent LLPS experiments. They could be clearer here by stating clearly which of these conditions refer to experiments with purified proteins and which ones to experiments in cells. It would maybe also be helpfull to indicate what the hexanediol is used for as this sounds like a very technical point now accessible only to readers of the LLPS literature. In the same paragraph, the authors state that these conditions can lead to misinterpretations - which is correct, but is also a fairly bold statement. So maybe they could indicate which sort of misinterpretations could arise and why their experimental approach avoids this problem.

Response. We thank the reviewer for this point. As requested, we have clarified which previous experiments were performed with recombinant NuMA in the presence of a crowding reagent and which were conducted in cells. We have also provided the caveats of interpreting hexanediol-based data. In addition, we have slightly revised our sentence to make it less of a bold statement. To address potential misinterpretations, we cited McSwiggen et al., 2019, Alberti et al., 2019; and Hedtfeld et al., 2024, which discuss the limitations of linking observations from purified proteins or overexpression directly to LLPS.

(3) The authors report in their experiments shown in Fig. 4D/E to which extent a certain NuMA construct localizes to/accumulates at spindle poles. It seems important to clarify whether pole localization means also rescue of pole formation or not. This is because one expects that knockdown of endogenous NuMA will cause disruption of bipolar spindles (turbulent state

described in papers from the Dumont lab, i.e. no poles). If pole localization is observed, does it mean that the construct also is able to rescue pole formation? Or is the knockdown sufficiently incomplete that poles can still form and the assay only tests whether constructs can localize to poles under conditions where levels of endogenous dynein/NuMA are sufficient to make poles?

Response. We thank the reviewer for raising this point. The turbulent state described in the Dumont lab paper (Hueschen et al., 2019, *Curr. Biol.*, Fig. 1) was observed in RPE1 cells where NuMA was knocked out using an inducible CRISPR–Cas9 system. In contrast, strong NuMA depletion via auxin-induced degradation in HT116 cells produces a less pronounced phenotype (Tsuchiya et al., 2021, *Curr. Biol.*, Fig. 1). In our study, expression of several NuMA-based constructs that fail to localize to spindle poles—following endogenous NuMA depletion—results in substantial chromosome instability (40–90%, Fig. EV4F). Given the direct correlation between impaired pole localization and chromosome instability in these mutants, we believe that defective pole localization is the primary driver of the observed phenotypes. As the reviewer noted, the relatively milder phenotype compared to Hueschen et al., could be due to 1) the use of HeLa (this study) and HT116 (Tsuchiya et al., 2021) cells instead of RPE1, or 2) Residual NuMA levels in our system and Tsuchiya et al., 2021. We have now briefly discussed this point in the revised legend for Fig. EV4F. In future work, live-cell imaging of these mutants will allow us to directly link their pole localization defects with specific spindle assembly phenotypes.

Reviewer #3

We are deeply thankful to the reviewer for their kind words. We have now addressed to remaining 2 minor points raised by the reviewer.

Fig. 3: Scale bars mm instead of micrometer

Response. Thanks, fixed

Fig. 7: "anti T288P" should probably read "anti-Aurora A T288P"

Response. We have initially defined on p. 7 and 9 that Aurora A phosphorylated at T288 is referred to as T288^P. Thus, the nomenclature for the antibody against Aurora A phosphorylated at T288 as anti-T288^P appears correct in Fig. 7.

Prof. Sachin Kotak
Indian Institute of Science Bangalore
Microbiology and Cell Biology
C V Raman Avenue
Bangalore, Karnataka 560012
India

26th Aug 2025

Re: EMBOJ-2025-120705R1

Aurora A regulates the material property of spindle poles to orchestrate nuclear organization at mitotic exit

Dear Sachin,

Thank you for submitting your final revised manuscript for our consideration, and please excuse the holiday-related delay in processing it. I am pleased to inform you that we have now accepted it for publication in The EMBO Journal.

With kind regards,

Hartmut
